# ADAPTIVE INSTRUMENT DESIGN FOR INDIRECT EXPERIMENTS

**Yash Chandak**
Stanford University

**Shiv Shankar**
UMass Amherst

**Vasilis Syrgkanis**
Stanford University

**Emma Brunskill**
Stanford University

## ABSTRACT

Indirect experiments provide a valuable framework for estimating treatment effects in situations where conducting randomized control trials (RCTs) is impractical or unethical. Unlike RCTs, indirect experiments estimate treatment effects by leveraging (conditional) instrumental variables, enabling estimation through encouragement and recommendation rather than strict treatment assignment. However, the sample efficiency of such estimators depends not only on the inherent variability in outcomes but also on the varying compliance levels of users with the instrumental variables and the choice of estimator being used, especially when dealing with numerous instrumental variables. While adaptive experiment design has a rich literature for *direct* experiments, in this paper we take the initial steps towards enhancing sample efficiency for *indirect* experiments by adaptively designing a data collection policy over instrumental variables. Our main contribution is a practical computational procedure that utilizes influence functions to search for an optimal data collection policy, minimizing the mean-squared error of the desired (non-linear) estimator. Through experiments conducted in various domains inspired by real-world applications, we showcase how our method can significantly improve the sample efficiency of indirect experiments.

## 1 INTRODUCTION

Advances in machine learning, especially from large language models, are greatly expanding the potential of AI to augment and support humans in an enormous number of settings, such as helping direct teachers to students unproductively struggling while using math software (Holstein et al., 2018), providing suggestions to novice customer support operators (Brynjolfsson et al., 2023), and helping peer volunteers learn to be more effective mental health supporters (Hsu et al., 2023). AI-augmentation will often (importantly) provide autonomy to the human, who may consider information or suggestions provided by an AI, before ultimately making their own choice about how to proceed (we provide examples of more use-cases in Appendix B). In such settings it will be highly informative to be able to separate estimating the strength of the intervention on humans' treatment choices, as well as the actual treatment effects (the outcomes if the human chooses to follow the treatment of interest), instead of estimating only intent-to-treat effects. This estimation problem becomes more challenging given the likely existence of latent confounders, that may influence both whether a person is likely to uptake a particular recommendation (such as following a large language model's recommendation of what to say to a customer), and their downstream outcomes (on customer satisfaction). Fortunately, if the AI intervention satisfies the requirements of an instrumental variable (Hartford et al., 2017; Syrgkanis et al., 2019), it is well-known when and how consistent estimates of the treatment effect estimates can be obtained.

In this paper, we consider how to automatically learn data-efficient adaptive instrument-selection decision policies, in order to quickly estimate conditional average treatment effects. While adaptive experiment design is well-studied when direct experimentation is feasible (Murphy, 2005; Xiong et al., 2019; Chaloner & Verdinelli, 1995; Rainforth et al., 2023; Che & Namkoong, 2023; Hahn et al., 2011; Kato et al., 2020; Song et al., 2023), it remains largely unexplored in situations where direct assignment to interventions is impractical, unethical or undesirable, such as for the real-time AI support of teachers, customer support agents, and volunteer mental health support trainees. Some work has considered adaptive data collection to minimize regret when using instrument variables (Kallus, 2018; Della Vecchia & Basu, 2023), which also relates to work in the multi-armed bandit literature. Other work has considered when user compliance may be dynamic, and shaped over

Figure 1: In randomized control trials (direct experiments), there is no unobserved confounding, i.e., the value of the treatment is the same as that of the instrument. Using adaptive experiment design strategy of RCTs naively for indirect experiments, i.e., when confounding does exists, can result in performance worse than the uniform sampling strategy. In contrast, the proposed algorithm for designing instruments adaptively (DIA) is more effective for indirect experiments.

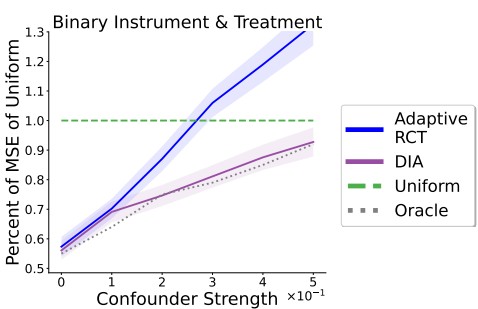

time (Ngo et al., 2021). In contrast, our work focuses on estimating treatment effects through the use of adaptive allocation of instruments, but the impact of those instruments is assumed to be static, though unknown. As Figure 1 demonstrates, we will shortly show that it is possible to create adaptive instrumental strategies that far outperform standard uniform allocation. In particular, our paper address this challenge of adaptive instrument design through two main avenues:

**1. A general framework for adaptive indirect experiments:** We envision modern scenarios where available instruments could be high-dimensional, even encompassing natural language (e.g., personalised texts to encourage participation in treatment or control groups). This necessitates modeling the sampling procedure with rich function approximators. Taking the first steps towards this goal, we do restrict ourselves to relatively small synthetic and semi-synthetic experiments but prioritize understanding the fundamental challenges of creating such a flexible framework.

In Section 3 we introduce influence functions estimators for black-box double causal estimators and a multi-rejection sampler. These tools enable us to perform a gradient-based optimization of a data collection policy $\pi$. Importantly, the proposed method is flexible to support $\pi$ that is modeled using deep networks, and is applicable to various (non)-linear (conditional) IV estimators.

**2. Balancing sample complexity and computational complexity:** Experiment design is most valuable when sample efficiency is paramount as data is relatively costly in terms of time or resources compared to computation. Therefore, we advocate for leveraging computational resources to enhance the search for an optimal data collection strategy $\pi$. Our framework is designed to minimize the need for expert knowledge and can readily scale with computational capabilities.

Perhaps the most relevant is the work by Gupta et al. (2021) for enhancing sample efficiency for a specific (linear) IV estimator by learning a sampling distribution over a few *data sources*, where each source provides different moment conditions. This is complementary to our work as we consider the problem of adaptive sampling, even within a single data source, and using potentially non-linear estimators. Therefore, a detailed related work discussion is deferred to Appendix A.

## 2  BACKGROUND AND PROBLEM SETUP

Let $X \in \mathcal{X}$ be the observable covariates, $U \in \mathcal{U}$ be the unobserved confounding variables, $Z \in \mathcal{Z}$ be the instruments, $A \in \mathcal{A}$ be the actions/treatments, and $Y \in \mathbb{R}$ be the outcomes. Let $\pi : \mathcal{X} \to \Delta(\mathcal{Z})$ be an instrument sampling policy, and $\Pi$ be the set of all such policies. Let $D_n := \{S_i\}_{i=1}^n$ be a dataset of size $n$, where each data sample $S_i := \{X_i, Z_i, A_i, Y_i\}$ is collected using (potentially different) policy $\pi_i \in \Pi$. With the model given in Figure 2, we assume that the causal relationship can be represented with the following structural model:

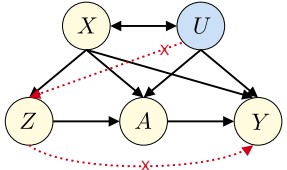

Figure 2: Causal graph.

$$Y = g(X, A) + U, \quad \mathbb{E}[U] = 0, \quad \mathbb{E}[U|A, X] \neq 0, \tag{1}$$

where $g$ is an unknown (potentially non-linear) function. Importantly, note that the variable $U$ can affect not only the outcomes $Y$ but also $X$ and $A$. We consider treatment effects to be homogenous across instruments, i.e., the CATE matches the conditional LATE (Angrist et al., 1996).

Following Newey & Powell (2003) and Hartford et al. (2017), we define the counterfactual prediction as $f(x, a) := g(x, a) + \mathbb{E}[U|x]$, where $\mathbb{E}[U|x]$ corresponds to the fixed baseline effect across all treatments $a \in \mathcal{A}$, for a given covariate $x \in \mathcal{X}$. This definition of $f$ is useful, as it can be identified (discussed later) and can also be used to determine the effect of treatment $a_1 \in \mathcal{A}$ compared to

another treatment $a_2 \in \mathcal{A}$ as $f(x, a_1) - f(x, a_2) = g(x, a_1) - g(x, a_2)$. Further, in the simpler setting, if $U \perp\!\!\!\perp X$, then $f(x, a) = g(x, a)$.

In the supervised learning setting (which assumes $U \perp\!\!\!\perp A, X$), the prediction model estimates $\mathbb{E}[Y|x, a]$, however, this can result in a biased (and inconsistent) treatment effect estimate as $\mathbb{E}[Y|x, a] = g(x, a) + \mathbb{E}[U|x, a] \neq f(x, a)$. This issue can often be alleviated using *instruments*. For $Z$ to be a valid instrument, $Z$ should satisfy the following assumptions (red arrows in Fig 2),

**Assumption 1.** **(a) Exclusive:** $Z \perp\!\!\!\perp Y|X, A$, *i.e., instruments do not effect the outcomes directly.* **(b) Relevance:** $Z \not\perp\!\!\!\perp A|X$, *i.e., the action chosen can be influenced using instruments.* **(c) Independence:** $Z \perp\!\!\!\perp U|X$, *i.e., the unobserved confounder does not affect the instrument.*

Under Assumption 1, an inverse problem can be designed for estimating $f$, with the help of $z$,

$$\mathbb{E}[Y|x, z] = \mathbb{E}[g(x, A)|x, z] + \mathbb{E}[U|x] = \mathbb{E}[f(x, A)|x, z] = \int_a f(x, a)\mathrm{d}P(a|x, z). \quad (2)$$

However, identifiability of $f$ using 2 depends on specific assumptions. In the simple setting where $f$ is linear and $\mathbb{E}[U|x, z] = 0$, Angrist et al. (1996) discuss how two-stage least-squares estimator (2SLS) can be used to identify $f$. In contrast, if $f$ is non-linear (or non-parametric), exact recovery of $f$ may not be viable as equation 2 results in an ill-posed inverse problem (Newey & Powell, 2003; Xu et al., 2020). Different techniques have been proposed to mitigate this issue (Newey & Powell, 2003; Xu et al., 2020; Syrgkanis et al., 2019; Frauen & Feuerriegel, 2022); see Wu et al. (2022) for a survey. These methods often employ a two-stage procedure, where each stage solves a set of moment conditions. We formalize these below.

Let $f : \mathcal{X} \times \mathcal{A} \to \mathbb{R}$ be parametrized using $\theta \in \Theta$, where $\Theta$ is the set of all parameters. Let $\theta_0 \in \Theta$ be the true parameter for $f$. Let $q(S_i; \phi)$ and $m(S_i; \theta, \phi)$ be the moment condition for nuisance parameters $\phi$ and parameters of interest $\theta$, respectively, for each sample $S_i$. Let $M_n$ and $Q_n$ be the corresponding sample average moments such that,

$$M_n\left(\theta, \hat{\phi}\right) := \frac{1}{n}\sum_{i=1}^{n} m(S_i; \theta, \hat{\phi}), \qquad Q_n(\phi) := \frac{1}{n}\sum_{i=1}^{n} q(S_i; \phi), \quad (3)$$

where $\hat{\phi}$ is the solution to $Q_n(\phi) = \mathbf{0}$, and the estimated parameters of interest $\hat{\theta}$ are obtained as a solution to $M_n(\theta, \hat{\phi}) = \mathbf{0}$. For example, in the basic 2SLS setting without covariates $X_i$ (Pearl, 2009), first stage moment $q(S_i; \phi) := \nabla_\phi(Z_i\phi - A_i)^2$ and the second stage moment $m(S_i; \theta, \hat{\phi}) := \nabla_\theta(\tilde{A}_i\theta - Y_i)^2$, where $\tilde{A}_i = Z_i\hat{\phi}$. Similarly, to solve 2 in the non-linear setting with deep networks, Hartford et al. (2017) use $q(S_i; \phi) := -\nabla_\phi \log \Pr(A_i|X_i, Z_i; \phi)$ to estimate density $\mathrm{d}P(A_i|X_i, Z_i; \phi)$ using a neural density estimator, and subsequently define $m(S_i; \theta, \hat{\phi}) := \nabla_\theta(Y_i - \int f(X_i, a; \theta)\mathrm{d}P(a|X_i, Z_i; \hat{\phi}))^2$, where $f$ is also a neural network parametrized by $\theta$.

To make the dependency of the estimated parameters of $f$ on the data explicit, we denote the estimated parameter as $\theta(D_n)$. Further, we assume that the $\theta(D_n)$ is symmetric in $D_n$, i.e., $\theta(D_n)$ is invariant to the ordering of samples in the dataset $D_n$. This holds for 2SLS, DeepIV (Hartford et al., 2017), and most other popular estimators (Syrgkanis et al., 2019; Wu et al., 2022). We also consider $\theta(D_n)$ to be deterministic for a given $D_n$. In Appendix H.2, we discuss how the proposed algorithm can be used as-is for the setting where $\theta(D_n)$ is also stochastic given $D_n$.

**Problem Statement:** While these estimators mitigate the confounder bias in estimation of $f$, they are often subject to high variance (Imbens, 2014). In this work, we aim to develop an adaptive data collection policy that can improve sample-efficiency of the estimate of the counterfactual prediction $f$. Importantly, we aim to develop an algorithmic framework that can work with general (non)-linear two-stage estimators. Specifically, let $X, A \sim \mathbb{P}_{x,a}$ be samples from a *fixed* distribution over which the expected mean-squared error is evaluated,

$$\mathcal{L}(\pi) := \mathop{\mathbb{E}}_{D_n \sim \pi}[\mathrm{MSE}(\theta(D_n))], \quad \text{and } \mathrm{MSE}(\theta) := \mathbb{E}_{X,A}\left[(f(X, A; \theta_0) - f(X, A; \theta))^2\right]. \quad (4)$$

To begin, we aim to find a *single* policy $\pi^* \in \arg\min_{\pi \in \Pi} \mathcal{L}(\pi)$ for the most sample-efficient estimation of $\theta$. In Section 3.3, we will adapt this solution for sequential collection of data.

## 3 APPROACH

As Figure 1 illustrates, selecting instruments strategically to generate a dataset can have a substantial impact on the efficiency of indirect experiments. Importantly, as $\mathcal{L}(\pi)$ is a function of both the data $D_n$ and the estimator $f(\cdot; \theta)$, an optimal $\pi^*$ not only depends on the data-generating process (e.g., heteroskedasticity in compliances $Z|X$ and outcomes $Y$, even given a specific covariate $X$) but also on the actual procedure used to compute an estimate of the parameter of interest $\theta$, such as 2SLS, etc. In Appendix C, we present a simple illustrative case of linear models for the conditional average treatment effect (CATE) estimation, where each aspect can be observed explicitly.

Now we provide the key insights for a general gradient-based optimization procedure to find an instrument selection decision policy $\pi^*$ that tackles all of these problems simultaneously. A formal analysis of the claims and the algorithmic approaches we make, and how they contrast with some other alternate choices, will be presented later in Section 4.

As our main objective is to search for $\pi^*$, we propose computing the gradient of $\mathcal{L}(\pi)$, which we can express as the following using the popular REINFORCE approach (Williams, 1992),

$$\nabla \mathcal{L}(\pi) = \mathop{\mathbb{E}}_{D_n \sim \pi} \left[ \text{MSE}(\theta(D_n)) \frac{\partial \log \Pr(D_n; \pi)}{\partial \pi} \right] \stackrel{(a)}{=} \mathop{\mathbb{E}}_{D_n \sim \pi} \left[ \text{MSE}(\theta(D_n)) \sum_{i=1}^{n} \frac{\partial \log \pi(Z_i|X_i)}{\partial \pi} \right] \quad (5)$$

where $(a)$ follows because $\Pr(D_n; \pi)$ is the probability of observing the entire dataset $D_n$,

$$\Pr(D_n; \pi) = \prod_{i=1}^{n} \Pr(X_i, U_i) \pi(Z_i|X_i) \Pr(A_i|X_i, U_i, Z_i) \Pr(Y_i|X_i, U_i, A_i). \quad (6)$$

The gradient in 5 is appealing as it provides an unbiased direction to update $\pi$, and encapsulates properties of both the data-generating process and the estimators as a black-box in $\text{MSE}(\theta(D_n))$. However, using $\nabla \mathcal{L}(\pi)$ for optimization is impractical due to two main challenges:

- **Challenge 1:** As we will state more formally in Section 4, the variance of sample estimate of $\nabla \mathcal{L}(\pi)$ can be $\Theta(n)$, i.e., it can grow *linearly* in the size of $D_n$. This can make the optimization process prohibitively inefficient as we collect more data.

- **Challenge 2:** Optimization using 5 will also require evaluating $\nabla \mathcal{L}(\pi)$ for a $\pi$ different from $\pi'$ that was used to collect the data. If off-policy correction is done using importance weighting (Precup, 2000), then the variance might grow *exponentially* with $n$.

To address these two issues we now demonstrate how *influence functions* and *multi-rejection importance sampling* can be used to leverage the structure in our indirect experiment design setup to not only significantly reduce the variance of the resulting gradient estimate, but also do so in a computationally practical manner that is compatible with deep neural networks. The pseudocode for our algorithm approach is shown in Appendix H.

### 3.1 INFLUENCE FUNCTIONS FOR CONTROL VARIATES

To address challenge 1, a common approach in reinforcement learning and other areas is to introduce control variates, that reduce the variance of estimation without introducing additional bias. In particular, we observe that for our setting, using the MSE computed using all other data points $D_{n \setminus i}$ (i.e., all of $D_n$, except the sample $S_i$ used in the $\log \pi(Z_i|X_i)$ term) serves as a control variate,

$$\hat{\nabla}^{\text{CV}} \mathcal{L}(\pi) := \sum_{i=1}^{n} \frac{\partial \log \pi(Z_i|X_i)}{\partial \pi} \big( \text{MSE}(\theta(D_n)) - \text{MSE}\big(\theta(D_{n \setminus i})\big) \big). \quad (7)$$

As we illustrate in Figure 3 and formalize in Section 4, this immediately results in a substantial variance reduction over the originally proposed gradient (5). However, the computation required for estimating $\hat{\nabla}^{\text{CV}} \mathcal{L}(\pi)$ can be prohibitively expensive as now for *each* $\nabla \log \pi(Z_i|X_i)$ in 7, an entire separate re-training process is required to estimate the control variate $\text{MSE}(\theta(D_{n \setminus i}))$.

To be efficient in terms of *both* the variance and the computation, we now show that $\text{MSE}(\theta(D_n)) - \text{MSE}\big(\theta(D_{n \setminus i})\big)$ can be estimated using influence functions (Fisher & Kennedy, 2021). As we discuss below, this requires only a *single* optimization process for estimating $\text{MSE}(\theta(D_n)) - \text{MSE}(\theta(D_{n \setminus i}))$ across *all* $i$'s, and thus preserves the best of both $\hat{\nabla}^{\text{CV}} \mathcal{L}(\pi)$ and $\hat{\nabla} \mathcal{L}(\pi)$.

Figure 3: Bias-variance of different gradient estimators, for a given policy $\pi$, when using a 2SLS estimators under model-misspecification. **Observe the scale on the y-axes.**

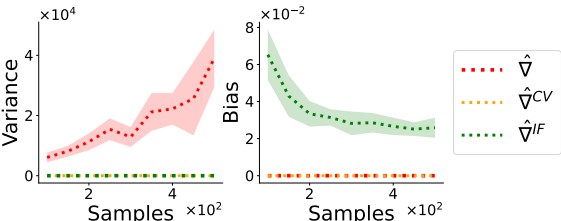

Let $\mathbb{D}$ be the distribution function induced by the dataset $D_n$, and with a slight overload of notation, we let $\theta(D_n) \equiv \theta(\mathbb{D})$. Let $\mathbb{D}_{i,\epsilon} := (1 - \epsilon)\mathbb{D} + \epsilon\delta(S_i)$ be a distribution perturbed in the direction of $S_i$, where $\delta(\cdot)$ denotes the Dirac distribution. Note that for $\epsilon = -1/n$, distribution $\mathbb{D}_{i,\epsilon}$ corresponds to the distribution function without the sample $S_i$, i.e., the distribution induced by $D_{n \setminus i}$. From this perspective, $\mathrm{MSE}(\theta(\cdot))$ corresponds to a *functional* of the data distribution function $\mathbb{D}$, and thus we can do a Von Mises expansion (Fernholz, 2012), i.e., a distributional analog of the Taylor expansion for statistical functionals, to directly approximate $\mathrm{MSE}(\theta(\mathbb{D}_{i,\epsilon}))$ in terms of $\mathrm{MSE}(\theta(\mathbb{D}))$ as

$$\mathrm{MSE}(\theta(\mathbb{D}_{i,\epsilon})) = \mathrm{MSE}(\theta(\mathbb{D})) + \sum_{k=1}^{\infty} \frac{\epsilon^k}{k!} \mathcal{I}_{\mathrm{MSE}}^{(k)}(S_i), \tag{8}$$

where $\mathcal{I}_{\mathrm{MSE}}^{(k)}$ is the $k$-th order *influence function* (see the work by Fisher & Kennedy (2021); Kahn (2015) for an accessible introduction to influence functions). Therefore, when all the higher order influence function exists, we can use 8 to recover an estimate of $\mathrm{MSE}(\theta(D_n)) - \mathrm{MSE}(\theta(D_{n \setminus i}))$ *without* re-training. In practice, approximating 8 with a $K$-th order expansion often suffices, which permits a gradient estimator $\hat{\nabla}^{\mathrm{IF}}\mathcal{L}(\pi)$ that avoids any re-computation of the mean-square-error,

$$\hat{\nabla}^{\mathrm{IF}}\mathcal{L}(\pi) := \sum_{i=1}^{n} \frac{\partial \log \pi(Z_i|X_i)}{\partial \pi} \mathcal{I}_{\mathrm{MSE}}(S_i), \quad \text{where,} \quad \mathcal{I}_{\mathrm{MSE}}(S_i) := -\sum_{k=1}^{K} \frac{\epsilon^k}{k!} \mathcal{I}_{\mathrm{MSE}}^{(k)}(S_i). \tag{9}$$

In fact, as we illustrate in Figure 3 and state formally in Section 4, when using finite $K$ terms, the bias of the gradient in 9 is of the order $n^{-K}$, while still providing significant variance reduction. This enables the bias to be neglected even when using $K = 1$. Intuitively, for $k = 1$, $\mathcal{I}_{\mathrm{MSE}}^{(k)}$ operationalizes the concept of the first-order derivative for the functional $\mathrm{MSE}(\theta(\cdot))$, i.e., it characterizes the effect on MSE when the training sample $S_i$ is infinitesimaly up-weighted during optimization,

$$\mathcal{I}_{\mathrm{MSE}}^{(1)}(S_i) := \frac{\mathrm{d}}{\mathrm{d}t}\mathrm{MSE}\Big(\theta\big((1-t)\mathbb{D} + t\delta(S_i)\big)\Big)\Big|_{t=0}. \tag{10}$$

A key contribution is to derive (Theorem 2 in Appendix D) the following form for $\mathcal{I}_{\mathrm{MSE}}^{(1)}(S_i)$,

$$\mathcal{I}_{\mathrm{MSE}}^{(1)}(S_i) = \frac{\mathrm{d}}{\mathrm{d}\theta}\mathrm{MSE}(\hat{\theta})\,\mathcal{I}_{\theta}^{(1)}(S_i), \tag{11}$$

$$\text{where,} \quad \mathcal{I}_{\theta}^{(1)}(S_i) = -\frac{\partial M_n}{\partial \theta}^{-1}(\hat{\theta}, \hat{\phi})\left[m(S_i, \hat{\theta}, \hat{\phi}) - \frac{\partial M_n}{\partial \phi}(\hat{\theta}, \hat{\phi})\left[\frac{\partial Q_n}{\partial \phi}^{-1}(\hat{\phi})q(S_i, \hat{\phi})\right]\right]. \tag{12}$$

Note that these expressions generalize the influence function for standard machine learning estimators (e.g., least-squares, classification) (Koh & Liang, 2017) to more general estimators (e.g., IV estimator, GMM estimators, double machine-learning estimators) that involve black-box estimation of nuisance parameters $\phi$ alongside the parameters of interest $\theta$, as discussed in 3.

In Appendix H we show how $\mathcal{I}_{\theta}^{(1)}(S_i)$ can be computed efficiently, without ever explicitly storing or inverting derivatives of $M_n$ or $Q_n$, using Hessian-vector products (Pearlmutter, 1994) readily available in auto-diff packages (Bradbury et al., 2018; Paszke et al., 2019). Our approaches are inspired by techniques used to scale implicit gradients and influence function computation to models with millions (Lorraine et al., 2020) of parameters. In Appendix H, we also discuss how $\mathcal{I}_{\theta}^{(1)}(S_i)$ can be further re-used to *partially* estimate $\mathcal{I}_{\mathrm{MSE}}^{(2)}(S_i)$, thereby making $\mathcal{I}_{\mathrm{MSE}}(S_i)$ more accurate.

## 3.2 MULTI-REJECTION SAMPLING FOR DISTRIBUTION CORRECTION

Recall from Challenge 2 that gradient based search of $\pi^*$ would require evaluating gradient at $\pi'$ using data collected from $\pi \neq \pi'$. A common technique in RL to address this is by using importance

Figure 4: **(Left)** Change in $k$ did not impact the $\pi$ optima as the orderings were preserved. **(Right)** variance of rejection sampling (RS) vs importance sampling (IS). See Figure 8 for the plot with log-scale. **Observe the scale on the y-axes.**

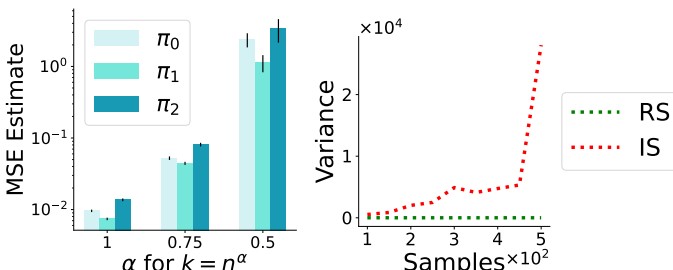

sampling (IS) (Precup, 2000; Thomas, 2015). Unfortunately, the effective importance ratio, when using IS in our setting, will be a product of the importance ratios for all the $n$ data points (see Equation 165 in Appendix F). This can result in the variance of gradient estimates being *exponential* in the number of samples $n$, in the worst case. We illustrate this in Figure 4 and discuss formally in Appendix F.

We now introduce a multi-rejection importance sampling approach to estimate the gradient under a decision policy different from that used to gather the historical data. In contrast, unlike in RL, in our setting, the estimator $\theta(D_n)$ can be considered symmetric for most of the popular estimators as the order of samples in $D_n$ does not matter. Therefore, instead of performing importance sampling over the whole prior data, we propose creating a new dataset of size $k$ by employing *rejection sampling*. Critically, the importance ratio used to select (or reject) each sample $S_i$ to evaluate a new policy $\pi'$ depends only on the importance ratio $\rho(s)$ for that **single** sample $s$, instead of the product of those ratios over the entire dataset:

$$S_i = \begin{cases} \text{Accept} & \text{if} \quad \xi_i \leq \rho(S_i)/\rho_{\max} \\ \text{Reject} & \text{Otherwise.} \end{cases} \qquad \rho(S_i) := \frac{\Pr(S_i; \pi')}{\Pr(S_i; \pi_i)} = \frac{\pi'(Z_i|X_i)}{\pi_i(Z_i|X_i)}, \qquad (13)$$

where $\xi_i \sim \mathtt{Uniform}(0,1)$ and $\rho_{\max} := \max_s \rho(s)$. Note that 13 allows simulating samples from the data distribution governed by $\pi'$, without requiring any knowledge of the unobserved confounders $U_i$. As we will prove in Section 4, this ensures that the variance of the resulting dataset scales with the worst *single* sample importance ratio, instead of *exponentially* over the entire dataset size $n$. This process can therefore be used to generate a smaller $k$ sized dataset with the same distribution of data as sampling from the desired policy $\pi'$.

As we illustrate in Figure 4 and discuss formally in Appendix F.4, it is reasonable to consider that instrument-selection policy *ordering* (in terms of their MSE) is robust to the dataset size, i.e., $\mathbb{E}_\pi[\mathrm{MSE}(\theta(D_n))] \geq \mathbb{E}_{\pi'}[\mathrm{MSE}(\theta(D_n))]$ implies that $\mathbb{E}_\pi[\mathrm{MSE}(\theta(D_k))] \geq \mathbb{E}_{\pi'}[\mathrm{MSE}(\theta(D_k))]$, where $k < n$. In addition, as we will state formally in Property 4, if $k = o(n)$, then the probability of the failure case, where the expected number of samples $n^{RS}$ accepted by rejection sampling is less than the desired $k$ samples, *decays exponentially*.

This insight is advantageous as we can now directly obtain datasets $D_k$ from $\pi'$, albeit of size $k < n$, to evaluate $\pi'$, as desired. We implement this by first running the rejection process for all samples, yielding a set of samples of size $n^{RS}$ that are drawn from $\pi$, and then selecting a random subset of size $k$ from the $n^{RS}$ accepted samples. For the sub-sample $D_k$, analogous to the true MSE in 4 we calculate an estimate of the MSE using $X_i$ and $A_i$ drawn from a held-out dataset with $M$ samples from $\mathbb{P}_{x,a}$ as

$$\widehat{\mathrm{MSE}}(\theta(D_k)) = \frac{1}{M} \sum_{i=1}^{M} (f(X_i, A_i; \theta(D_n)) - f(X_i, A_i; \theta(D_k)))^2. \qquad (14)$$

Note that in general, our historical data may be a sequence of decision policies, as we adaptively deploy new instrument-selection decision policies. In this setting, a straightforward application of rejection sampling will require the support assumption which necessitates $\pi_i(Z_i|X_i) > 0$ if $\pi'(Z_i|X_i) > 0$ for all $i \in \{1, ..., n\}$. This may restrict the class of new instrument-selection policies that can be deployed. Instead we propose the following *multi*-rejection sampling strategy:

$$S_i = \begin{cases} \text{Accept} & \text{if} \quad \xi_i \leq \bar{\rho}(S_i)/\bar{\rho}_{\max} \\ \text{Reject} & \text{Otherwise.} \end{cases} \qquad \bar{\rho}(S_i) := \frac{\pi'(Z_i|X_i)}{\frac{1}{n} \sum_j \pi_j(Z_i|X_i)} \qquad (15)$$

where $\bar{\rho}_{\max} = \max_s \bar{\rho}(s)$. As we show in Appendix G, this relaxes the support assumption enforced for every $\pi_i$ to support assumption over the *union* of supports of $\{\pi_i\}_{i=1}^n$, while still ensuring that the accepted samples follow the distribution specified by $\pi'$. For e.g., if the initial data collected or available used a uniform instrument-selection decision policy, then subsequent policies can be *arbitrarily* deterministic and the multi-rejection procedure can still use all the data and remain valid. This makes it particularly well suited for our adaptive experiment design framework. In Appendix G, we further prove that the expected number of samples $n^{\mathrm{MRS}}$ accepted under multi-rejection will *always* be more than or equal to $n^{RS}$ obtained using rejection sampling, *irrespective* of $\{\pi_i\}_{i=1}^n$.

### 3.3 DIA: DESIGNING INSTRUMENTS ADAPTIVELY

Using influence functions and multi-rejection sampling we addressed both the challenges about high variance. Leveraging the flexibility offered by the proposed gradient-based procedure, we can now readily extend the idea for sequential collection of data. To do so, we propose using a parameterization of $\pi$ to account for data already collected, which is important during adaptive instrument-selection policy design.

Let $n$ be the size of the dataset $D_n = \{S_i\}_{i=1}^n$ collected so far, and let $N$ be the budget for the total number of samples that can be collected. At each stage of data collection, DIA searches for a policy $\pi_\varphi$, parameterized using $\varphi$, such that when the remaining $D_{N-n} = \{S_i\}_{i=n+1}^N$ samples are collected using $\pi_\varphi$, the *combined* distribution $D_N := D_n \cup D_{N-n}$ would approximate the optimal distribution for estimating $\theta$. Let $\pi_\varphi^{\mathrm{eff}}$ operationalize $D_N$, if $D_{N-n}$ were to be collected using $\pi_\varphi$,

$$\pi_\varphi^{\mathrm{eff}}(z|x) := \frac{n}{N}\left(\frac{1}{n}\sum_{i=1}^n \pi_i(z|x)\right) + \left(1 - \frac{n}{N}\right)\pi_\varphi(z|x). \tag{16}$$

Let $\mathcal{I}_{\widehat{\mathrm{MSE}}}(S_i)$ be the influence function similar to 9, where the MSE is replaced with its proxy 14, and let $\{S_i\}_{i=1}^k$ in the following be the samples for $\pi_\varphi^{\mathrm{eff}}$ obtained using multi-rejection sampling in 15. Then for the first batch, $\pi_\varphi$ is designed to be a uniform distribution, and for the later batches we leverage 9 and 14 to update $\pi_\varphi$ using

$$\hat{\nabla}^{\mathrm{IF}}\mathcal{L}(\pi_\varphi^{\mathrm{eff}}) := \sum_{i=1}^k \frac{\partial \log \pi_\varphi^{\mathrm{eff}}(Z_i|X_i)}{\partial \varphi}\mathcal{I}_{\widehat{\mathrm{MSE}}}(S_i). \tag{17}$$

We use $\pi_\varphi$ to collect the next batch of data and then update $\pi_\varphi$ again, as discussed above, to ensure that the final distribution $D_N = \{S_i\}_{i=1}^N$ approximates the optimal distribution, for estimating $\theta$, as closely as possible. Intuitively, this technique is similar to receding horizon control, where the controller is planned for the entire remaining horizon of length $N - n$, but is updated and refined again periodically. A pseudo-code for the proposed algorithm is provided in Appendix H.

## 4 THEORY

We now provide theoretical statements about the impact of the algorithmic design choices, over alternatives, in terms of their benefit on the variance. Proofs and more formal detailed statements of the theorems are deferred to Appendix E. We first state that variance of the naive gradient can scale poorly with the dataset size:

**Property 1** (Informal). *If* $\mathrm{MSE}(\theta(D_n))$ *is leave-one-out stable and* $n\|\mathrm{MSE}(\theta(D_n))\|_{L^2}^2 = \Omega(\alpha(n)) = \omega(1)$, *we have* $\mathrm{Var}(\hat{\nabla}\mathcal{L}(\pi)) = \Omega(\alpha(n))$ *(c.f. Theorem 3 in Appendix).*

In many cases, we would expect that the mean squared error decays at the rate of $n^{-1/2}$ (unless we can invoke a fast parametric rate of $1/n$, which holds under strong convexity assumptions); in such cases, we have that $\alpha(n) = \omega(1)$ and $\mathrm{Var}(\hat{\nabla}\mathcal{L}(\pi))$ can grow with the sample size. Moreover, if the mean squared error does not converge to zero, due to persistent approximation error, or local optima in the optimization, or if 2 is ill-posed when dealing with non-parametric estimators, then the variance can grow linearly, i.e. $\alpha(n) = \Theta(n)$. See Figure 3. In contrast, the variance of the gradient of the loss with MSE covariates 7 is often independent of the data set size:

**Property 2** (Informal). $\mathbb{E}[\hat{\nabla}^{CV}\mathcal{L}(\pi)] = \nabla\mathcal{L}(\pi)$. *Further, if* $\mathrm{MSE}(\theta(D_n))$ *is leave-one-out stable then* $\mathrm{Var}(\hat{\nabla}^{CV}\mathcal{L}(\pi)) = \mathcal{O}(1)$ *(c.f. Theorem 4 in Appendix).*

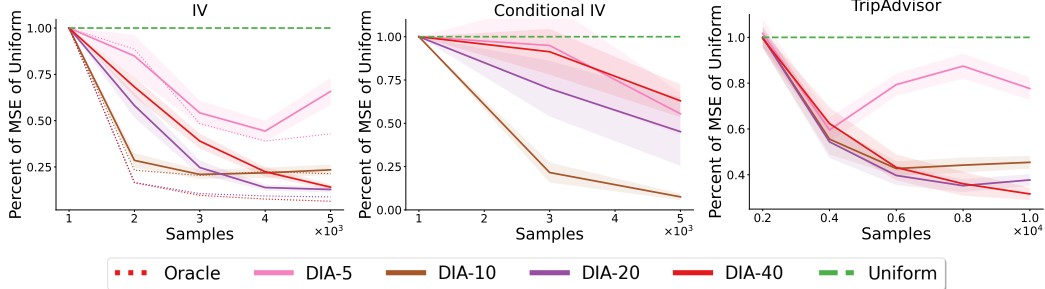

Figure 5: Performance gains when considering a different number of instrumental variables, with a total of 5 re-allocation. First allocation (1000 samples) is done using uniform sampling, and thus all the methods have the same performance initially. DIA-X refers to the application of the proposed algorithm to the domain setting where the number of instruments is X. Note that since MSE under uniform distribution also keeps increasing, a saturating or increasing trend of the *ratio* does not imply that the MSE is increasing. **(Left)** Results for the IV setting (using closed-form linear estimator). Dotted lines correspond to the oracle estimate for the respective number of instruments. **(Middle)** Results for the conditional IV setting (using logistic regression based estimator). **(Right)** Results for the TripAdvisor domain (Syrgkanis et al., 2019) (using neural network based estimator). Error bars correspond to one stderr.

Unlike Property 1, Property 2 holds irrespective of $\alpha(n)$, thus providing a reliable gradient estimator even if the function approximator is mis-specified, or optimization is susceptible to local optima, or if 2 is ill-posed when dealing with non-parametric estimators. Further, our loss using influence functions 9 can also yield similar variance reduction, at much lower computational cost:

**Property 3** (Informal). *If the Von-Mises expansion in 8 exists, $\mathbb{E}[\hat{\nabla}^{IF}\mathcal{L}(\pi)] = \nabla\mathcal{L}(\pi) + \mathcal{O}(n^{-K})$. If $\mathrm{MSE}(\theta(D_n))$ is leave-one-out stable then $\mathrm{Var}(\hat{\nabla}^{IF}\mathcal{L}(\pi)) = \mathcal{O}(1)$ (c.f. Theorem 5 in Appendix).*

We then prove how rejection sampling can impact the MSE, enabling a variance that has an exponential reduction in variance compared to importance sampling, whose variance can depend on $\rho_{\max}^k$ (c.f. Theorem 7 in Appendix), at the expense of only a bounded failure rate of not producing the desired number of samples $k$. For this result, let $\mathrm{MSE}_k^{RS}$ be the estimate of MSE computed using the mean of $\widehat{\mathrm{MSE}}(\theta(D_k))$ across all the subsets $D_k$ from the $n^{RS}$ accepted samples.

**Property 4** (Informal). *Let $\mathbb{E}_\pi[\mathrm{MSE}(\theta(D_k))^2] = \alpha(k)^2$ then $\mathrm{Var}(\mathrm{MSE}_k^{RS}) = \mathcal{O}(\alpha(k)^2 k\rho_{\max}/n + \alpha(n))$ and $Bias(\mathrm{MSE}_k^{RS}) = \mathcal{O}(e^{-n/\rho_{\max}} + \sqrt{\alpha(n)})$. Further, the failure probability is bounded as $\Pr(n^{RS} < k) \leq e^{-n/(8\rho_{\max})}$ (c.f. Theorem 8 in Appendix).*

We also show in the Appendix (c.f. Corollary 1) that the above theorem, together with further regularity conditions, can be used to argue that ranking policies based on $\mathrm{MSE}_k^{RS}$ leads to approximately optimal policy decisions, with respect to $\mathbb{E}[\mathrm{MSE}(\theta(D_n))]$, in a strong approximation sense.

## 5 EXPERIMENTS

In this section we empirically investigate the flexibility and efficiency of our approach. We provide the key takeaway points here, and more experimental details are deferred to Appendix I.

**A. Flexibility of the proposed approach:** One of the key benefits of DIA is that it can be used as-is for a variety of estimators (linear and non-linear) and settings (unconditional and conditional IV setting). Drawing inspiration from real-world use cases, consider binary treatment settings, where we only have access to a plethora of instruments that can be used to encourage (e.g., using different types of notifications, emails, etc.) treatment uptake. Specifically, we consider three regimes:

**IV:** This simulator is a basic IV setting with homogenous treatment effects and which thus permits a *closed-form* solution using the standard two-stage least-squares procedure (Pearl, 2009). This domain has heteroskedastic outcome noise and varying levels of compliance for different instruments. Figure 5 (left) presents the results for this setting.

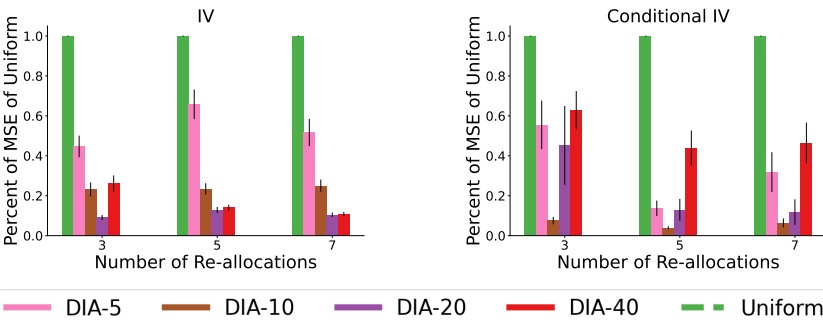

Figure 6: DIA-X refers to the application of the proposed algorithm to the domain setting where the number of instruments is X. The plots illustrate the percent of MSE of uniform, after all the 5000 samples are collected, with the given number of re-allocations. Error bars correspond to one stderr.

**Conditional IV:** Second we consider the important setting where instruments/encouragements can be allocated in a context-dependent way, and the objective is to estimate the conditional average treatment effect. Our simulator has covariates containing a mix of binary and continuous-valued features, and heteroskedastic noises and compliances, across instruments and outcomes, for every covariate. We solve the counterfactual prediction $f$ in 2 using the two-stage moment conditions in 3 involving logistic regression and this estimator does *not* have a closed-form solution. Figure 5 (middle) presents the results for this setting.

**TripAdvisor:** We use the simulator built from TripAdvisor customer data (Syrgkanis et al., 2019). The goal is to estimate the effect on revenue if a customer becomes a subscribed member. The original setup had instruments corresponding to easier/promotional sign-up options, but to provide ablation studies we consider a larger, synthetically augmented, set of instruments. Covariates correspond to demographic data about the customer, and their past interaction behavior on the website. For this setting, we use the estimator by Hartford et al. (2017) and model all the functions $f$, $\mathrm{d}P$ (from 2), and $\pi$, using neural networks. Figure 5 (right) presents the results for this setting.

**B. Understanding the performance gains:** In order to simulate a realistic situation where deployed policy can be updated only periodically, we consider a batched allocation setting. To measure the performance gain, we compute the relative improvement in MSE over uniform allocation strategy, i.e., a metric popular in *direct* experiment design literature (Che & Namkoong, 2023) (lower is better). In the IV setting, in Figure 5 (left), we also provide a comparison with the oracle obtained using brute force search. To assess the robustness of our method, we also consider varying the number of available instruments and the number of batch re-allocations possible.

Across all the domains, we observe that DIA can provide substantial gains for estimating the counterfactual prediction $f$ by adaptively designing the instruments for indirect experiments (Figure 5). Performance gains are observed even as we vary the number of instruemnts present in the domain. Importantly, DIA achieves this across different (linear and non-linear) estimators, thereby illustrating its flexibility. In addition, in general there is a benefit to increasing the number of times the instrument-design policy is updated (Figure 6).

It is worth highlighting that DIA can improve the accuracy of the base estimator by a factor of $1.2\times$ to $10\times$: equivalently, for some desired target treatment effect estimation accuracies, DIA needs only 80% to an order-of-magnitude less sample data.

Our results also suggest a tension behind the amount of data and the complexity of the problem, creating a 'U'-trend (see e.g. Figure 6). With a larger number of instruments, there is more *potential* advantage of being strategic, especially when for many contexts, most instruments have a weak influence. However, learning the optimal strategy to realize those gains also becomes harder when there are so many instruments (and the sample budget is held fixed). In the middle regime gains are substantial and the method can also quickly learn an effective instrument-selection design $\pi^*$.

## 6 CONCLUSION

The increasing prevalence of human-AI interaction systems presents an important development. However, as AI systems can often only be *suggestive* and not *prescriptive*, estimating the effect of its suggested action necessitates the development of sample-efficient methods to estimate treatment effect merely through suggestions, while leaving the agency of decision to the user. This work took the initial step to lay the foundation for adaptive data collection for such indirect experiments. We characterized the key challenges, theoretically assessed the proposed remedies, and validated them empirically on domains inspired by real-world settings. Scaling the framework to settings with natural language instruments, and doing inference with adaptively collected data (Zhang et al., 2021; Gupta et al., 2021) remain exciting future directions.

## 7 ACKNOWLEDGEMENT

We thank Stefan Wager, Jann Spiess, and Art Owen for their valuable feedback. Earlier versions of the draft also benefited from feedback from Jonathan Lee and Allen Nie. This work was supported in part by 2023 Amazon Research Award.

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

# Adaptive Instrument Design for Indirect Experiments
# (Appendix)

CONTENTS

## A  EXTENDED DISCUSSION ON RELATED WORK

Experiment design has a rich literature and no effort is enough to provide a detailed review. We refer readers to the work by Chaloner & Verdinelli (1995); Rainforth et al. (2023) for a survey. Researchers have considered adaptive estimation of (conditional) average treatment effect (Hahn et al., 2011; Kato et al., 2020), leveraging Gaussian processes (Song et al., 2023), active estimation of the sub-population benefiting the most from the treatment (Curth et al., 2022), for estimating the treatment effects when samples are collected irregularly (Vanderschueren et al., 2023), and adaptive data collection to find the best arm while also maximizing welfare (Kasy & Sautmann, 2021). Che & Namkoong (2023) consider adaptive experiment design using differentiable programming. Our work shares their philosophy in terms of balancing sample complexity with computational complexity. Li & Owen (2023) consider using methods from double ML for adaptive direct experiments. In contrast, we consider adaptive indirect experiments that often make use of two-stage and double ML estimators. However, all these works only consider design for direct experiments.

For instrumental variables, researchers have developed methods that can automatically decide how to efficiently combine different instruments to enhance the efficiency of the estimator (Kuang et al., 2020; Yuan et al., 2022). In contrast, our work considers how to collect more data online. Further, while considered IV estimation under specific assumptions, we refer readers to the work by Angrist et al. (1996); Clarke & Windmeijer (2012); Wang et al. (2021) for discussions on alternate assumptions. Particularly, we considered conditional LATE to be equal to CATE. For more discussion on the relation of CATE and conditional LATE, we refer readers to the work by Heckman & Urzua (2010); Imbens (2010); Athey & Wager (2021).

Some of the tools that we developed are also related to a literature outside of causal estimation and experiment design. Our treatment of the dataset selection policy as a factorized distribution and development of the leave-one-out sample is related to ideas in action-factorized baselines in reinforcement learning (Wu et al., 2018; Liu et al., 2017) and leave-one-out control variates (Richter et al., 2020; Salimans & Knowles, 2014; Shi et al., 2022; Kool et al., 2019; Titsias & Shi, 2022). In contrast to these, our work is for experiment design, and to be compute efficient in our setup we had to additionally develop black-box influence functions for two-stage estimators.

Our work also complements prior work on influence function for conditional IVs (Kasy, 2009; Ichimura & Newey, 2022) as they were not directly applicable to deep neural network-based estimators. Further, our characterization of the leave-one-out estimate of the MSE requires existence of all the higher-order influence functions. Similar conditions have been used by prior works in the context of uncertainty quantification (Alaa & Van Der Schaar, 2020), model selection (Alaa & Van Der Schaar, 2019; Hines et al., 2022; Debruyne et al., 2008), and dataset distillation (Loo et al., 2023).

Tang (2022) quantifies and develops first-order gradient-based control-variates to perform bias-variance trade-off of the naive REINFORCE gradient in the meta-reinforcement learning setting. However, in their analysis, the effective 'reward' function admits a simple additive decomposition, where in our setting it corresponds to $\mathrm{MSE}(\theta(D_n))$ which cannot be analysed using their technique. Further, potential non-linearity of $\theta$ makes the analysis a lot more involved in our case. Similarly, to deal with off-policy correction in RL, researchers have also considered using multi-importance-sampling (Papini et al., 2019; Liotet et al., 2022). However, using multi-importance sampling (as opposed to multi-rejection sampling) in our setup can still result in exponential variance. Our multi-rejection sampling is designed by leveraging the symmetric structure of our estimator.

Further, our $\widehat{\mathrm{MSE}}$ estimator is related to MSE estimation using sub-sampling bootstrap (Politis et al., 1999; Geyer, 2006), where the mean-squared-error is computed by comparing the estimate from the entire dataset of size $n$ with the estimator obtained using the subset of the dataset of size $k < n$, and then rescaling this appropriately using the convergence rate of the estimator (Hall, 1990; Cao, 1993). In our setting, we can avoid using the rescaling factor (which requires knowledge of convergence rates) since we only care about computing the gradient of the $\widehat{\mathrm{MSE}}$. Additionally, we had to address the distribution shift issue for our setting. For more background on statistical bootstrap, we refer readers to some helpful lectures notes (Chen, 2017a; Mikusheva, 2013; Shi, 2012; Hall, 2016). See (Chen, 2017b, 2020) for connections between bootstrap and influence functions, and (Takahashi, 1988) for connections between Von-Mises expansion (used for LOO estimation using influence functions) and Edgeworth expansions (used for bootstrap theory).

Finally, while we leverage tools from RL, our problem setup presents several unique challenges that make the application of standard RL methods as-is ineffective. Off-policy learning is an important requirement for our method, and policy-gradient and Q-learning are the two main pillars of RL. As we discussed in Section 4, conventional importance-sampling-based policy gradient methods can have variance exponential in the number of samples (in the worst-case), rendering them practically useless for our setting. On the other hand, it is unclear how to efficiently formulate the optimization problem using Q-learning. From an RL point of view, the 'reward' corresponds to the MSE, and the 'actions' corresponds to instruments. Since this reward depends on all the samples, the effective 'state space' for Q-learning style methods would depend on the combinatorial set of the covariates, i.e., let $\mathcal{X}$ be the set of covariates and $n$ be the number of samples, then the state $s \in \mathcal{X}^n$. This causes additional trouble as $n$ continuously changes/increases in our setting as we collect more data.

## B EXAMPLES OF PRACTICAL USE-CASES

Due to space constraints, we presented the motivation concisely in the main paper. To provide more discussion on some real-world use cases, below we have mentioned three use cases across different fields of application:

- **Education:** It is important for designing an education curriculum to estimate the effect of home-work assignments, extra reading material, etc. on a student's final performance (Eren & Henderson, 2008). However, as students cannot be forced to (not) do these, conducting an RCT becomes infeasible. Nonetheless, students can be encouraged via a multitude of different methods to participate in these exercises. These different forms of encouragement provide various instruments and choosing them strategically can enable sample efficient estimation of the desired treatment effect.

- **Healthcare:** Similarly, for mobile gym applications, or remote patient monitoring settings (Ferstad et al., 2022), users retain the agency to decide whether they would like to follow the suggested at-home exercise or medical intervention, respectively. As we cannot perform random user assignment to a control/treatment group, RCTs cannot be performed. However, there is a variety of different messages and reminders (in terms of how to phrase the message, when to notify the patient, etc.) that serve as instruments. Being strategic about the choice of the instrument can enable sample efficient estimation of treatment effect.

- **Digital marketing:** Many online digital platforms aim at estimating the impact of premier membership on both the company's revenue and also on customer satisfaction. However, it is infeasible to randomly make users a member or not, thereby making RCTs inapplicable. Nonetheless, various forms of promotional offers, easier sign-up options, etc. serve as instruments to encourage customers to become members. Strategically personalizing these instruments for customers can enable sample-efficient treatment effect estimation. In our work, we consider one such case study using the publicly available TripAdvisor domain (Syrgkanis et al., 2019).

## C LINEAR CONDITIONAL INSTRUMENTAL VARIABLES

In this section, we aim to elucidate how different factors in (a) the data-generating process (e.g., heteroskedasticity in compliance and outcome noise, structure in the covariates) and (b) the choice of estimators can affect the optimal data collection strategy.

Since this discussion is aimed at a more qualitative (instead of quantitative results like those in Section 5) we will consider a (partially) linear CATE IV model,

$$Y = \theta_0' X A + f_0(X) + \epsilon, \tag{18}$$

where $\mathbb{E}[\epsilon|X, A] \neq 0$ but $\mathbb{E}[\epsilon|X, Z] = 0$. As $\mathbb{E}[XZ\epsilon] = \mathbb{E}[XZ\mathbb{E}[\epsilon|XZ]] = 0$, the estimate $\hat{\theta}$ is constructed based on the solution to the moment vector:

$$\mathbb{E}[(Y - \theta' X A - \hat{f}(X)) X Z] = 0, \tag{19}$$

where $Z$ is a scalar, $A \in \{0, 1\}$ and $\mathbb{E}[Z \mid X] = 0$ (i.e. we know the propensity of the instrument and we can always exactly center the instrument). Moreover, $\hat{f}(X)$ is an arbitrary function that converges in probability to some function $f_0$.

An example: $Z = w(X)'V$ and $\mathbb{E}[V \mid X] = 0$. This can simulate a situation where $V$ represents the one-hot-encoding of a collection of binary instruments, $w(X)$ selects a linear combination of these

instruments to observe, yielding the observed instrument $Z$ (e.g. $w(X) \in \{e_1, \ldots, e_m\}$ encodes that we can only select one base instrument to observe). Moreover, each of these base instruments comes from a known propensity and we can always exactly center it conditional on $X$.

The estimate $\hat{\theta}$ is the solution to:

$$\frac{1}{n} \sum_i (Y_i - \theta' X_i A_i - \hat{f}(X_i)) X_i Z_i = 0. \tag{20}$$

We care about the MSE of the estimate $\hat{\theta}$:

$$\text{MSE}(\hat{\theta}) = \mathbb{E}_X[(\hat{\theta}'X - \theta_0'X)^2]. \tag{21}$$

**Lemma 1.** *Let $V = \mathbb{E}[XX']$ and $J = \mathbb{E}[AZXX']$ and $\Sigma = \mathbb{E}[\epsilon^2 Z^2 XX']$, with $\epsilon = Y - \theta_0'XA - f_0(X)$. Finally, let*

$$U = V^{1/2} J^{-1} \Sigma J^{-1} V^{1/2}. \tag{22}$$

*The mean and the variance of the MSE converge to:*

$$n\mathbb{E}[\text{MSE}(\hat{\theta})] \to \|U\|_{nuclear} \qquad n^2 \text{Var}[\text{MSE}(\hat{\theta})] \to 2\|U\|_{fro}^2. \tag{23}$$

*Proof.* The proof is structured as the following,

- **Part A:** We first recall the asymptotic properties of GMM estimators.

- **Part B:** We then use this result to estimate the asymptotic property of $\sqrt{n}(\hat{\theta} - \theta)$, and also of $n\text{MSE}(\hat{\theta})$.

- **Part C:** Finally, we use singular value decomposition to express the mean and variance for $\text{MSE}(\hat{\theta})$ in simplified terms.

**Part A:** Let $S_i = \{X_i, Z_i, A_i, Y_i\}$. Recall that for a GMM-estimator, $\hat{\theta}$ is the solution to

$$\frac{1}{n} \sum_i m(\theta, S_i) = 0, \tag{24}$$

where $m(\theta, S_i)$ is the moment for sample $S_i$. For large enough $n$ we can represent the difference between the estimates $\hat{\theta}$ and the true $\theta$ by use of a Taylor expansion (Kahn, 2015),

$$\hat{\theta} = \theta + \frac{1}{n} \sum_i \text{IF}(\theta, S_i) + \text{higher order terms}, \tag{25}$$

where $\text{IF}(\theta, S_i)$ is the influence of the sample $i$, and is given by $\mathbb{E}[\nabla m(\theta, S)]^{-1} m(\theta, S_i)$. Therefore,

$$\sqrt{n}(\hat{\theta} - \theta) = \frac{1}{\sqrt{n}} \sum_i \text{IF}(\theta, S_i) + o_p(1). \tag{26}$$

Moreover, $\mathbb{E}[\text{IF}(\theta, S)] = 0$ and $\text{Cov}(\text{IF}(\theta, S)) = \mathbb{E}[\text{IF}(\theta, S)\text{IF}(\theta, S)']$. See (Newey & McFadden, 1994) for a more formal argument.

**Part B:** Therefore, the parameters estimated by the empirical analogue of the vector of moment equations in 20 are asymptotically linear:

$$\sqrt{n}(\hat{\theta} - \theta_0) = \frac{1}{\sqrt{n}} \sum_{i=1}^n \mathbb{E}[AZXX']^{-1} X_i Z_i (Y_i - \theta_0'X_i A_i - f_0(X_i)) + o_p(1). \tag{27}$$

For simplicity, let's take $f_0(X) = 0$. We can always redefine $Y \to Y - f_0(X)$. Moreover, let $v(X) = \mathbb{E}[AZ \mid X] = \text{Cov}(A, Z \mid X)$ and $J = \mathbb{E}[v(X)XX']$. Let $\epsilon_i = Y_i - \theta_0'X_i A_i$. Then we have:

$$\sqrt{n}(\hat{\theta} - \theta_0) = \frac{1}{\sqrt{n}} \sum_{i=1}^n J^{-1} X_i Z_i \epsilon_i + o_p(1). \tag{28}$$

We care about the average prediction error:

$$\mathbb{E}[(\theta'X - \theta_0'X)^2] = (\theta - \theta_0)'\mathbb{E}[XX'](\theta - \theta_0). \tag{29}$$

Let $V = \mathbb{E}[XX']$. Then,

$$\mathbb{E}[(\theta'X - \theta_0'X)^2] = \|V^{1/2}(\theta - \theta_0)\|^2. \tag{30}$$

Invoking the asymptotic linearity of $\sqrt{n}(\hat{\theta} - \theta_0)$:

$$n\mathbb{E}[(\theta'X - \theta_0'X)^2] = \left\|\frac{1}{\sqrt{n}}\sum_i \epsilon_i Z_i V^{1/2} J^{-1} X_i\right\|^2 + o_p(1). \tag{31}$$

Moreover, we have:

$$\frac{1}{\sqrt{n}}\sum_i \epsilon_i Z_i V^{1/2} J^{-1} X_i \Rightarrow_d N\left(0, V^{1/2} J^{-1}\mathbb{E}[\epsilon^2 Z^2 XX']J^{-1}V^{1/2}\right). \tag{32}$$

Let $\Sigma = \mathbb{E}[\epsilon^2 Z^2 XX']$ and $U = V^{1/2} J^{-1} \Sigma J^{-1} V^{1/2}$. Thus the mean squared error is distributed as the squared of the $\ell_2$ norm of a multivariate Gaussian vector $N(0, U)$.

**Part C:** If we let $U = C\Lambda C'$, be the SVD decomposition, then note that if $v \sim N(0, U)$ then since $C'C = I$, we have:

$$\|Cv\|^2 = v'C'Cv = v'v = \|v\|^2. \tag{33}$$

Now note that $Cv \sim N(0, C'UC) = N(0, \Lambda)$. Thus $\|Cv\|^2$ is distributed as the weighted sum of $d$ independent chi-squared distributions, i.e. we can write:

$$\|v\|^2 = \sum_{i=1}^d \lambda_j \mathcal{X}_j^2, \tag{34}$$

where $\mathcal{X}_j^2$ are independent $\chi^2(1)$ distributed random variables and $\lambda_j$ are the singular values of the matrix:

$$U = V^{1/2} J^{-1} \Sigma J^{-1} V^{1/2}. \tag{35}$$

Using the mean and variance of the chi-squared distribution, asymptotic mean and variance of the RMSE can be expressed as:

$$n\,\mathbb{E}[\text{MSE}(\hat{\theta})] \to \sum_{j=1}^d \lambda_j \tag{36}$$

$$n^2\text{Var}[\text{MSE}(\hat{\theta})] \to \sum_{j=1}^d \text{Var}(\lambda_j \mathcal{X}_j^2) = \sum_j \lambda_j^2 \text{Var}(\mathcal{X}_j^2) = \sum_j 2\,\lambda_j^2. \tag{37}$$

Thus the expected MSE is the trace norm (or nuclear norm) of $U$ and the variance of the MSE is twice the square of the Forbenius norm. $\qquad\square$

### C.1 Specific Instantiations of Lemma 1

In this subsection, we instantiate Lemma 1 to understand the impact of heteroskedastic compliance, heteroskedastic outcome errors, and structure of covariates on the optimal data collection policy.

Note that $U$ in Lemma 1 takes the form:

$$U = \mathbb{E}[XX']^{1/2}\mathbb{E}[AZXX']^{-1}\mathbb{E}[\epsilon^2 Z^2 XX']\mathbb{E}[AZXX']^{-1}\mathbb{E}[XX']^{1/2}. \tag{38}$$

Noting that by the instrument assumption $\epsilon \perp\!\!\!\perp Z \mid X$, and denoting with

$$\sigma^2(X) = \mathbb{E}[\epsilon^2 \mid X] \qquad \text{(residual variance of the outcome)} \tag{39}$$

$$u^2(X) = \mathbb{E}[Z^2 \mid X] \qquad \text{(variance of the instrument)} \tag{40}$$

$$\gamma(X) = \frac{\mathbb{E}[A\,Z \mid X]}{\mathbb{E}[Z^2 \mid X]} \qquad \text{(heteroskedastic compliance)} \tag{41}$$

where the heteroskedastic compliance corresponds to the coefficient in the regression $A \sim Z$ conditional on $X$. Then we can simplify:

$$U = \mathbb{E}[XX']^{1/2}\mathbb{E}[\gamma(X)u^2(X)XX']^{-1}\mathbb{E}[\sigma^2(X)u^2(X)XX']\mathbb{E}[\gamma(X)u^2(X)XX']^{-1}\mathbb{E}[XX']^{1/2} \tag{42}$$

**Remark 1** (Homoskedastic Compliance, Outcome Error and Instrument Propensity). *If we have a lot of homoskedasticity, i.e.: $\sigma^2(X) = \sigma^2$, $u^2(X) = u^2$ and $\gamma^2(X) = \gamma$ then $U$ simplifies to:*

$$U = \gamma^{-2}u^{-2}\sigma^2 V^{1/2}V^{-1}VV^{-1}V^{1/2} = \gamma^{-2}u^{-2}\sigma^2 I \tag{43}$$

*and we get:*

$$\mathbb{E}[\mathrm{MSE}(\hat{\theta})] = \frac{d}{n}\gamma^{-2}u^{-2}\sigma^2 \tag{44}$$

$$\mathrm{Var}[\mathrm{MSE}(\hat{\theta})] = \frac{2\,d}{n^2}\gamma^{-4}u^{-4}\sigma^4 \tag{45}$$

*Thus we are just looking for instruments that maximize $\gamma\,u$ where $\gamma := \mathbb{E}[AZ]/\mathbb{E}[Z^2]$ is the OLS coefficient of $A \sim Z$ (the first stage coefficient in 2SLS) and $u = \sqrt{\mathrm{Var}(Z)}$ is the standard deviation of the instrument.*

**Remark 2** (Homoskedastic Compliance and Outcome Error). *If we have homoskedasticity in compliance (i.e. $\gamma(X) = \gamma$) and in outcome ($\sigma(X) = \sigma$), then:*

$$U = \sigma^2\gamma^{-2}\mathbb{E}[XX']^{1/2}\mathbb{E}[u^2(X)XX']^{-1}\mathbb{E}[XX']^{1/2} \tag{46}$$

*Or equivalently, we want to maximize the trace of the inverse of $U$:*

$$\mathbb{E}[XX']^{-1/2}\mathbb{E}\left[\gamma^2 u^2(X)XX'\right]\mathbb{E}[XX']^{-1/2} \tag{47}$$

*As $XX'$ is positive semi-definite, the latter is maximized if we simply maximize $\gamma u(X)$ for each $X$. Thus assuming that all the instruments in our choice set have a homogeneous compliance, then we are looking for the instrument policy that maximizes: $\gamma u(X)$, where $\gamma$ is the compliance coefficient and $u(X)$ is the conditional standard deviation of the instrument. If for instance, in our choice set we have only binary instruments and we can fully randomize them, then we should always randomize the instrument equiprobably and we should choose the binary instrument with the largest $\gamma$.*

It is harder to understand how the trace norm or the schatten-2 norm of these more complex matrices behave. Though maybe some matrix tricks could lead to further simplifications of these.

**Remark 3** (Orthonormal Supported $X$). *If $X$ takes values on the orthonormal basis $\{e_1, \ldots, e_j\}$, then by denoting $\sigma_i^2 = \mathbb{E}[\epsilon^2 \mid X = e_i]$, $u_i^2 = \mathbb{E}[Z^2 \mid X = e_i]$ and $\gamma_i = \frac{\mathbb{E}[A\,Z|X=e_i]}{\mathbb{E}[Z^2|X=e_i]}$ and by $\Sigma = diag\{\sigma_1^2, \ldots, \sigma_d^2\}$ and $K = diag\{u_1^2, \ldots, u_d^2\}$ and $\Gamma = diag\{\gamma_1, \ldots, \gamma_d\}$, let $V = \mathbb{E}[XX'] = W\Lambda W'$ be the eigen-representation of $V$, then we have:*

$$\mathbb{E}\left[\sigma^2(X)u^2(X)XX'\right] = W\Lambda K\Sigma W', \tag{48}$$

$$\mathbb{E}\left[\gamma(X)u^2(X)XX'\right] = W\Lambda\Gamma K W'. \tag{49}$$

*Leading to:*

$$U = W\Lambda^{1/2}W'W\Lambda^{-1}\Gamma^{-1}K^{-1}W'W\Lambda K\Sigma W'W\Lambda^{-1}\Gamma^{-1}K^{-1}W'W\Lambda^{1/2}W, \tag{50}$$

*which by orthonormality of the $W$, simplifies to:*

$$U = W\Gamma^{-2}K^{-1}\Sigma W'. \tag{51}$$

*hence we get that:*

$$n\mathbb{E}[MSE(\hat{\theta})] = \sum_i \frac{\sigma_i^2}{\gamma_i^2 u_i^2} = \sum_i \frac{\mathbb{E}[\epsilon^2 \mid X = e_i]\mathbb{E}[Z^2 \mid X = e_i]}{\mathbb{E}[AZ \mid X = e_i]^2}, \tag{52}$$

$$n^2\mathrm{Var}[MSE(\hat{\theta})] = 2\sum_i \frac{\sigma_i^4}{\gamma_i^4 u_i^4}, \tag{53}$$

| Estimator | Compliance X=-1 | X = 1 | Instrument Var. $E[Z^2|X]$ $X = -1$ | $X = 1$ | Optimal? |
|---|---|---|---|---|---|
| $\mathbb{E}_n\big[(Y - \theta^\top X\,A)X\,Z\big] = 0$ | 1 | 1/3 | 1/4 1/4 | 0 1/4 | Yes No |
| | 1 | 1/2 | 1/4 1/4 | 0 1/4 | No Yes |
| $\mathbb{E}_n\big[\frac{\gamma(X)}{\sigma^2(X)}(Y - \theta^\top X\,A)X\,Z\big] = 0$ | 1 | 1/3 | 1/4 1/4 | 0 1/4 | No Yes |
| | 1 | 1/2 | 1/4 1/4 | 0 1/4 | No Yes |

Table 1: In general the optimal instrument-selection policy will depend not only on compliance per covariates but also on the estimation method. We demonstrate this here for a simple linear CATE estimation where $g(X, A) := \theta_0^\top X A$, instrument $Z$ is a scalar, $A \in \{0, 1\}$, $X \in \{-1, 1\}$, $\mathbb{E}[Z \mid X] = 0$, and outcome error is homoskedastic. For the first estimator, the optimal instrument-selection policy for $X = 1$ changes depending on the compliance for $X = 1$, where we characterize optimality in terms of the asymptotic mse. In contrast, for the second estimator, randomizing the instrument is best for both compliance values of $X = 1$.

*and the problem decomposes into optimizing separately for each $i$, maximize the terms:*

$$\gamma_i \sqrt{u_i^2} = \frac{\mathbb{E}[AZ \mid X = e_i]}{\mathbb{E}[Z^2 \mid X = e_i]}\sqrt{\mathbb{E}[Z^2 \mid X = e_i]} \tag{54}$$

*assuming that our constraints on the policy for choosing $Z$ are decoupled across the values of $X$. So similarly, we just want to maximize the compliance coefficient, multiplied by the standard deviation of the instrument, conditional on each $X$.*

Beyond this case, when $X$ doesn't only take values on the orthonormal basis, then the problem of optimizing the choice of an instrument distribution so as to minimize the expected MSE doesn't seem to decouple across the values of $X$.

**Remark 4** (Scalar $X$). *One can see the intricacies of the problem when there is no homoskedastic compliance, even when we have a scalar $X$. In this case, we are trying to maximize:*

$$U^{-1} := \frac{\mathbb{E}[\gamma(X)u^2(X)X^2]^2}{\mathbb{E}[X^2]\mathbb{E}[\sigma^2(X)u^2(X)X^2]} \tag{55}$$

*Suppose for instance that $X \sim U(\{-1, 1\})$ and that our policy can only choose the conditional variance of $Z$, but not the conditional compliance $\gamma(X)$ (e.g. we have a fixed binary instrument and we can only play with its randomization probability). Then we are maximizing:*

$$\frac{(\gamma_1 u_1^2 + \gamma_2 u_2^2)^2}{\sigma_1^2 u_1^2 + \sigma_2^2 u_2^2} \tag{56}$$

*over $u_1^2, u_2^2 \in [0, 1/4]$ (the variance of $Z$ for each value of $X$). Without loss of generality, we can take $\gamma_1 = 1$ and $\sigma_1 = 1$ in the above problem, since we can always take out these factors and rename $\gamma_2/\gamma_1 \to \gamma_2$ and $\sigma_2/\sigma_1 \to \sigma_2$. We then get the simplified problem:*

$$\frac{(u_1^2 + \gamma u_2^2)^2}{u_1^2 + \sigma^2 u_2^2} \tag{57}$$

*where $\gamma, \sigma \in (0, \infty)$.*

*Take for instance the case when $\sigma = 1$ (i.e. homoskedastic outcome error) and $\gamma = 1/3$ (i.e. compliance is $30\%$ less for $X = 1$ than for $X = -1$. Then the optimal solution is $u_1^2 = 1/4, u_2^2 = 0$, i.e. we don't want to randomize the instrument at all when $X = 1$, yielding a value of $1/4 = 0.25$ for $U^{-1}$. If we were for instance to randomize also when $X = 1$, then we would get a value of $(1/3)^2/(1/2) = 2/9 \approx 0.22$.*

*If on the other hand $\gamma = 1/2$, then if we randomize on both points we get $(3/8)^2/(1/2) = 9/32 \approx 0.281$, while if we only randomize when $X = 1$, then we get again $1/4 = 0.25$. Thus the relative compliance strength for different values of $X$, changes the optimal randomization solution for $Z$.*

**Remark 5** (Efficient Instrument). *It may very well be the case that the above contorted optimal solution is an artifact of the estimator that we are using. Under such a heteroskedastic compliance, most probably the estimate that we are using is not the efficient estimate, and we should be dividing the instruments by the compliance measure (i.e. construct efficient instruments, e.g. $\tilde{Z} = \frac{Z\gamma(X)}{\sigma^2(X)}$). Then maybe once we use an efficient instrument estimator, its variance most probably would always be improved if we full randomize on all $X$ points. But at least the above shows that for some fixed estimator (and in particular one that is heavily used in practice), it can very well be the case that we don't want to fully randomize the instrument all the time. If we use such an instrument $\tilde{Z}$ then observe that for this instrument, by definition $\tilde{\gamma}(X) = \sigma^2(X)$. Then U simplifies to:*

$$\mathbb{E}[XX']^{1/2}\mathbb{E}\left[\frac{u^2(X)\gamma^2(X)}{\sigma^2(X)}XX'\right]^{-1}\mathbb{E}[XX']^{1/2} \tag{58}$$

*Thus we just want to maximize $\frac{u^2(X)\gamma^2(X)}{\sigma^2(X)}$, i.e. maximize the product of the OLS coefficient of $A \sim Z$ conditional on $X$[1] and the conditional standard deviation of the instrument.*

*Revisiting the simple example in the previous remark, when we use an efficient instrument, the matrix $U^{-1}$ takes the form:*

$$u_1^2 + \frac{u_2^2\gamma^2}{\sigma^2} \tag{59}$$

*For the case when $\gamma = 1/3$ and $\sigma = 1$, if we choose $u_1^2 = u_2^2 = 1/4$, we get $1/4 + 1/36$ (compared to the $1/4$ we would achieve with the inefficient estimator under the optimal randomization policy). When $\gamma = 1/2$, with the same $u_i^2$, we would get $1/4 + 1/16 \approx 0.3125$, which is larger than the $\approx 0.281$ we would get under the optimal policy with the inefficient estimator.*

**Remark 6.** *(Non-conditional IV) Even when $X = 1$ always, it can be shown that randomizing a single binary instrument equiprobably might be sub-optimal. To show this, we will consider the setting where there is heteroskedasticty in outcomes with respected to the treatment $A$, thus with a slight abuse of notation let $\sigma^2(Z) = \mathbb{E}[\epsilon^2 \mid Z]$. Under this setting, we can simplify,*

$$U = \mathbb{E}[XX']^{1/2}\mathbb{E}[\gamma(X)u^2(X)XX']^{-1}\mathbb{E}[\sigma^2(Z)u^2(X)XX']\mathbb{E}[\gamma(X)u^2(X)XX']^{-1}\mathbb{E}[XX']^{1/2} \tag{60}$$

*to*

$$U = \frac{\mathbb{E}[\sigma^2(Z)u^2]}{\mathbb{E}[\gamma u^2]^2} \tag{61}$$

$$= \frac{\mathbb{E}[\sigma^2(Z)\mathbb{E}[\tilde{Z}^2]]}{(\gamma\mathbb{E}[\tilde{Z}^2])^2} \qquad \because u^2(X) = \mathbb{E}[\tilde{Z}^2|X] \tag{62}$$

*where $\tilde{Z}$ is the centered instrument. Now assume $\gamma = 1$ (full compliance, i.e., $A = Z$, such that $\sigma^2(Z = 0) = \sigma_0^2 =$ variance in outcome for treatment $A = 0$, and $\sigma_1^2$ is defined similarly.), and $Z \in \{0, 1\}$ with probability $p$ and $(1 - p)$,*

$$U = \frac{p^2(1-p)\sigma_0^2 + p(1-p)^2\sigma_1}{p^2(1-p)^2} \tag{63}$$

$$= \frac{p\sigma_0^2 + (1-p)\sigma_1^2}{p(1-p)} \tag{64}$$

$$= \frac{\sigma_0^2}{1-p} + \frac{\sigma_1^2}{p}. \tag{65}$$

*When $\sigma_0^2 = 1$ and $\sigma_1^2 = 2$, then the minimizer is $p \approx 0.6$, thereby illustrating that drawing the binary instrument equiprobably can be sub-optimal even in this simple setting.*

---

[1]Equivalently the efficient instrument can be viewed as the normalized residualized efficient instrument $Z\gamma(X) := \mathbb{E}[A \mid Z, X] - \mathbb{E}[A \mid X]$, and $\gamma(X)$ is the heterogeneous compliance; normalized by the conditional variance of the conditional moment $\sigma^2(X)$.

## D INFLUENCE FUNCTIONS

Influence functions have been widely studied in the machine learning literature from the perspective of robustness and interpretability (Koh & Liang, 2017; Bae et al., 2022; Schioppa et al., 2022; Loo et al., 2023), and dates back to the seminal work in robust statistics of (Cook & Weisberg, 1982). Asymptotic influence functions for two stage procedures and semi-parametric estimators have also been well-studied in the econometrics and semi-parametric inference literature (see e.g. (Ichimura & Newey, 2022; Breheny, 2020; Kahn, 2015)). Here we derive a finite sample influence function of a two-stage estimator that arises in our IV setting, from the perspective of a robust statistics definition of an influence function, that approximates the leave-one-out variation of the mean-squared-error of our estimate.

Influence functions characterize the effect of a training sample $S_i := \{X_i, Z_i, A_i, Y_i\}$ on $\hat{\theta}$ (for brevity, let $\hat{\theta} \equiv \theta(D_n)$), i.e., the change in $\hat{\theta}$ if moments associated with $S_i$ were perturbed by a small $\epsilon$. Specifically, let $\hat{\theta}$ be a solution to the $M_n(\theta, \hat{\phi}) = \mathbf{0}$ and $\hat{\phi}$ be a solution to $Q_n(\phi) = \mathbf{0}$, as defined in 3. Similarly, let $\hat{\theta}_{\epsilon, \hat{\phi}_\epsilon}$ be a solution to the following perturbed moment conditions $\bar{M}(\epsilon, \theta, \hat{\phi}_\epsilon) = \mathbf{0}$, and $\hat{\phi}_\epsilon$ is the solution to $\bar{Q}(\epsilon, \phi) = \mathbf{0}$, where

$$\bar{M}(\epsilon, \theta, \phi) = M_n(\theta, \phi) + \epsilon m(S_i, \theta, \phi) = 0 \tag{66}$$

$$\bar{Q}(\epsilon, \phi) = Q_n(\phi) + \epsilon q(S_i, \phi) = 0. \tag{67}$$

The rate of change in MSE due to an infinitesimal perturbation $\epsilon$ in the moments of $S_i$ can be obtained by the (first-order) influence function $\mathcal{I}_{\mathrm{MSE}}^{(1)}$, which itself can be decomposed using the chain-rule in terms of the (first-order) influence $\mathcal{I}_\theta^{(1)}$ on the estimated parameter $\hat{\theta}$ by perturbing the moment associated with $S_i$,

$$\mathcal{I}_{\mathrm{MSE}}^{(1)}(S_i) := \frac{\mathrm{d}}{\mathrm{d}\theta} \mathrm{MSE}\left(\hat{\theta}_{\epsilon, \hat{\phi}_\epsilon}\right) \mathcal{I}_\theta^{(1)}(S_i)\Big|_{\epsilon=0} \quad \text{where,} \quad \mathcal{I}_\theta^{(1)}(S_i) := \frac{\mathrm{d}\hat{\theta}_{\epsilon, \hat{\phi}_\epsilon}}{\mathrm{d}\epsilon}\Big|_{\epsilon=0}. \tag{68}$$

To derive the influence function for our estimate we will use the classical implicit function theorem:

**Theorem 1** (Implicit Function Theorem (Krantz & Parks, 2002)). *Consider a multi-dimensional vector-valued function $F(x, y) \in \mathbb{R}^n$, with $x \in \mathbb{R}^m$ and $y \in \mathbb{R}^n$ and fix any point $(x_0, y_0)$, such that*

$$F(x_0, y_0) = 0 \tag{69}$$

*Suppose that $F$ is differentiable and let $F_x(x, y)$ and $F_y(x, y)$, denote the Jacobians of $F$ with respect to its first and second arguments correspondingly. Suppose that $\det(F_y(x_0, y_0)) \neq 0$. Then there exists an open set $U$ containing $x_0$ and a unique function $\psi(x)$, such that $\psi(x_0) = y_0$ and such that $F(x, \psi(x)) = 0$ for all $x \in U$. Moreover, the Jacobian matrix of partial derivatives of $\psi$ in $U$ is given by:*

$$D_x\psi(x) := \left[\frac{\partial}{\partial x_j}\psi_i(x)\right]_{n \times m} = -\left[F_y(x, \psi(x))\right]^{-1} F_x(x, \psi(x)) \tag{70}$$

*If $F$ is $k$-times differentiable, then there exists an open set $U$ and a unique function $\psi$ that satisfies the above properties and is also $k$-times differentiable.*

**Theorem 2** (Influence Functions for Black-box DML Estimators). *Assuming that the inverses exist,*

$$\mathcal{I}_\theta^{(1)}(S_i) = -\frac{\partial M_n}{\partial \theta}^{-1}(\hat{\theta}, \hat{\phi})\left[m(S_i, \hat{\theta}, \hat{\phi}) - \frac{\partial M_n}{\partial \phi}(\hat{\theta}, \hat{\phi})\left[\frac{\partial Q_n}{\partial \phi}^{-1}(\hat{\phi})q(S_i, \hat{\phi})\right]\right]. \tag{71}$$

*Proof.* Let $\hat{\theta}, \hat{\phi}$ be the empirical solutions without perturbation. Note that the $\epsilon$-perturbed estimates $\theta(\epsilon), \phi(\epsilon)$ are defined as the solution to the zero equations:

$$\bar{M}(\epsilon, \theta, \phi) = M_n(\theta, \phi) + \epsilon m(S_i, \theta, \phi) = 0 \tag{72}$$

$$\bar{Q}(\epsilon, \phi) = Q_n(\phi) + \epsilon q(S_i, \phi) = 0 \tag{73}$$

Defining $x = \epsilon$, $y = (\theta; \phi)$ and $F(\epsilon, (\theta, \phi)) = [\bar{M}(\epsilon, \theta, \phi); \bar{Q}(\epsilon, \theta)]$. Suppose that:

$$F_y(0, (\hat{\theta}, \hat{\phi})) = \left[ \begin{array}{cc} D_\theta M_n(\hat{\theta}, \hat{\phi}) & D_\phi M_n(\hat{\theta}, \hat{\phi}) \\ 0 & D_\phi Q_n(\hat{\phi}) \end{array} \right] \tag{74}$$

is invertible (equiv. has a non-zero determinant). Note that for the latter it suffices that the diagonal block matrices are invertible, since it is an upper block triangular matrix.

Applying the implicit function theorem, we get that there exists an open set $U$ containing $\epsilon = 0$ and a unique function $\psi(\epsilon) = (\theta(\epsilon), \phi(\epsilon))$, such that $\psi(0) = (\hat{\theta}, \hat{\phi})$ and such that $\psi(\epsilon)$ solves the zero equations for all $\epsilon \in U$. Moreover, for all $\epsilon \in U$:

$$\left[ \begin{array}{c} D_\epsilon \theta(\epsilon) \\ D_\epsilon \phi(\epsilon) \end{array} \right] = -\left[ F_y(\epsilon, (\theta(\epsilon), \phi(\epsilon))) \right]^{-1} F_x(\epsilon, (\theta(\epsilon), \phi(\epsilon))) \tag{75}$$

Using the form of the inverse of an upper triangular matrix[2] and applying the latter at $\epsilon = 0$ we get that $\left[ \begin{array}{c} D_\epsilon \theta(0) \\ D_\epsilon \phi(0) \end{array} \right]$ takes the form:

$$-\left[ \begin{array}{cc} \left[ D_\theta M_n(\hat{\theta}, \hat{\phi}) \right]^{-1} & -\left[ D_\theta M_n(\hat{\theta}, \hat{\phi}) \right]^{-1} D_\phi M_n(\hat{\theta}, \hat{\phi}) \left[ D_\phi Q_n(\hat{\phi}) \right]^{-1} \\ 0 & \left[ D_\phi Q_n(\hat{\phi}) \right]^{-1} \end{array} \right] \left[ \begin{array}{c} m(S_i, \hat{\theta}, \hat{\phi}) \\ q(S_i, \hat{\phi}) \end{array} \right] \tag{76}$$

Since the influence function is $\mathcal{I}_\theta^{(1)}(S_i) := D_\epsilon \theta(0)$, we get the result by applying the matrix multiplication.

Higher order influence functions can be calculated in a similar manner, using repeated applications of the chain rule and the implicit function theorem. Note that writing an explicit form for a general $k$-th order can often be tedious, and computing it can be practically challenging. Discussion about higher-order influences for the **single-stage** estimators can be found in the works by Alaa & Van Der Schaar (2019) and Robins et al. (2008). As we formally show in Theorem 5, for our purpose, we recommend using $K = 1$ as it suffices to dramatically reduce the variance of the gradient estimator, incurring bias that decays quickly, while also being easy to compute $\qquad \square$

*(Alternative) Informal Proof.* Here we provide an alternate way to derive the form of the influence function without invoking the implicit function theorem. This derivation only makes use of the chain rule of the derivatives. In the following, we will use $\partial f$ to denote partial derivative with respect to the immediate argument of the function, whereas $\mathrm{d}f$ is for the total derivative. That is, let $f(x, g(x)) := xg(x)$ then

$$\frac{\partial f}{\partial x}(x, g(x)) = g(x) \qquad\qquad \frac{\mathrm{d}f}{\mathrm{d}x}(x, g(x)) = g(x) + x\frac{\partial g}{\partial x}(x). \tag{77}$$

From 67 we know that $\hat{\theta}_{\epsilon, \hat{\phi}_\epsilon}$ is the solution to the moment conditions $\bar{M}\left(\epsilon, \theta, \hat{\phi}_\epsilon\right) = \mathbf{0}$

$$\bar{M}\left(\epsilon, \hat{\theta}_{\epsilon, \hat{\phi}_\epsilon}, \hat{\phi}_\epsilon\right) = \frac{1}{n}\sum_{j=1}^n m(S_j, \hat{\theta}_{\epsilon, \hat{\phi}_\epsilon}, \hat{\phi}_\epsilon) + \epsilon\, m(S_i, \hat{\theta}_{\epsilon, \hat{\phi}_\epsilon}, \hat{\phi}_\epsilon) = 0. \tag{78}$$

Therefore,

$$\frac{\mathrm{d}\bar{M}}{\mathrm{d}\epsilon}(\epsilon, \hat{\theta}_{\epsilon, \hat{\phi}_\epsilon}, \hat{\phi}_\epsilon) = \frac{1}{n}\sum_{j=1}^n \frac{\mathrm{d}m}{\mathrm{d}\epsilon}(S_j, \hat{\theta}_{\epsilon, \hat{\phi}_\epsilon}, \hat{\phi}_\epsilon) + m(S_i, \hat{\theta}_{\epsilon, \hat{\phi}_\epsilon}, \hat{\phi}_\epsilon) + \epsilon\frac{\mathrm{d}m}{\mathrm{d}\epsilon}(S_i, \hat{\theta}_{\epsilon, \hat{\phi}_\epsilon}, \hat{\phi}_\epsilon) \tag{79}$$

$$= \frac{\mathrm{d}M_n}{\mathrm{d}\epsilon}(\hat{\theta}_{\epsilon, \hat{\phi}_\epsilon}, \hat{\phi}_\epsilon) + m(S_i, \hat{\theta}_{\epsilon, \hat{\phi}_\epsilon}, \hat{\phi}_\epsilon) + \epsilon\frac{\mathrm{d}m}{\mathrm{d}\epsilon}(S_i, \hat{\theta}_{\epsilon, \hat{\phi}_\epsilon}, \hat{\phi}_\epsilon) \tag{80}$$

$$= \frac{\partial M_n}{\partial \theta}(\hat{\theta}_{\epsilon, \hat{\phi}_\epsilon}, \hat{\phi}_\epsilon)\frac{\mathrm{d}\hat{\theta}_{\epsilon, \hat{\phi}_\epsilon}}{\mathrm{d}\epsilon} + \frac{\partial M_n}{\partial \phi}(\hat{\theta}_{\epsilon, \hat{\phi}_\epsilon}, \hat{\phi}_\epsilon)\frac{\mathrm{d}\hat{\phi}_\epsilon}{\mathrm{d}\epsilon} + m(S_i, \hat{\theta}_{\epsilon, \hat{\phi}_\epsilon}, \hat{\phi}_\epsilon) + \epsilon\frac{\mathrm{d}m}{\mathrm{d}\epsilon}(S_i, \hat{\theta}_{\epsilon, \hat{\phi}_\epsilon}, \hat{\phi}_\epsilon). \tag{81}$$

---

[2] For any block matrix: $\left[ \begin{array}{cc} A & X \\ 0 & B \end{array} \right]^{-1} = \left[ \begin{array}{cc} A^{-1} & -A^{-1}XB^{-1} \\ 0 & B^{-1} \end{array} \right].$

Further, note that,

$$\frac{\mathrm{d}\hat{\theta}_{\epsilon,\hat{\phi}_\epsilon}}{\mathrm{d}\epsilon} = \frac{\partial\hat{\theta}_{\epsilon,\hat{\phi}_\epsilon}}{\partial\epsilon} + \frac{\partial\hat{\theta}_{\epsilon,\hat{\phi}_\epsilon}}{\partial\phi}\frac{\partial\hat{\phi}_\epsilon}{\partial\epsilon}. \tag{82}$$

Using the fact that $\mathrm{d}\bar{M}(\epsilon, \hat{\theta}_{\epsilon,\hat{\phi}_\epsilon}, \hat{\phi}_\epsilon)/\mathrm{d}\epsilon = 0$, rearranging terms in equation 81,

$$\frac{\mathrm{d}\hat{\theta}_{\epsilon,\hat{\phi}_\epsilon}}{\mathrm{d}\epsilon} = -\frac{\partial M_n}{\partial\theta}^{-1}(\hat{\theta}_{\epsilon,\hat{\phi}_\epsilon}, \hat{\phi}_\epsilon)\left[m(S_i, \hat{\theta}_{\epsilon,\hat{\phi}_\epsilon}, \hat{\phi}_\epsilon) + \epsilon\frac{\mathrm{d}m}{\mathrm{d}\epsilon}(S_i, \hat{\theta}_{\epsilon,\hat{\phi}_\epsilon}, \hat{\phi}_\epsilon) + \frac{\partial M_n}{\partial\phi}(\hat{\theta}_{\epsilon,\hat{\phi}_\epsilon}, \hat{\phi}_\epsilon)\frac{\mathrm{d}\hat{\phi}_\epsilon}{\mathrm{d}\epsilon}.\right] \tag{83}$$

Now we expand the terms in blue. Similar to earlier, since $\hat{\phi}_\epsilon$ is the solution to the moment condition $\bar{Q}(\epsilon, \phi) = \mathbf{0}$,

$$Q\left(\epsilon, \hat{\phi}_\epsilon\right) = \frac{1}{n}\sum_{j=1}^{n} q(S_j, \hat{\phi}_\epsilon) + \epsilon\, q(S_i, \hat{\phi}_\epsilon) = 0. \tag{84}$$

Therefore,

$$\frac{\mathrm{d}\bar{Q}}{\mathrm{d}\epsilon}\left(\epsilon, \hat{\phi}_\epsilon\right) = \frac{1}{n}\sum_{j=1}^{n}\frac{\mathrm{d}q}{\mathrm{d}\epsilon}(S_j, \hat{\phi}_\epsilon) + q(S_i, \hat{\phi}_\epsilon) + \epsilon\frac{\mathrm{d}q}{\mathrm{d}\epsilon}(S_i, \hat{\phi}_\epsilon) \tag{85}$$

$$= \frac{\mathrm{d}Q_n}{\mathrm{d}\epsilon}(\hat{\phi}_\epsilon) + q(S_i, \hat{\phi}_\epsilon) + \epsilon\frac{\mathrm{d}q}{\mathrm{d}\epsilon}(S_i, \hat{\phi}_\epsilon) \tag{86}$$

$$= \frac{\partial Q_n}{\partial\phi}(\hat{\phi}_\epsilon)\frac{\mathrm{d}\hat{\phi}_\epsilon}{\mathrm{d}\epsilon} + q(S_i, \hat{\phi}_\epsilon) + \epsilon\frac{\mathrm{d}q}{\mathrm{d}\epsilon}(S_i, \hat{\phi}_\epsilon) \tag{87}$$

Now using the fact that $\mathrm{d}\bar{Q}(\epsilon, \hat{\phi}_\epsilon)/\mathrm{d}\epsilon = 0$, rearranging terms in equation 87,

$$\frac{\mathrm{d}\hat{\phi}_\epsilon}{\mathrm{d}\epsilon} = -\frac{\partial Q_n}{\partial\phi}^{-1}(\hat{\phi}_\epsilon)\left[q(S_i, \hat{\phi}_\epsilon) + \epsilon\frac{\mathrm{d}q}{\mathrm{d}\epsilon}(S_i, \hat{\phi}_\epsilon)\right]. \tag{88}$$

Combining equation 83 and equation 88,

$$\frac{\mathrm{d}\hat{\theta}_{\epsilon,\hat{\phi}_\epsilon}}{\mathrm{d}\epsilon} = -\frac{\partial M_n}{\partial\theta}^{-1}(\hat{\theta}_{\epsilon,\hat{\phi}_\epsilon}, \hat{\phi}_\epsilon)\left[m(S_i, \hat{\theta}_{\epsilon,\hat{\phi}_\epsilon}, \hat{\phi}_\epsilon) + \epsilon\frac{\mathrm{d}m}{\mathrm{d}\epsilon}(S_i, \hat{\theta}_{\epsilon,\hat{\phi}_\epsilon}, \hat{\phi}_\epsilon)\right. \tag{89}$$

$$\left. -\frac{\partial M_n}{\partial\phi}(\hat{\theta}_{\epsilon,\hat{\phi}_\epsilon}, \hat{\phi}_\epsilon)\left[\frac{\partial Q_n}{\partial\phi}^{-1}(\hat{\phi}_\epsilon)\left[q(S_i, \hat{\phi}_\epsilon) + \epsilon\frac{\mathrm{d}q}{\mathrm{d}\epsilon}(S_i, \hat{\phi}_\epsilon)\right]\right]\right]. \tag{90}$$

At $\epsilon = 0$, notice that $\hat{\theta}_{0,\hat{\phi}_0} = \hat{\theta}$ and $\hat{\phi}_0 = \hat{\phi}$. Therefore, using equation 90,

$$\left.\frac{\mathrm{d}\hat{\theta}_{\epsilon,\hat{\phi}_\epsilon}}{\mathrm{d}\epsilon}\right|_{\epsilon=0} = -\frac{\partial M_n}{\partial\theta}^{-1}(\hat{\theta}, \hat{\phi})\left[m(S_i, \hat{\theta}, \hat{\phi}) - \frac{\partial M_n}{\partial\phi}(\hat{\theta}, \hat{\phi})\left[\frac{\partial Q_n}{\partial\phi}^{-1}(\hat{\phi})q(S_i, \hat{\phi})\right]\right]. \tag{91}$$

$\square$

# E    BIAS AND VARIANCES OF GRADIENT ESTIMATORS

To convey the key insights, we will consider $\nabla\log\pi(S_i) \in \mathbb{R}$ to be scalar for simplicity. Similar results would follow when $\nabla\log\pi(S_i) \in \mathbb{R}^d$.

## E.1    NAIVE REINFORCE ESTIMATOR

For any random variable $Z$, we let $\|Z\|_{L^p} = (\mathbb{E}[Z^p])^{1/p}$.

**Theorem 3.** $\mathbb{E}\left[\hat{\nabla}\mathcal{L}(\pi)\right] = \nabla\mathcal{L}(\pi)$. *Moreover, let:*

$$X_1 = \mathrm{MSE}\big(\theta(\{S_i\}_{j=2}^n)\big) \tag{92}$$

denote the leave-one-out MSE and let:

$$\Delta_1 = \text{MSE}\big(\theta(\{S_i\}_{j=1}^n)\big) - \text{MSE}\big(\theta(\{S_i\}_{j=2}^n)\big) \tag{93}$$

denote the leave-one-out stability. Suppose that for a sufficiently large $n$,

$$\|X_1\|_{L^4} \leq C_{2,4} \|X_1\|_{L^2} \tag{94}$$

and for $Y_i := \nabla \log \pi(S_i)$, let $\forall i, \|Y_i\|_{L^\infty} \leq C$ and $\|Y_i\|_{L^2} \geq c > 0$ for some universal constants $c, C, C_{2,4}$. Then the variance of the naive REINFORCE estimator is upper and lower bounded as:

$$\frac{c^2}{2} n\|X_1\|_{L^2}^2 - O\big(n^2\|\Delta_1\|_{L^2}^2\big) \leq \text{Var}\Big(\hat{\nabla}\mathcal{L}(\pi)\Big) \leq \frac{3C^2}{2} n\|X_1\|_{L^2}^2 + O\big(n^2\|\Delta_1\|_{L^2}^2\big) \tag{95}$$

Thus, when $n\|\Delta_1\|_{L^2} = O(1)$ and $n\|\text{MSE}\big(\theta(\{S_i\}_{j=2}^n)\big)\|_{L^2}^2 = \Omega(\alpha(n)) = \omega(1)$, we have

$$\text{Var}\Big(\hat{\nabla}\mathcal{L}(\pi)\Big) = \Omega(\alpha(n)). \tag{96}$$

**Remark 7.** *In many cases, we would expect that the mean squared error decays at the rate of $n^{-1/2}$ (unless we can invoke a fast parametric rate of $1/n$, which holds under strong convexity assumptions); in such cases, we have that $\alpha(n) = \omega(1)$ and $\text{Var}\Big(\hat{\nabla}\mathcal{L}(\pi)\Big)$ can grow with the sample size. Moreover, if the mean squared error does not converge to zero, due to persistent approximation error, or local optima in the optimization, then the variance can grow linearly, i.e. $\alpha(n) = \Theta(n)$.*

**Remark 8.** *The quantity $\Delta$ is a leave-one-out stability quantity. Hence, the property that $n\|\Delta\|_{L^2} = O(1)$ is a leave-one-out-stability property of the estimator, which is a well-studied concept (see e.g. Kearns & Ron (1997); Celisse & Guedj (2016); Abou-Moustafa & Szepesvári (2019)). It states that the estimator is $1/n$-leave-one-out-stable. This will typically hold for many M-estimators over parametric spaces.*

*Proof.* In the following, we first define some new notation to make the proof more concise. Subsequently, we show (I) unbiasedness, and then (II) we discuss the variance of $\hat{\nabla}\mathcal{L}(\pi)$.

Let $D_n$ be the entire dataset $\{S_i\}_{i=1}^n$, where $S_i$ is the i-th sample.

$$Z_i := \text{MSE}(\theta(\{S_i\}_{i=1}^n))\nabla \log \pi(S_i) \tag{97}$$

$$X := \text{MSE}(\theta(\{S_i\}_{i=1}^n)) \tag{98}$$

$$X_i := \text{MSE}\big(\theta(\{S_i\}_{j=1,j\neq i}^n)\big) \tag{99}$$

$$\Delta_i := X - X_i \tag{100}$$

$$Y_i := \nabla \log \pi(S_i) \tag{101}$$

**(I) Unbiased**

$$\mathbb{E}\Big[\hat{\nabla}\mathcal{L}(\pi)\Big] = \mathbb{E}_\pi\left[\sum_i \text{MSE}(\theta(\{S_i\}_{i=1}^n))\nabla \log \pi(S_i)\right] \tag{102}$$

$$\overset{(a)}{=} \mathbb{E}_\pi[\text{MSE}(\theta(D_n))\nabla \log \text{Pr}(D_n; \pi)] \tag{103}$$

$$= \nabla\mathbb{E}_\pi[\text{MSE}(\theta(D_n))] \tag{104}$$

$$= \nabla\mathcal{L}(\pi), \tag{105}$$

where $\mathbb{E}_\pi$ indicates that the dataset $D_n$ is sampled using $\pi$, and $(a)$ follows from 5.

**(II) Variance**

$$\text{Var}\Big(\hat{\nabla}\mathcal{L}(\pi)\Big) = \text{Var}\Big(\sum Z_i\Big) = \text{Var}\left(X \sum_{i=1}^n Y_i\right). \tag{106}$$

Notice $\{Y_i\}_{i=1}^n$ are i.i.d random variables, and $X$ is *dependent* on all of $\{Y_i\}_{i=1}^n$, which makes the analysis more involved. We begin with the following alternate expansion for variance,

$$\text{Var}\Big(\sum Z_i\Big) = \sum_i \sum_j \text{Cov}(Z_i, Z_j), \tag{107}$$

where, $\quad \text{Cov}(Z_i, Z_j) = \text{Cov}(XY_i, XY_j) = \text{Cov}((X_i + \Delta_i)Y_i, (X_i + \Delta_i)Y_j). \tag{108}$

Therefore,

$$\text{Var}\Big(\sum Z_i\Big) = \sum_{i,j} \text{Cov}(X_i Y_i, X_i Y_j) + \text{Cov}(X_i Y_i, \Delta_i Y_j) + \text{Cov}(\Delta_i Y_i, X_i Y_j) + \text{Cov}(\Delta_i Y_i, \Delta_i Y_j).$$
(109)

Now, observe that as $X_i \perp\!\!\!\perp Y_i$, $Y_i \perp\!\!\!\perp Y_j$, and $\mathbb{E}[Y_i] = \mathbb{E}[X_i Y_i] = \mathbb{E}[X_i Y_j] = 0$. Therefore,

$$\text{Cov}(X_i Y_i, X_i Y_j) = \mathbb{E}[X_i^2 Y_i Y_j] = \mathbb{E}[X_i^2 Y_j]\mathbb{E}[Y_i] = 0, \qquad \forall i \neq j \quad (110)$$

$$\text{Cov}(X_i Y_i, X_i Y_j) = \mathbb{E}[X_i^2 Y_i^2], \qquad \forall i = j. \quad (111)$$

For all $i, j$:

$$\text{Cov}(X_i Y_i, \Delta_i Y_j) = \mathbb{E}[X_i \Delta_i Y_i Y_j] - \mathbb{E}[X_i Y_i]\mathbb{E}[\Delta_i Y_j] = \mathbb{E}[X_i \Delta_i Y_i Y_j], \quad (112)$$

$$\text{Cov}(\Delta_i Y_i, X_i Y_j) = \mathbb{E}[X_i \Delta_i Y_i Y_j] - \mathbb{E}[\Delta_i Y_i]\mathbb{E}[X_i Y_j], \quad (113)$$

$$|\text{Cov}(\Delta_i Y_i, \Delta_i Y_j)| \overset{(a)}{\leq} \sqrt{\mathbb{E}[\Delta_i^2 Y_i^2]\mathbb{E}[\Delta_i^2 Y_j^2]} \leq C^2 \, \mathbb{E}[\Delta_i^2], \quad (114)$$

where (a) follows from Cauchy-Schwarz by noting that for any two random variables $A$ and $B$, $\text{Cov}(A, B) \leq \sqrt{\text{Var}(A)\text{Var}(B)}$ and that $\text{Var}(A) \leq \mathbb{E}[A^2]$. If we let $\bar{Y} = \sum_{j=1}^n Y_i$, we have:[3]

$$\left|\text{Var}\Big(\sum Z_i\Big) - \sum_i \mathbb{E}[X_i^2 Y_i^2]\right| \leq \left|\sum_i \sum_j 2\mathbb{E}[X_i \Delta_i Y_i Y_j] - \mathbb{E}[\Delta_i Y_i]\mathbb{E}[X_i Y_j]\right| + n^2 C^2 \mathbb{E}[\Delta_1^2]$$
(115)

$$= \left|\sum_i 2\mathbb{E}[X_i \Delta_i Y_i \bar{Y}] - \mathbb{E}[\Delta_i Y_i]\mathbb{E}[X_i \bar{Y}]\right| + n^2 C^2 \mathbb{E}[\Delta_1^2]. \quad (116)$$

Now, by assumption we have that $\|Y_i\|_{L^\infty} \leq C$ for some constant $C$. Further, from Cauchy-Schwarz, $|\mathbb{E}[AB]| \leq \sqrt{\mathbb{E}[A^2]\mathbb{E}[B^2]}$ for any two random variable $A$ and $B$. Thus we get:

$$\left|\mathbb{E}[\Delta_i Y_i]\mathbb{E}[X_i \bar{Y}]\right| \leq \sqrt{\mathbb{E}[\Delta_i^2]\mathbb{E}[Y_i^2]}\sqrt{\mathbb{E}[X_i^2]}\sqrt{\mathbb{E}[\bar{Y}^2]} \quad (117)$$

$$\overset{(a)}{=} \sqrt{\mathbb{E}[\Delta_i^2]\mathbb{E}[Y_i^2]}\sqrt{\mathbb{E}[X_i^2]}\sqrt{n\mathbb{E}[Y_i^2]} \quad (118)$$

$$\leq C^2 \sqrt{n\mathbb{E}[\Delta_i^2]\mathbb{E}[X_i^2]} \qquad \because \|Y_i\|_{L^\infty} \leq C \quad (119)$$

$$= C^2 \sqrt{n}\|\Delta_i\|_{L^2}\|X_i\|_{L^2} \quad (120)$$

where (a) follows from observing that $\mathbb{E}[(\sum_i Y_i)^2] = \mathbb{E}[\sum_{i,j} Y_i Y_j] = \mathbb{E}[\sum_i Y_i^2] = n\mathbb{E}[Y_i^2]$ as $\mathbb{E}[Y_i Y_j] = 0$ for $i \neq j$. The final expression uses that $\sqrt{\mathbb{E}[X_i^2]} = \|X_i\|_{L^2}$. Similarly,

$$\left|\mathbb{E}[X_i \Delta_i Y_i \bar{Y}]\right| \leq \sqrt{\mathbb{E}[\Delta_i^2 Y_i^2]}\sqrt{\mathbb{E}[X_i^2 \bar{Y}^2]} \leq C\sqrt{\mathbb{E}[\Delta_i^2]}\left(\mathbb{E}[X_i^4]\mathbb{E}[\bar{Y}^4]\right)^{1/4} \quad (121)$$

Now by the Marcinkiewicz-Zygmund inequality Rio (2009) we have that $\left\|\frac{1}{\sqrt{n}}\sum_i Y_i\right\|_{L^p} \leq C_p\|Y_i\|_{L^p}$ for any $p \geq 2$ and for some universal constant $C_p$. Therefore, $(\mathbb{E}[\bar{Y}^4])^{1/4} \leq \sqrt{n}C_4\|Y_i\|_{L^4}$ and

$$\left|\mathbb{E}[X_i \Delta_i Y_i \bar{Y}]\right| \leq CC_4\|\Delta_i\|_{L^2}\|X_i\|_{L^4}\sqrt{n}\|Y_i\|_{L^4} \quad (122)$$

$$\leq C^2 C_4 \sqrt{n}\|\Delta_i\|_{L^2}\|X_i\|_{L^4} \quad (123)$$

$$\leq C^2 C_4 C_{2,4} \sqrt{n}\|\Delta_i\|_{L^2}\|X_i\|_{L^2}, \quad (124)$$

where the last line follows as $\|X_i\|_{L^4} \leq C_{2,4}\|X_i\|_{L^2}$. Thus for some constant $C_1$, combining 116, 120, and 124, we conclude that:

$$\left|\text{Var}\Big(\sum Z_i\Big) - \sum_i \mathbb{E}[X_i^2 Y_i^2]\right| \leq C_1 \sum_i \sqrt{n}\|\Delta_i\|_{L^2}\|X_i\|_{L^2} + n^2 C^2 \|\Delta_1\|_{L^2}^2. \quad (125)$$

---

[3]We note here that an important aspect of the proof is to first push the summation inside to get $\bar{Y}$ and then to apply the Cauchy-Schwarz inequality, since $\bar{Y}$ concentrates around $\sqrt{n}$ and one saves an important extra $\sqrt{n}$ factor.

Recall from the AM-GM inequality that for any $a$ and $b$: $a \cdot b \leq \frac{1}{2\eta}a^2 + 2\eta b^2$ for any $\eta > 0$. Let $a = C_1\sqrt{n}\|\Delta_i\|$ and $b = \|X_i\|_{L^2} = \sqrt{\mathbb{E}[X_i^2]}$ and $\eta = c^2/4$, then

$$\left|\text{Var}\left(\sum Z_i\right) - \sum_i \mathbb{E}[X_i^2 Y_i^2]\right| \leq \sum_i \left(\frac{2C_1^2 n\|\Delta_i\|_{L^2}^2}{c^2} + \frac{c^2}{2}\mathbb{E}[X_i^2]\right) + n^2 C^2\|\Delta_1\|_{L^2}^2. \quad (126)$$

Since $c^2 \leq \mathbb{E}[Y_i^2] \leq C^2$ and $X_i \perp\!\!\!\perp Y_i$, we have $\mathbb{E}[X_i^2 Y_i^2] = \mathbb{E}[X_i^2]\mathbb{E}[Y_i^2] \leq C^2\mathbb{E}[X_i^2]$, and $\mathbb{E}[X_i^2 Y_i^2] \geq c^2\mathbb{E}[X_i^2]$. Further, as $|A - B| \leq C \implies A \leq B + C$ and $A \geq B - C$, we obtain the lower bound:

$$\text{Var}\left(\sum Z_i\right) \geq \sum_i \left(\mathbb{E}[X_i^2]c^2 - \frac{2C_1^2 n\|\Delta_i\|_{L^2}^2}{c^2} - \frac{c^2}{2}\mathbb{E}[X_i^2]\right) - n^2 C^2\|\Delta_1\|_{L^2}^2 \quad (127)$$

$$= n\frac{c^2}{2}\mathbb{E}[X_1^2] - O(n^2\|\Delta_1\|_{L^2}^2). \quad (128)$$

Similarly, for the upper bound:

$$\text{Var}\left(\sum Z_i\right) \leq \sum_i \left(\mathbb{E}[X_i^2]C^2 + \frac{2C_1^2 n\|\Delta_i\|_{L^2}^2}{c^2} + \frac{c^2}{2}\mathbb{E}[X_i^2]\right) + n^2 C^2\|\Delta_1\|_{L^2}^2 \quad (129)$$

$$\leq n\frac{3C^2}{2}\mathbb{E}[X_1^2] + O\left(n^2\|\Delta_1\|_{L^2}^2\right). \qquad \because c^2 \leq C^2$$

Therefore, we obtain our result

$$\frac{c^2}{2}n\|X_1\|_{L^2}^2 - O\left(n^2\|\Delta_1\|_{L^2}^2\right) \leq \text{Var}\left(\hat{\nabla}\mathcal{L}(\pi)\right) \leq \frac{3C^2}{2}n\|X_1\|_{L^2}^2 + O\left(n^2\|\Delta_1\|_{L^2}^2\right). \quad (130)$$

$\square$

## E.2 REINFORCE ESTIMATOR WITH CV

**Theorem 4.** $\mathbb{E}\left[\hat{\nabla}^{CV}\mathcal{L}(\pi)\right] = \nabla\mathcal{L}(\pi)$. *Moreover, let:*

$$\Delta_1 = \text{MSE}\left(\theta(\{S_i\}_{j=1}^n)\right) - \text{MSE}\left(\theta(\{S_i\}_{j=2}^n)\right) \quad (131)$$

*denote the leave-one-out stability. Let $Y_i = \nabla\log\pi(S_i)$ and suppose that and that $\|Y_i\|_{L^\infty} \leq C$ for some universal constant $C$. Then the variance of the CV REINFORCE estimator is upper bounded as:*

$$\text{Var}\left(\hat{\nabla}^{CV}\mathcal{L}(\pi)\right) \leq C^2 n^2\|\Delta_1\|_{L^2}^2 \quad (132)$$

*Thus if the estimator satisfies a $n^{-1}$-leave-one-out stability, i.e. $n\|\Delta_1\|_{L^2} = O(1)$, then we have:*

$$\text{Var}\left(\hat{\nabla}^{CV}\mathcal{L}(\pi)\right) \leq O(1). \quad (133)$$

**Remark 9.** *This holds irrespective of $\alpha(n)$, thus providing a more reliable gradient estimator even if the function approximator is mis-specified or optimization is susceptible to local optima.*

*Proof.* Let $D_n$ be the entire dataset $\{S_i\}_{i=1}^n$, where $S_i$ is the i-th sample.

$$Z_i = \nabla\log\pi(S_i)\left[\text{MSE}\left(\theta(\{S_j\}_{j=1}^n)\right) - \text{MSE}\left(\theta(\{S_j\}_{j=1, j\neq i}^n)\right)\right] \quad (134)$$

$$\Delta_i = \text{MSE}\left(\theta(\{S_j\}_{j=1}^n)\right) - \text{MSE}\left(\theta(\{S_j\}_{j=1, j\neq i}^n)\right) \quad (135)$$

$$Y_i = \nabla\log\pi(S_i) \quad (136)$$

**(I) Unbiased**

$$\mathbb{E}\left[\hat{\nabla}^{\text{CV}}\mathcal{L}(\pi)\right] = \mathbb{E}_\pi\left[\sum_i \nabla \log \pi(S_i)\left[\text{MSE}\left(\theta(\{S_j\}_{j=1}^n)\right) - \text{MSE}\left(\theta(\{S_j\}_{j=1, j\neq i}^n)\right)\right]\right] \tag{137}$$

$$= \mathbb{E}_\pi\left[\hat{\nabla}\mathcal{L}(\pi)\right] - \mathbb{E}_\pi\left[\sum_i \nabla \log \pi(S_i)\text{MSE}\left(\theta(\{S_j\}_{j=1, j\neq i}^n)\right)\right] \tag{138}$$

$$\stackrel{(a)}{=} \nabla\mathcal{L}(\pi) - \mathbb{E}_\pi\left[\sum_i \text{MSE}\left(\theta(D_{n\setminus i})\right)\underbrace{\mathbb{E}_\pi\left[\nabla \log \pi(S_i)\big|D_{n\setminus i}\right]}_{=0}\right] \tag{139}$$

$$= \nabla\mathcal{L}(\pi), \tag{140}$$

where (a) follows as $S_i \perp\!\!\!\perp D_{n\setminus i}$, and $\mathbb{E}_\pi[\nabla \log \pi(S_i)] = 0$.

**(II) Variance**

Let $\bar{Z}_i = Z_i - \mathbb{E}[Z_i]$. First, observe that

$$\frac{1}{n^2}\text{Var}\left(\sum_i Z_i\right) = \text{Var}\left(\frac{1}{n}\sum_i Z_i\right) = \mathbb{E}\left[\left(\frac{1}{n}\sum_i \bar{Z}_i\right)^2\right]. \tag{141}$$

From Jensen's inequality, as $\left(\frac{1}{n}\sum_i \bar{Z}_i\right)^2 \leq \frac{1}{n}\sum_i \left(\bar{Z}_i\right)^2$,

$$\frac{1}{n^2}\text{Var}\left(\sum_i Z_i\right) \leq \frac{1}{n}\sum_i \mathbb{E}[\bar{Z}_i^2] = \frac{1}{n}\sum_i \text{Var}(Z_i). \tag{142}$$

Therefore,

$$\text{Var}\left(\hat{\nabla}^{\text{CV}}\mathcal{L}(\pi)\right) = \text{Var}\left(\sum Z_i\right) \leq n\sum \text{Var}(Z_i) \leq n^2\mathbb{E}[\Delta_1^2 Y_1^2] \leq n^2 C^2\mathbb{E}[\Delta_1^2]. \tag{143}$$

$$\square$$

### E.3   REINFORCE ESTIMATOR WITH INFLUENCE

**Theorem 5.** $\mathbb{E}\left[\hat{\nabla}^{IF}\mathcal{L}(\pi)\right] = \nabla\mathcal{L}(\pi) + \mathcal{O}(n^{-K})$. *Moreover, let:*

$$\Delta_1 = \text{MSE}\left(\theta(\{S_i\}_{j=1}^n)\right) - \text{MSE}\left(\theta(\{S_i\}_{j=2}^n)\right) \tag{144}$$

*denote the leave-one-out stability. Let* $\text{MSE}\left(\theta(\{S_i\}_{j=1}^n)\right)$ *admit a convergent Taylor series expansion. Let* $Y_i = \nabla \log \pi(S_i)$ *and suppose that and that* $\|Y_i\|_{L^\infty} \leq C$ *for some universal constant C. Then the variance of the Influence function based REINFORCE estimator is upper bounded as:*

$$\text{Var}\left(\hat{\nabla}^{IF}\mathcal{L}(\pi)\right) \leq O\left(n^2\|\Delta_1\|_{L^2}^2 + n^{2(1-K)}\right) \tag{145}$$

*Thus if the estimator satisfies a* $n^{-1}$*-leave-one-out stability, i.e.* $n\|\Delta_1\|_{L^2} = O(1)$*, and* $K \geq 1$*, then we have:*

$$\text{Var}\left(\hat{\nabla}^{IF}\mathcal{L}(\pi)\right) \leq O(1). \tag{146}$$

*Proof.* Let $\{Y_i\}_{i=1}^n$ be i.i.d random variables, and let $\Delta_i$ be a variable dependent on all of $\{Y_i\}_{i=1}^n$.

$$Z_i = \nabla \log \pi(S_i)\tilde{\Delta}_i \tag{147}$$

$$\tilde{\Delta}_i = \sum_{k=1}^K \frac{(-1/n)^k}{k!}\mathcal{I}_{\text{MSE}}^{(k)} = \Delta_i + O(n^{-K}) \tag{148}$$

$$\Delta_i = \text{MSE}\left(\theta(\{S_j\}_{j=1}^n)\right) - \text{MSE}\left(\theta(\{S_j\}_{j=1, j\neq i}^n)\right) \tag{149}$$

$$Y_i = \nabla \log \pi(S_i) \tag{150}$$

Notice $\{Y_i\}_{i=1}^n$ are i.i.d random variables, and $\tilde{\Delta}_i$ is dependent on all of $\{Y_i\}_{i=1}^n$. The bias follows from standard results on $K$-th order remainder term in a convergent Taylor series expansion. Finally, it can be observed that $\mathrm{Var}(\sum Z_i)$ follows the exact same analysis as done for $\hat{\nabla}^{\mathrm{CV}}\mathcal{L}(\pi)$:

$$\mathrm{Var}\left(\hat{\nabla}^{\mathrm{IF}}\mathcal{L}(\pi)\right) \leq n^2 C^2 \mathbb{E}[\tilde{\Delta}_1^2] \leq n^2 C^2 2\mathbb{E}[(\Delta_1^2 + O(n^{-2K})] = O\left(n^2 \mathbb{E}[\Delta_1^2] + \frac{n^2}{n^{2K}}\right). \quad (151)$$

$\square$

# F  ESTIMATION ERROR, BIAS AND VARIANCE ANALYSIS

## F.1  PRELIMINARIES ON U-STATISTICS

We first present a result on the variance of U-statistics with a growing order that primarily follows from recent prior work. For references see Fan et al. (2018); Hoeffding (1948); Wager & Athey (2018b) and some concise course notes [A, B, C, D, E]. With a slight abuse of notation, in the following let $Z_S$ represents a set of $S$ random variables (and is unrelated to notation for instrumental variables). Let $n$ be the number of random variables available and $k$ be the size of the selected random variables.

**Theorem 6** (Variance of U-Statistics). *Consider any $k$-order U-statistic*

$$U = \frac{1}{\binom{n}{k}} \sum_{S \subseteq [n]:|S|=k} h(Z_S)$$

*where $h$ is a symmetric function in its arguments. Let:*

$$\eta_c = \mathrm{Var}(\mathbb{E}[h(Z) \mid Z_{1:c}]) \quad (152)$$

*Then, we have that:*

$$\mathrm{Var}(U) = \binom{n}{k}^{-1} \sum_{c=1}^{k} \binom{k}{c}\binom{n-k}{k-c}\eta_c. \quad (153)$$

*Moreover, one can show that:*

$$\frac{k^2}{n}\eta_1 \leq \mathrm{Var}(U) \leq \frac{k^2}{n}\eta_1 + \frac{k^2}{n^2}\eta_k, \quad (154)$$

*and that*

$$\eta_1 \leq \frac{1}{k}\eta_k. \quad (155)$$

These results, for instance, follow from the classical Hoeffding's canonical decomposition of a U-statistic and an inequality due to Wager & Athey (2018b), to upper bound the non-leading term, for every $k, n, \eta_1, \ldots, \eta_k$ not just asymptotically and for fixed $k, \eta_1, \ldots, \eta_k$, and letting $n \to \infty$ as in Hoeffding's classical analysis. The proof is included in Appendix F.5.

Typically, due to computational considerations, one considers an incomplete version of a U-statistic, where instead of calculating all possible subsets of size $k$, we draw $B$ random subsets $\{S_1, \ldots, S_b\}$, each of size $k$, uniformly at random among all subsets of size $k$. Then we define the incomplete U-statistic approximation as:

$$\hat{U} = \frac{1}{B}\sum_{b=1}^{B} h(Z_{S_b}) \quad (156)$$

We can also derive the following finite sample variance characterization for an incomplete U-statistic:

**Lemma 2.** *The variance of the incomplete U-statistic $\hat{U}$ is:*

$$\mathrm{Var}(\hat{U}) = \left(1 - \frac{1}{B}\right)\mathrm{Var}(U) + \frac{1}{B}\eta_k. \quad (157)$$

*Proof.* Note that $\hat{U}$ is an unbiased estimate of $U$, conditional on the samples, i.e.:

$$\mathbb{E}[\hat{U} \mid Z_{1:n}] = U. \tag{158}$$

Moreover, the variance of an incomplete U-statistic can be decomposed by the law of total variance, as:

$$\text{Var}(\hat{U}) = \text{Var}(\mathbb{E}[\hat{U} \mid Z_{1:n}]) + \mathbb{E}[\text{Var}(\hat{U} \mid Z_{1:n})] = \text{Var}(U) + \mathbb{E}[\text{Var}(\hat{U} \mid Z_{1:n})]. \tag{159}$$

Note that, conditional on $Z_{1:n}$, the incomplete statistic $\hat{U}$ is the mean of i.i.d. samples of $h(Z_{S_b})$, where $S$ is a random subset of the original $n$ samples of size $k$. Thus:

$$\text{Var}(\hat{U} \mid Z_{1:n}) = \frac{1}{B}\text{Var}(h(Z_S) \mid Z_{1:n}). \tag{160}$$

Combining 159 and 160,

$$\text{Var}(\hat{U}) = \text{Var}(U) + \frac{1}{B}\mathbb{E}[\text{Var}(h(Z_S) \mid Z_{1:n})]. \tag{161}$$

Moreover, note that the un-conditional variance of the first $k$ samples, is equal to the unconditional variance of a random $k$ subset of the $n$ samples, since we can equivalently think of drawing $k$ i.i.d. samples, by first drawing $n$ samples and then choosing a random subset of them.

$$\text{Var}(h(Z_{1:k})) = \text{Var}(h(Z_S)) = \text{Var}(\mathbb{E}[h(Z_S) \mid Z_{1:n}]) + \mathbb{E}[\text{Var}(h(Z_S) \mid Z_{1:n})]. \tag{162}$$

Moreover, note that by the definition of the random sample $S$ and the U-statistic:

$$\mathbb{E}[h(Z_S) \mid Z_{1:n}] = U \implies \text{Var}(\mathbb{E}[h(Z_S) \mid Z_{1:n}]) = \text{Var}(U). \tag{163}$$

Thus we get:

$$\text{Var}(\hat{U}) = \text{Var}(U) + \frac{1}{B}\text{Var}(h(Z_{1:k})) - \frac{1}{B}\text{Var}(U) = \left(1 - \frac{1}{B}\right)\text{Var}(U) + \frac{1}{B}\eta_k. \tag{164}$$

$\square$

## F.2  Variance of IS procedure

Observe that for the full $n$ sample size, the importance sampling based estimate of MSE is

$$\text{MSE}_n^{\text{IS}} = \mathop{\mathbb{E}}_{D_n \sim \{\pi_i\}_{i=1}^n} \left[ \left( \prod_{i=1}^n \frac{\Pr(D_n; \pi)}{\Pr(D_n; \pi_i)} \right) \text{MSE}(\theta(D_n)) \right] \stackrel{(a)}{=} \mathop{\mathbb{E}}_{D_n \sim \{\pi_i\}_{i=1}^n} \left[ \left( \prod_{i=1}^n \frac{\pi(Z_i|X_i)}{\pi_i(Z_i|X_i)} \right) \text{MSE}(\theta(D_n)) \right] \tag{165}$$

where (a) follows from expanding $\Pr(D_n; \pi)$ and $\Pr(D_n; \pi_i)$ using 6. Let $\text{MSE}_k^{\text{IS}}$ be the estimate of MSE for a subset dataset of size $k$, obtained using importance sampling and a complete enumeration over all subsets of $n$ of size $k$. Similarly, let $\widehat{\text{MSE}}_k^{\text{IS}}$ denote it's computationally friendly approximation that draws $B$ subsets of size $k$ uniformly at random.

Moreover, for simplicity, we will assume here, that given a sub-sample of size $k$ we can exactly calculate the MSE of the estimate that is produced from that sub-sample. Since our goal is to showcase the poor behavior of the IS approach, it suffices to showcase such a behavior in this ideal scenario.

**Theorem 7.** *If $\|\text{MSE}(\theta(D_k))\|_{L^2} = \alpha(k)$ then*

$$\text{Var}(\text{MSE}_k^{\text{IS}}) \le \alpha(k)^2 \binom{n}{k}^{-1} \sum_{c=1}^k \binom{k}{c}\binom{n-k}{k-c}\rho_{\max}^c \tag{166}$$

$$\le \alpha(k)^2 \left( \frac{k^2}{n}\rho_{\max} + \frac{k^2}{n^2}\rho_{\max}^k \right). \tag{167}$$

$$\text{Var}(\widehat{\text{MSE}}_k^{\text{IS}}) \le \alpha(k)^2 \left( \frac{k^2}{n}\rho_{\max} + \frac{k^2}{n^2}\rho_{\max}^k + \frac{1}{B}\rho_{\max}^k \right). \tag{168}$$

*Moreover, for any deterministic data generating process without covariates, and for any deterministic policy $\pi$, the variance can be exactly characterized as:*

$$\mathrm{Var}(\mathrm{MSE}_k^{\mathcal{IS}}) = \alpha(k)^2 \binom{n}{k}^{-1} \sum_{c=1}^{k} \binom{k}{c} \binom{n-k}{k-c} (\rho_{\max}^c - 1). \tag{169}$$

$$\mathrm{Var}(\widehat{\mathrm{MSE}}_k^{\mathcal{IS}}) = \left(1 - \frac{1}{B}\right) \mathrm{Var}(\mathrm{MSE}_k^{\mathcal{IS}}) + \alpha(k)^2 \frac{1}{B} \left(\rho_{\max}^k - 1\right). \tag{170}$$

Even though the lower bound formula for the complete U-statistic is hard to parse in terms of a rate, we see that for many reasonable values of $n$ and $k$, its dependence in $\rho_{\max}$ is exponential (see Figure 7). Moreover, it is obvious that for any finite $B$ (which would be the typical application), the dependence in $\rho_{\max}$ is exponential in $k$, due to the extra variance stemming from the random sampling of subsets.

*Proof.* Let $n$ be the total number of samples. Let $k = o(n)$ be the desired number of samples. To consider analysis with all possible combination $C(n, k)$, we will construct a U-estimator. Let the $h$ be the kernel for the importance weighted base estimator, and $D_k = \{S_i\}_{i=1}^k$,

$$h(\{S_i\}_{i=1}^k) = \rho_{1:k} \mathrm{MSE}(\theta(D_k)), \tag{171}$$

$$\rho_{1:k} = \prod_{i=1}^{k} \rho(S_i). \tag{172}$$

The U-statistic corresponding to this kernel is

$$U = \frac{k!(n-k)!}{n!} \sum_{i_1 < i_2 < \ldots < i_k} h\big(\{S_{i_j}\}_{j=1}^k\big). \tag{173}$$

For this kernel, we can derive that:

$$\mathbb{E}[h(Z) \mid Z_{1:c}] = \mathbb{E}\left[\rho_{1:k} \mathrm{MSE}(\theta(D_k)) \mid S_{1:c}\right] \tag{174}$$

$$= \mathbb{E}\left[\rho_{c+1:k} \mathrm{MSE}(\theta(D_k)) \mid S_{1:c}\right] \rho_{1:c} \tag{175}$$

$$= \mathbb{E}_\pi \left[\mathrm{MSE}(\theta(D_k)) \mid S_{1:c}\right] \rho_{1:c}. \tag{176}$$

Thus we have that:

$$\eta_c = \mathrm{Var}\left(\mathbb{E}[h(Z)|Z_{1:c}]\right) \tag{177}$$

$$= \mathbb{E}\left[\mathbb{E}_\pi \left[\mathrm{MSE}(\theta(D_k)) \mid S_{1:c}\right]^2 \rho_{1:c}^2\right] - \mathbb{E}_\pi[\mathrm{MSE}(\theta(D_k))]^2 \tag{178}$$

$$= \mathbb{E}_\pi\left[\mathbb{E}_\pi \left[\mathrm{MSE}(\theta(D_k)) \mid S_{1:c}\right]^2 \rho_{1:c}\right] - \mathbb{E}_\pi[\mathrm{MSE}(\theta(D_k))]^2 \tag{179}$$

$$\leq \mathbb{E}_\pi\left[\mathbb{E}_\pi \left[\mathrm{MSE}(\theta(D_k))^2 \mid S_{1:c}\right] \rho_{1:c}\right] - \mathbb{E}_\pi[\mathrm{MSE}(\theta(D_k))]^2. \tag{180}$$

$$\leq \mathbb{E}_\pi\left[\mathbb{E}_\pi \left[\mathrm{MSE}(\theta(D_k))^2 \mid S_{1:c}\right]\right] \rho_{\max}^c - \mathbb{E}_\pi[\mathrm{MSE}(\theta(D_k))]^2. \tag{181}$$

$$= \mathbb{E}_\pi\left[\mathrm{MSE}(\theta(D_k))^2\right] \rho_{\max}^c - \mathbb{E}_\pi[\mathrm{MSE}(\theta(D_k))]^2. \tag{182}$$

**Upper bound**    From this we can derive the upper bound:

$$\eta_c \leq \mathbb{E}_\pi\left[\mathrm{MSE}(\theta(D_k))^2\right] \rho_{\max}^c = \alpha(k)^2 \rho_{\max}^c. \tag{183}$$

Which, by invoking Theorem 6 yields the upper bound:

$$\mathrm{Var}(U) \leq \alpha(k)^2 \binom{n}{k}^{-1} \sum_{c=1}^{k} \binom{k}{c} \binom{n-k}{k-c} \rho_{\max}^c. \tag{184}$$

Moreover, from the looser upper bounds in Theorem 6, we also get that:

$$\mathrm{Var}(U) \leq \alpha(k)^2 \left(\frac{k^2}{n} \rho_{\max} + \frac{k^2}{n^2} \rho_{\max}^k\right). \tag{185}$$

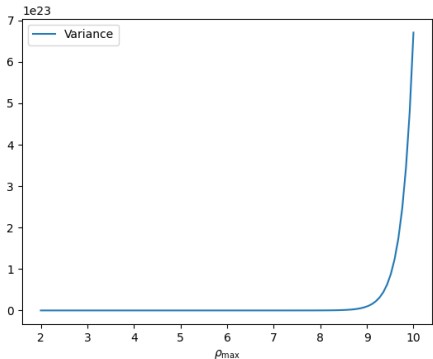

Figure 7: Behavior of complete U-statistic variance for $n = 1000$, $k = 100$ as a function of $\rho_{\max}$.

Then, by invoking Lemma 2, we have that the incomplete version of our estimate with $B$ random sub-samples of size $k$ has:

$$\text{Var}(\hat{U}) = \left(1 - \frac{1}{B}\right) \text{Var}(U) + \frac{1}{B}\eta_k \tag{186}$$

$$\leq \left(1 - \frac{1}{B}\right) \text{Var}(U) + \frac{1}{B}\alpha(k)^2 \rho_{\max}^k \tag{187}$$

$$\leq \alpha(k)^2 \left(\frac{k^2}{n}\rho_{\max} + \frac{k^2}{n^2}\rho_{\max}^k + \frac{1}{B}\rho_{\max}^k\right). \tag{188}$$

where the first inequality comes by upper bounding Equation 182 and using our assumption in the Theorem that $\mathbb{E}_\pi \left[\text{MSE}(\theta(D_k))^2\right] = \alpha(k)^2$, and the second inequality comes from substituting in Equation 185.

**Lower bound**   In general it is hard to characterize the lower bound. However, in the following, we will show that for a certain class of problems, the lower bound on the variance from using the importance sampling procedure can grow exponentially. Specifically, for this theoretical analysis, we consider the case where there are no covariates, all aspects of the data generating process is deterministic, and the policy $\pi$ is also deterministic. Figure 4 (Right) provides an empirical evidence of exponential growth of variance for a more general setting with both stochastic policy and a stochastic data generating process.

Notice that (182) follows with equality when steps (180) and (181) follow with equality. This holds when the entire data generating process is deterministic and the policy $\pi$ is also deterministic. Further, since there are no covariates and that $\pi$ is a deterministic policy, then $\rho_i = \rho_{\max}$ almost surely when $S_i$ is drawn from $\pi$. Therefore,

$$\eta_c = \mathbb{E}_\pi \left[\text{MSE}(\theta(D_k))^2\right] \rho_{\max}^c - \mathbb{E}_\pi \left[\text{MSE}(\theta(D_k)\right]^2 \tag{189}$$

$$= \mathbb{E}_\pi \left[\text{MSE}(\theta(D_k))^2\right] (\rho_{\max}^c - 1) \tag{190}$$

$$= \alpha(k)^2(\rho_{\max}^c - 1), \tag{191}$$

where the last equality holds because $\pi$ is deterministic, and by our assumption in the Theorem, $\mathbb{E}_\pi \left[\text{MSE}(\theta(D_k))^2\right] = \alpha(k)^2$. Moreover, from Theorem 6, the variance of the U-statistic is:

$$\text{Var}(U) = \alpha(k)^2 \binom{n}{k}^{-1} \sum_{c=1}^k \binom{k}{c}\binom{n-k}{k-c}(\rho_{\max}^c - 1). \tag{192}$$

Similarly, using Lemma 2, the variance of the incomplete U-statistic is:

$$\text{Var}(\hat{U}) = \left(1 - \frac{1}{B}\right)\text{Var}(U) + \frac{1}{B}\eta_k \tag{193}$$

$$= \left(1 - \frac{1}{B}\right)\text{Var}(U) + \alpha(k)^2\frac{1}{B}\left(\rho_{\max}^k - 1\right), \tag{194}$$

where in the second equality we have used Equation 191. $\qquad\square$

### F.3 GUARANTEES FOR REJECTION SAMPLING PROCEDURE

We will analyze the bias and variance of the estimate of the mean squared error of the rejection sampling procedure. Almost identical analysis also applies to the analysis of the bias and variance of policy gradient estimates based on the rejection sampling procedure, but we omit it for conciseness.

Let $n$ be the total number of available samples. Let $\mathrm{MSE}_k^{\mathrm{RS}}$ be the estimate of MSE for a subset dataset of size $k$, obtained using rejection sampling. This estimates first accepts a random subset of size $N'$ and if $N' < k$ returns 0, otherwise if $N' \geq k$, returns the estimate that goes over all subsets of size $k$ of the $N'$ accepted samples:

$$\mathrm{MSE}_k^{\mathrm{RS}} = \frac{1}{c} \sum_{i=1}^{c} \widehat{\mathrm{MSE}}(\theta(\{S_{i_j}\}_{j=1}^k)) \qquad c = \binom{N'}{k} \qquad (195)$$

$$\widehat{\mathrm{MSE}}(\theta(D_k)) = \frac{1}{M} \sum_{i=1}^{M} (f(X_i, A_i; \theta(D_n)) - f(X_i, A_i; \theta(D_k)))^2, \qquad (196)$$

where $\widehat{\mathrm{MSE}}(\theta(D_k))$ is the estimate of the mean-squared error produced by the samples in $D_k$, (same as 14) and where $\{X_i, A_i\}_{i=1}^M$ are held-out samples of covariates and treatments (outcomes for these are not available), which is not used in the estimation of $\theta$, i.e. is separate from $D_n$. We will assume throughout that $M >> n$.

We will also consider an alternative estimate $\widehat{\mathrm{MSE}}_k^{\mathrm{RS}}$ that is the computationally friendly approximation that draws $B$ subsets of size $k$ uniformly at random from the $N'$ accepted samples, instead of performing complete enumeration over subsets of size $k$ done for $\mathrm{MSE}_k^{\mathrm{RS}}$.

**Theorem 8** (Consistency of Rejection Sampling). *Consider the rejection sampling estimate $\mathrm{MSE}_k^{RS}$ as defined in Equation (195) and its incomplete U-statistic variant $\widehat{\mathrm{MSE}}_k^{RS}$. For any dataset $D_n$ of iid samples of si'ze $n$ drawn from policy $\pi$, define:*

$$\alpha(n) := \|\mathrm{MSE}(\theta(D_n))\|_{L^2} \qquad (197)$$

*Assume that $k \leq n/2$ and $n$ is large enough, such that $n/\log(n) \geq 8\rho_{\max}$. Then:*

$$\mathrm{Var}\big(\mathrm{MSE}_k^{RS}\big) \leq O\left(\alpha(k)^2 \frac{k\rho_{\max}}{n} + \frac{1}{M} + \alpha(n)\right), \qquad (198)$$

$$\mathrm{BIAS}\big(\mathrm{MSE}_k^{RS}\big) \leq O\left(\alpha(k)e^{-n/8\rho_{\max}} + \frac{1}{\sqrt{M}} + \sqrt{\alpha(n)}\right), \qquad (199)$$

*and similarly for the incomplete variant:*

$$\mathrm{Var}\left(\widehat{\mathrm{MSE}}_k^{RS}\right) \leq O\left(\alpha(k)^2 \frac{k\rho_{\max}}{n} + \alpha(k)^2 \frac{1}{B} + \frac{1}{M} + \alpha(n)\right), \qquad (200)$$

$$\mathrm{BIAS}\left(\widehat{\mathrm{MSE}}_k^{RS}\right) \leq O\left(\alpha(k)e^{-n/8\rho_{\max}} + \frac{1}{\sqrt{M}} + \sqrt{\alpha(n)}\right). \qquad (201)$$

*Moreover, the probability of failure, i.e. $\Pr(N' < k)$, is bounded as: $e^{-n/(8\rho_{\max})}$.*

*Proof.* Let $n$ be the total number of samples and $N'$ be the number of of samples accepted by the rejection sampler, which in expectation equals to $n' := \mathbb{E}[N'] = n/\rho_{\max}$. Like before, let $k$ be the desired number of samples. Let $D_k = \{S_{1:k}\}$.

We first consider the difference between $\mathrm{MSE}_k^{\mathrm{RS}}$ from 195 and its ideal analogue (note that this ideal version is biased as it returns 0 when $N' < k$),

$$U = \frac{1}{c} \sum_{i=1}^{c} \mathrm{MSE}(\theta(\{S_{i_j}\}_{j=1}^k)) \qquad c = \binom{N'}{k} \qquad (202)$$

$$\mathrm{MSE}(\theta(D_k)) = \mathbb{E}_{X,A}\left[(f(X, A; \theta_0) - f(X, A; \theta(D_k)))^2\right] \qquad (203)$$

We now break the proof into three parts:

- Part I: Express bias and variance of $\mathrm{MSE}_k^{\mathrm{RS}}$ in terms of bias and variance of $U$.

- Part II: Compute Variance of $U$ (for both complete and incomplete U-statistic).

- Part III: Compute bias of $U$ (for both complete and incomplete U-statistic).

**(Part I) Bias and variance of** $\mathrm{MSE}_k^{\mathrm{RS}}$ **in terms of that of** $U$**:** We will first express the error of $\mathrm{MSE}_k^{\mathrm{RS}}$ in terms of the the convergence rate of $\mathrm{MSE}(\theta(D_n))$, as $\theta(D_n)$ is used as a proxy for $\theta_0$ by $\mathrm{MSE}_k^{\mathrm{RS}}$. Observe that,

$$\mathbb{E}[(\mathrm{MSE}_k^{\mathrm{RS}} - U)^2] \tag{204}$$

$$= \mathbb{E}\left[\left(\mathbb{E}_n[\widehat{\mathrm{MSE}}(\theta(D_k)] - \mathbb{E}_n[\mathrm{MSE}(\theta(D_k))]\right)^2\right] \tag{205}$$

$$= \mathbb{E}\left[\left(\mathbb{E}_n[\widehat{\mathrm{MSE}}(\theta(D_k) - \mathrm{MSE}(\theta(D_k))]\right)^2\right] \tag{206}$$

$$\leq \mathbb{E}\left[\mathbb{E}_n\left[\left(\widehat{\mathrm{MSE}}(\theta(D_k)) - \mathrm{MSE}(\theta(D_k))\right)^2\right]\right] \qquad \text{(Jensen's inequality)} \tag{207}$$

$$= \mathbb{E}\left[\left(\widehat{\mathrm{MSE}}(\theta(D_k)) - \mathrm{MSE}(\theta(D_k))\right)^2\right]. \tag{207}$$

$$\overset{(a)}{=} \mathbb{E}\left[\left(\widehat{\mathrm{MSE}}(\theta(D_k)) \pm \widetilde{\mathrm{MSE}}(\theta(D_k)) - \mathrm{MSE}(\theta(D_k))\right)^2\right] \tag{208}$$

$$\overset{(b)}{\leq} 2\mathbb{E}\left[\left(\widehat{\mathrm{MSE}}(\theta(D_k)) - \widetilde{\mathrm{MSE}}(\theta(D_k))\right)^2\right] + 2\mathbb{E}\left[\left(\widetilde{\mathrm{MSE}}(\theta(D_k)) - \mathrm{MSE}(\theta(D_k))\right)^2\right] \tag{209}$$

where $(a)$ follows from defining an intermediate quantity $\widetilde{\mathrm{MSE}}$, which unlike $\widehat{\mathrm{MSE}}(\theta(D_k))$ in 195 computes the true expectation instead of empirical average,

$$\widetilde{\mathrm{MSE}}(\theta(D_k)) = \mathbb{E}_{X,A}\left[(f(X_i, A_i; \theta(D_n)) - f(X_i, A_i; \theta(D_k)))^2\right], \tag{210}$$

and $(b)$ follows from $(a+b)^2 \leq 2(a^2 + b^2)$. Now we bound the two terms in the RHS of 209.

To bound the first term in the RHS of 209, note that since predictions are uniformly bounded and the $M$ samples are independent of the samples in $D_n$, we have:

$$\mathbb{E}\left[\left(\widehat{\mathrm{MSE}}(\theta(D_k)) - \widetilde{\mathrm{MSE}}(\theta(D_k))\right)^2\right] \leq \frac{C}{M}, \tag{211}$$

for some constant $C$, since $\widetilde{\mathrm{MSE}}(\theta(D_k))$ is the mean of $\widehat{\mathrm{MSE}}(\theta(D_k))$, conditional on $D_n$ and $\widehat{\mathrm{MSE}}(\theta(D_k))$ is the sum of $M$ iid variables, conditional on $D_n$.

To bound the second term in the RHS of 209, we have that:

$$\mathbb{E}\left[\left(\mathrm{MSE}(\theta(D_k)) - \widetilde{\mathrm{MSE}}(\theta(D_k))\right)^2\right] \tag{212}$$

$$= \mathbb{E}\left[\mathbb{E}_{X,A}\left[(f(X, A; \theta_0) - f(X, A; \theta(D_k)))^2 - (f(X, A; \theta(D_n)) - f(X, A; \theta(D_k)))^2\right]^2\right]$$

$$\overset{(a)}{\leq} \mathbb{E}\left[\mathbb{E}_{X,A}[(f(X, A; \theta_0) - f(X, A; \theta(D_n)))]^2\right] \tag{213}$$

$$\leq \mathbb{E}\left[\mathbb{E}_{X,A}\left[(f(X, A; \theta_0) - f(X, A; \theta(D_n)))^2\right]\right] \qquad \text{(Jensen's inequality)}$$

$$= \mathbb{E}\left[\mathrm{MSE}(\theta(D_n))\right] \leq \sqrt{\mathbb{E}\left[\mathrm{MSE}(\theta(D_n))^2\right]} = \alpha(n), \tag{214}$$

where (a) follows as $(a-b)^2 - (c-b)^2 \leq (a-c)^2$. Thus, combining 209, 211, and 214:

$$\mathbb{E}[(\mathrm{MSE}_k^{\mathrm{RS}} - U)^2] \leq \frac{2C}{M} + 2\alpha(n). \tag{215}$$

Therefore, using 215, bias and variance of $\mathrm{MSE}_k^{\mathrm{RS}}$ can be expressed using the bias and variance of the idealized estimate $U$ as the following,

$$\mathrm{Var}(\mathrm{MSE}_k^{\mathrm{RS}}) = \mathrm{Var}(\mathrm{MSE}_k^{\mathrm{RS}} + U - U) \tag{216}$$

$$\leq 2\mathrm{Var}(U) + 2\mathrm{Var}(\mathrm{MSE}_k^{\mathrm{RS}} - U) \quad \because \mathrm{Var}(A+B) \leq 2(\mathrm{Var}(A) + \mathrm{Var}(B))$$

$$\leq 2\mathrm{Var}(U) + 2\mathbb{E}[(\mathrm{MSE}_k^{\mathrm{RS}} - U)^2] \tag{217}$$

$$\leq 2\mathrm{Var}(U) + \frac{4C}{M} + 4\alpha(n). \tag{218}$$

$$\mathrm{Bias}(\mathrm{MSE}_k^{\mathrm{RS}}) = |\mathbb{E}[\mathrm{MSE}_k^{\mathrm{RS}}] - \mathbb{E}[\mathrm{MSE}(\theta(D_k))]| \tag{219}$$

$$= |\mathbb{E}[\mathrm{MSE}_k^{\mathrm{RS}}] \pm \mathbb{E}[U] - \mathbb{E}[\mathrm{MSE}(\theta(D_k))]| \tag{220}$$

$$\leq |\mathbb{E}[U] - \mathbb{E}[\mathrm{MSE}(\theta(D_k))]| + \sqrt{\mathbb{E}[(\mathrm{MSE}_k^{\mathrm{RS}} - U)^2]} \tag{221}$$

$$\leq \mathrm{Bias}(U) + \sqrt{\frac{2C}{M} + 2\alpha(n)}. \tag{222}$$

Thus it suffices to analyze the bias and variance of the idealized estimate $U$.

**(Part II) Analysis of variance of $U$:** Note that the idealized estimate $U$ returns 0 if $N' < k$ and otherwise returns a U-statistic (or an incomplete version of it):

$$U_{N'} = \frac{1}{c}\sum_{i=1}^{c} h\big(\{S_{i_j}\}_{j=1}^k\big) \qquad\qquad c = \binom{N'}{k} \tag{223}$$

So if we let $U$ be our estimate of the MSE on $k$ samples, we have:

$$U = U_{N'}1\{N' \geq k\}. \tag{224}$$

**Variance of complete U-statistic**  First we note that:

$$\mathrm{Var}(U) = \mathrm{Var}(\mathbb{E}[U \mid N']) + \mathbb{E}[\mathrm{Var}(U \mid N')] \tag{225}$$

The latter term, i.e. $\mathrm{Var}(U \mid N')$ is the variance of a U-statistic of order $k$ with $N'$ samples drawn iid from distribution $\pi$. Thus based on Theorem 6, we have that for $N' \geq k$:

$$\mathrm{Var}(U \mid N') \leq \frac{k^2}{N'}\eta_1 + \frac{k^2}{(N')^2}\eta_k \tag{226}$$

$$\leq \frac{k}{N'}\eta_k + \frac{k^2}{(N')^2}\eta_k \qquad\qquad (\because \eta_1 \leq \tfrac{1}{k}\eta_k)$$

$$\leq 2\frac{k}{N'}\mathrm{Var}_\pi(h(S_{1:k})) \qquad\qquad (k/N' \leq 1)$$

$$\leq 2\frac{k}{N'}\mathbb{E}_\pi[\mathrm{MSE}(\theta(D_k))^2] = 2\frac{k}{N'}\alpha(k)^2. \tag{227}$$

Moreover, for $N' < k$, we have by the definition of $U$ that $\mathrm{Var}(U \mid N') = 0$. Thus we get that:

$$\mathbb{E}[\mathrm{Var}(U \mid N')] = 2k\alpha(k)^2\mathbb{E}\left[\frac{1_{\{N' \geq k\}}}{N'}\right] \leq 2k\alpha(k)^2\mathbb{E}\left[\frac{1}{\max\{N',k\}}\right] \tag{228}$$

Note that $N'$ is a binomially distributed random variable, $\mathrm{Bin}(n, 1/\rho_{\max})$. Thus by a Chernoff bound, and reminding that $n' = \mathbb{E}[N'] = n/\rho_{\max}$, we have that:

$$\Pr(N' \leq (1-\delta)n') \leq e^{-\delta^2 n'/2}. \tag{229}$$

Therefore to upperbound $\mathbb{E}\left[\frac{1}{\max\{N',k\}}\right]$, we split it into the following two cases

$$\frac{1}{\max\{N',k\}} \leq \begin{cases} \frac{1}{(1-\delta)n'} & \text{when } N' > (1-\delta)n' \\ \frac{1}{k} & \text{otherwise} \end{cases} \tag{230}$$

where the first case holds because irrespective of the value of $N'$, the value of $1/\max\{N', k\} \leq 1/N'$. Further, when $N' > (1-\delta)n'$, then $1/N' \leq 1/(1-\delta)n'$. Thus we get that:

$$\mathbb{E}[\mathrm{Var}(U \mid N')] \leq 2k\alpha(k)^2 \left( \frac{\Pr(N' > (1-\delta)n')}{(1-\delta)n'} + \frac{\Pr(N' \leq (1-\delta)n')}{k} \right) \tag{231}$$

$$\leq 2k\alpha(k)^2 \left( \frac{1}{(1-\delta)n'} + \frac{1}{k}e^{-\delta^2 n'/2} \right) \tag{232}$$

$$= 2\alpha(k)^2 \left( \frac{k}{(1-\delta)n'} + e^{-\delta^2 n'/2} \right) \tag{233}$$

$$= 2\alpha(k)^2 \left( \frac{k\rho_{\max}}{(1-\delta)n} + e^{-\delta^2 n/(2\rho_{\max})} \right) \tag{234}$$

Choosing $\delta = \sqrt{\frac{2\rho_{\max}\log(n)}{n}}$, we get:

$$\mathbb{E}[\mathrm{Var}(U \mid N')] \leq 2\alpha(k)^2 \left( \frac{k\rho_{\max}}{(1-\delta)n} + \frac{1}{n} \right) \leq 4\alpha(k)^2 \frac{k\rho_{\max}}{(1-\delta)n} \tag{235}$$

For $n$ large enough such that $n/\log(n) \geq 8\rho_{\max}$, we get $\delta \leq 1/2$ and:

$$\mathbb{E}[\mathrm{Var}(U \mid N')] \leq 8\alpha(k)^2 \frac{k\rho_{\max}}{n} \tag{236}$$

We now consider the first part in the variance decomposition in Equation (225). Note that for $N' \geq k$:

$$\mathbb{E}[U \mid N'] = \mathbb{E}[h(Z_{1:k})] = \mathbb{E}_\pi[\mathrm{MSE}(\theta(D_k))] \tag{237}$$

While for $N' < k$, we have $\mathbb{E}[U \mid N'] = 0$. Thus we have:

$$\mathrm{Var}(\mathbb{E}[U \mid N']) = \mathrm{Var}(\mathbb{E}_\pi[\mathrm{MSE}(\theta(D_k))]1\{N' \geq k\}) \tag{238}$$

$$= \mathbb{E}_\pi[\mathrm{MSE}(\theta(D_k))]^2\mathrm{Var}(1\{N' \geq k\}) \tag{239}$$

$$\leq \mathbb{E}_\pi[\mathrm{MSE}(\theta(D_k))^2]\mathrm{Var}(1\{N' \geq k\}) \tag{240}$$

$$\leq \alpha(k)^2\mathrm{Var}(1\{N' \geq k\}) \tag{241}$$

$$= \alpha(k)^2\Pr(N' \geq k)(1 - \Pr(N' \geq k)) \tag{242}$$

$$\leq \alpha(k)^2\Pr(N' < k). \tag{243}$$

If we let $\delta$, be such that $k = (1-\delta)n$, then we have by a Chernoff bound

$$\Pr(N' < k) \leq e^{-\delta^2 n'/2} = e^{-\delta^2 n/(2\rho_{\max})} = e^{-(1-k/n)^2 n/(2\rho_{\max})}$$

Assuming that $k \leq n/2$, we have:

$$\Pr(N' < k) \leq e^{-n/(8\rho_{\max})}. \tag{244}$$

Since, we assume that $n/\log(n) \geq 8\rho_{\max}$, we have:

$$\Pr(N' < k) \leq \frac{1}{n}. \tag{245}$$

Thus overall we have derived that:

$$\mathrm{Var}(U) \leq 9\alpha(k)^2 \frac{k\rho_{\max}}{n}. \tag{246}$$

**Variance of incomplete U-statistic** With an identical analysis, if we instead use an incomplete U-statistic with $B$ random sub-samples (Lemma 2), then we can apply the exact same analysis, albeit, when $N' \geq k$:

$$\mathrm{Var}(U \mid N') \leq 2\frac{k}{N'}\alpha(k)^2 + \frac{1}{B}\mathrm{Var}_\pi(h(S_{1:k})) \tag{247}$$

$$\leq 2\frac{k}{N'}\alpha(k)^2 + \frac{1}{B}\alpha(k)^2. \tag{248}$$

Following then an identical analysis, we can derive:

$$\text{Var}(U) \leq \alpha(k)^2 \left( \frac{9\,k\rho_{\max}}{n} + \frac{1}{B} \right). \tag{249}$$

**(Part III) Analysis of bias of** $U$**:**   Finally, we also quantify the bias of the rejection sampling estimate. Note that when $N' \geq k$, we have:

$$\mathbb{E}[U \mid N'] = \mathbb{E}[h(S_{1:k})] = \mathbb{E}_\pi[\text{MSE}(\theta(D_k))] \tag{250}$$

and the estimate is un-biased. However, when $N' < k$, then we output $0$, hence we incur a bias of:

$$|\mathbb{E}[U \mid N'] - \mathbb{E}_\pi[\text{MSE}(\theta(D_k))]| = \mathbb{E}_\pi[\text{MSE}(\theta(D_k))] \leq \alpha(k). \tag{251}$$

Thus overall we have a bias of:

$$|\mathbb{E}[U] - \mathbb{E}_\pi[\text{MSE}(\theta(D_k))]| \leq |\mathbb{E}[U \mid N'] - \mathbb{E}_\pi[\text{MSE}(\theta(D_k))]| \Pr(N' < k) \tag{252}$$
$$\leq \alpha(k)e^{-n/(8\rho_{\max})} \tag{253}$$

The bias of the incomplete U-statistic variant is identical, since both variants are un-biased when $N' \geq k$.

Combining the aforementioned bounds on the idealized estimate $U$ with our error bounds between $U$ and $\text{MSE}_k^{\text{RS}}$, we get the result.  $\square$

### F.4   POLICY ORDERING CONSISTENCY GUARANTEES

Let:

$$\mathcal{L}_\pi(k) = \mathbb{E}_\pi[\text{MSE}(\theta(D_k))], \qquad \alpha_\pi(k) = \sqrt{\mathbb{E}_\pi\left[\text{MSE}(\theta(D_k))^2\right]}. \tag{254}$$

Suppose that we assume that the mean-squared-error, irrespective of sampling distribution, satisfies that:

$$r_n\mathcal{L}_\pi(n) \to V_\pi, \qquad r_n\alpha_\pi(n) = \Theta(1), \tag{255}$$

for some random variable $V_\pi$ and some rate $r_n$ (e.g. $r_n = \sqrt{n}$). Then we can consider the normalized estimate:

$$r_k\widehat{\text{MSE}}_k^{\text{RS}}. \tag{256}$$

Suppose we choose $M >> \frac{n}{k\alpha(k)^2}$ and $B >> \frac{n}{k}$ and we choose $k$ such that $r_k^2/r_n \to 0$ and $k/n \to 0$. Then, from Theorem 8, we have that the normalized estimate of the mean-squared-error from 256 has:

$$\text{Var}(r_k\widehat{\text{MSE}}_k^{\text{RS}}) = O\left( r_k^2\alpha(k)^2\frac{k}{n} + r_k^2\alpha(n) \right) = O\left( \frac{k}{n} + r_k^2/r_n \right) \to 0 \tag{257}$$

$$r_k\text{BIAS}\left(\widehat{\text{MSE}}_k^{\text{RS}}\right) \leq O\left( r_k\alpha(k)e^{-n/8\rho_{\max}} + r_k\sqrt{\alpha(n)} \right) = O\left( e^{-n/8\rho_{\max}} + r_k/\sqrt{r_n} \right) \to 0 \tag{258}$$

Thus we can conclude that the normalized mean squared error of the estimate will satisfy:

$$r_k^2\mathbb{E}[(\widehat{\text{MSE}}_k^{\text{RS}} - \mathcal{L}_\pi(k))^2] \to 0 \tag{259}$$

Thus we get that:

$$r_k\widehat{\text{MSE}}_k^{\text{RS}} = r_k\mathcal{L}_\pi(k) + r_k(\widehat{\text{MSE}}_k^{\text{RS}} - \mathcal{L}_\pi(k)) = V_\pi + o_p(1). \tag{260}$$

From this we conclude that optimizing $r_k\widehat{\text{MSE}}_k^{\text{RS}}$ over policies $\pi$ approximately optimizes $V_\pi$, up to an asymptotically neglibible error. Thus for a large enough sample $n$, we should expect that we are choosing an approximately optimal policy $\pi$.

**Corollary 1** (Approximate Policy Ordering). *For any dataset $D_n$ of iid samples of size $n$ drawn from policy $\pi$, define:*

$$\mathcal{L}_\pi(k) = \mathbb{E}_\pi\left[\mathrm{MSE}(\theta(D_k))\right] \qquad \alpha_\pi(k) = \sqrt{\mathbb{E}_\pi\left[\mathrm{MSE}(\theta(D_k))^2\right]} \qquad (261)$$

*Suppose that for some appropriately defined growth rate $r_n$, we have that:*

$$|r_n\mathcal{L}_\pi(n) - V_\pi| \le \epsilon_n \qquad\qquad r_n\alpha_\pi(n) = O(1) \qquad (262)$$

*for some rate $r_n$ that is policy independent and for some approximation error $\epsilon_n \to 0$. Let $\widehat{\mathrm{MSE}}_k^{RS}(\pi)$ denote the rejection sampling estimate of the mean squared error of $\pi$. Suppose that we choose $k$ such that $r_k^2/r_n \to 0$ and $k/n \to 0$ and $k \to \infty$ and $M, B$ are large enough. Then for any fixed policy $\pi$:*

$$\mathbb{E}[(r_k\widehat{\mathrm{MSE}}_k^{RS}(\pi) - V_\pi)^2] = \mathbb{E}[(r_k\widehat{\mathrm{MSE}}_k^{RS}(\pi) \pm r_k\mathcal{L}_\pi(k) - V_\pi)^2] \qquad (263)$$

$$\le 2\epsilon_k^2 + 2r_k^2\mathbb{E}[(\widehat{\mathrm{MSE}}_k^{RS} - \mathcal{L}_\pi(k))^2] \qquad (264)$$

$$\overset{(a)}{\le} O\left(\epsilon_k^2 + \frac{k}{n} + \frac{r_k^2}{r_n} + e^{-n/(8\rho_{\max})}\right) =: \gamma(n,k) \to 0, \qquad (265)$$

*where (a) follows from using the bias and variance terms from Theorem 8 to expand the mean-squared-error $\mathbb{E}[(\widehat{\mathrm{MSE}}_k^{RS} - \mathcal{L}_\pi(k))^2]$.*

*Let $\pi_1, \pi_2$ be two candidate policies. if we choose $\pi_2$ over $\pi_1$ based on $\hat{V}_\pi$ (i.e., $\hat{V}_{\pi_1} < \hat{V}_{\pi_2}$), By our assumptions on the convergence of $r_n\mathcal{L}_\pi(n)$, we have:*

$$r_n(\mathcal{L}_{\pi_1}(n) - \mathcal{L}_{\pi_2}(n)) \le V_{\pi_1} - V_{\pi_2} + 2\epsilon_n \qquad (266)$$

$$\le \hat{V}_{\pi_1} - \hat{V}_{\pi_2} + 2\left(\epsilon_n + \max_{\pi \in \{\pi_1,\pi_2\}} |V_\pi - \hat{V}_\pi|\right) \qquad (267)$$

$$\le 2\left(\epsilon_n + \max_{\pi \in \{\pi_1,\pi_2\}} |V_\pi - \hat{V}_\pi|\right) \qquad (268)$$

*By Markov's inequality and a union bound we have w.p. $1 - \delta$:*

$$|V_\pi - \hat{V}_\pi| \le \frac{2}{\delta}\mathbb{E}[|V_\pi - \hat{V}_\pi|] \le \frac{2}{\delta}\sqrt{\mathbb{E}[|V_\pi - \hat{V}_\pi|^2]} \le \frac{2}{\delta}\sqrt{\gamma(n,k)} \qquad (269)$$

*for $\pi \in \{\pi_1, \pi_2\}$. Thus overall we get w.p. $1 - \delta$:*

$$r_n(\mathcal{L}_{\pi_1}(n) - \mathcal{L}_{\pi_2}(n)) \le O\left(\epsilon_n + \frac{1}{\delta}\sqrt{\gamma(n,k)}\right). \qquad (270)$$

*Thus the policy that is best according to the estimated criterion $\hat{V}_\pi$ is also approximately best according to the normalized expected mean-squared-error with $n$ samples, i.e. $r_n\mathcal{L}_\pi(n)$.*

### F.5 PROOF OF THEOREM 6

For the first part of the Lemma, i.e. proving Equation (153), we first observe that:

$$\mathrm{Var}(U) = \binom{n}{k}^{-2} \sum_{1 \le i_1 < \ldots < i_k \le n} \sum_{1 \le j_1 < \ldots < j_k \le n} \mathrm{Cov}\left(h(Z_{i_1},\ldots,Z_{i_k}), h(Z_{i_1},\ldots,Z_{i_k})\right) \quad (271)$$

Moreover, note that if $|\{i_1,\ldots,i_k\} \cup \{j_1,\ldots,j_k\}| = c$ then

$$\mathrm{Cov}\left(h(Z_{i_1},\ldots,Z_{i_k}), h(Z_{i_1},\ldots,Z_{i_k})\right) = \eta_c,$$

with $\eta_0 = 0$. Furthermore, the number of subsets $\{i_1,\ldots,i_k\}$ and $\{j_1,\ldots,j_k\}$ that have intersection equal to $c$ is $\binom{n}{k}\binom{k}{c}\binom{n-k}{k-c}$. Thus we derive Equation (153):

$$\mathrm{Var}(U) = \binom{n}{k}^{-1} \sum_{c=1}^{k} \binom{k}{c}\binom{n-k}{k-c}\eta_c. \qquad (272)$$

The proof of the second part follows from along the lines of Hoeffding's canonical decomposition and bares resemblance to the asymptotic normality proofs in Fan et al. (2018); Wager & Athey (2018b). Consider the following projection functions:

$$h_1(z_1) = \mathbb{E}[h(z_1, Z_2, \ldots, Z_k)], \qquad \tilde{h}_1(z_1) = h_1(z_1) - \mathbb{E}[h], \qquad (273)$$

$$h_2(z_1, z_2) = \mathbb{E}[h(z_1, z_2, Z_3, \ldots, Z_k)], \qquad \tilde{h}_2(z_1, z_2) = h_2(z_1, z_2) - \mathbb{E}[h], \qquad (274)$$

$$\vdots \qquad (275)$$

$$h_k(z_1, z_2, \ldots, z_k) = \mathbb{E}[h(z_1, z_2, z_3, \ldots, z_k)], \quad \tilde{h}_k(z_1, z_2, \ldots, z_k) = h_k(z_1, z_2, \ldots, z_k) - \mathbb{E}[h], \qquad (276)$$

where $\mathbb{E}[h] = \mathbb{E}[h(Z_1, \ldots, Z_k)]$. Then we define the canonical terms of Hoeffding's $U$-statistic decomposition as:

$$g_1(z_1) = \tilde{h}_1(z_1), \qquad (277)$$

$$g_2(z_1, z_2) = \tilde{h}_2(z_1, z_2) - g_1(z_1) - g_2(z_2), \qquad (278)$$

$$g_3(z_1, z_2, z_3) = \tilde{h}_2(z_1, z_2, z_3) - \sum_{i=1}^{3} g_1(z_i) - \sum_{1 \le i < j \le 3} g_2(z_i, z_j), \qquad (279)$$

$$\vdots \qquad (280)$$

$$g_k(z_1, z_2, \ldots, z_k) = \tilde{h}_k(z_1, z_2, \ldots, z_k) - \sum_{i=1}^{k} g_1(z_i) - \sum_{1 \le i < j \le k} g_2(z_i, z_j) - \ldots \qquad (281)$$

$$\ldots - \sum_{1 \le i_1 < i_2 < \ldots < i_{k-1} \le k} g_{k-1}(z_{i_1}, z_{i_2}, \ldots, z_{i_{k-1}}). \qquad (282)$$

The key property is that all the canonical terms in the latter expression are uncorrelated and mean zero.

Subsequently the kernel of the $U$-statistic can be re-written as a function of the canonical terms:

$$\tilde{h}(z_1, \ldots, z_k) = h(z_1, \ldots, z_k) - \mathbb{E}[h] = \sum_{i=1}^{k} g_1(z_i) + \sum_{1 \le i < j \le k} g_2(z_i, z_j) + \ldots + g_k(z_1, \ldots, z_k).$$

By the uncorrelatedness of the canonical terms we have:

$$\mathrm{Var}[h(Z_1, \ldots, Z_k)] = \binom{k}{1} \mathbb{E}[g_1^2] + \binom{k}{2} \mathbb{E}[g_2^2] + \ldots + \binom{k}{k} \mathbb{E}[g_k^2]. \qquad (283)$$

We can now re-write the $U$ statistic also as a function of canonical terms:

$$U - \mathbb{E}[U] = \binom{n}{k}^{-1} \sum_{1 \le i_1 < i_2 < \ldots < i_k \le n} \tilde{h}(Z_{i_1}, \ldots, Z_{i_k}) \qquad (284)$$

$$= \binom{n}{k}^{-1} \left( \binom{n-1}{k-1} \sum_{i=1}^{n} g_1(Z_i) + \binom{n-2}{k-2} \sum_{1 \le i < j \le n} g_2(Z_i, Z_j) + \ldots \qquad (285) \right.$$

$$\left. + \binom{n-k}{k-k} \sum_{1 \le i_1 < i_2 < \ldots < i_k \le n} g_k(Z_{i_1}, \ldots, Z_{i_k}) \right). \qquad (286)$$

Now we define the Hájek projection to be the leading term in the latter decomposition:

$$U_1 = \binom{n}{k}^{-1} \binom{n-1}{k-1} \sum_{i=1}^{n} g_1(Z_i).$$

The variance of the Hajek projection is:

$$\sigma_1^2 = \mathrm{Var}[U_1] = \frac{k^2}{n} \mathrm{Var}[h_1(z_1)] = \frac{k^2}{n} \eta_1.$$

Moreover, since the canonical terms are un-correlated and mean-zero, we also have:

$$\text{Var}(U) = \text{Var}(U_1) + \text{Var}(U - \mathbb{E}[U] - U_1) = \text{Var}(U_1) + \mathbb{E}\left[(U - \mathbb{E}[U] - U_1)^2\right] \quad (287)$$

From an inequality due to Wager & Athey (2018a):

$$\mathbb{E}\left[(U - \mathbb{E}[U] - U_1)^2\right] = \binom{n}{k}^{-2} \left\{ \binom{n-2}{k-2}^2 \binom{n}{2} \mathbb{E}[g_2^2] + \ldots + \binom{n-k}{k-k}^2 \binom{n}{k} \mathbb{E}[g_k^2] \right\} \quad (288)$$

$$= \sum_{r=2}^{k} \left\{ \binom{n}{k}^{-2} \binom{n-r}{k-r}^2 \binom{n}{r} \mathbb{E}[g_r^2] \right\} \quad (289)$$

$$= \sum_{r=2}^{s} \left\{ \frac{k!(n-r)!}{n!(k-r)!} \binom{k}{r} \mathbb{E}[g_r^2] \right\} \quad (290)$$

$$\leq \frac{k(k-1)}{n(n-1)} \sum_{r=2}^{k} \binom{k}{r} \mathbb{E}[g_r^2] \quad (291)$$

$$\leq \frac{k^2}{n^2} \text{Var}\left[h(Z_1, \ldots, Z_k)\right] \quad (292)$$

$$\leq \frac{k^2}{n^2} \eta_k. \quad (293)$$

From the above we get that:

$$\frac{k^2}{n} \eta_1 = \text{Var}(U_1) \leq \text{Var}(U) \leq \text{Var}(U_1) + \frac{k^2}{n^2} \eta_k = \frac{k^2}{n} \eta_1 + \frac{k^2}{n^2} \eta_k. \quad (294)$$

Moreover, from Equation (283), we have that:

$$\eta_m = \text{Var}(h(Z_1, \ldots, Z_k)) \geq \binom{k}{1} \mathbb{E}[g_1(Z_1)^2] = \binom{k}{1} \text{Var}(h_1(Z_1)) = k\,\eta_1. \quad (295)$$

which completes the proof of the Lemma.

## G  MULTI-REJECTION SAMPLING

### G.1  REVIEW: STANDARD REJECTION SAMPLING

Recall that to simulate draws from a distribution $p$ using samples from another sequence of distribution $\{q_i\}_{i=1}^k$, the standard rejection sampling procedure accepts a sample $S_i$ if $\xi_i \leq p(S_i)/(q_i(S_i)C)$, where $C = \max_{s,i} p(s)/q_i(s)$ and $\xi \sim U$. While it is well-known in the literature, for completeness, we show that the samples drawn using the above rejection sampling procedure will simulate draws from $p(x)$,

$$\Pr(X = x) = \frac{1}{k} \sum_{i=1}^{K} \Pr(\text{Accept } x; q_i) q_i(x) \quad (296)$$

$$= \frac{1}{k} \sum_{i=1}^{k} \Pr\left(U \leq \frac{p(x)}{Cq_i(x)}\right) q_i(x) \quad (297)$$

$$= \frac{1}{k} \sum_{i=1}^{k} \frac{p(x)}{Cq_i(x)} q_i(x) \quad (298)$$

$$= \frac{1}{k} \sum_{i=1}^{k} \frac{p(x)}{C} \quad (299)$$

$$= \frac{p(x)}{C} \quad (300)$$

$$\propto p(x). \quad (301)$$

An instantiation of standard rejection sampling for our setup is provided in 13.

### G.2 PROPOSED: MULTI-REJECTION SAMPLING

Here, we consider discussing the proposed multiple-importance sampling procedure more generally. Consider the case when we have samples from multiple proposal distributions $\{q_i\}_{i=1}^k$, we can either treat each of them independently to sample from $p$, as discussed above. Alternatively, as we show below, we can treat them as a **mixture jointly**, such that even if a sample $X \sim q_i$, we form the acceptance ratio as $\frac{p(X)}{C\frac{1}{k}\sum_j q_j(X)}$, where $C := \max_x p(x)/(\frac{1}{k}\sum_{j=1}^k q_j(x))$. The following provides proof that the proposed procedure draws samples that simulates sampling from the desired distribution $p(x)$ :

$$\Pr(X = x) = \frac{1}{k}\sum_{i=1}^k \Pr(\text{Accept } x; q_i)q_i(x) \tag{302}$$

$$= \frac{1}{k}\sum_{i=1}^k \Pr\left(U \le \frac{p(x)}{C\frac{1}{k}\sum_{j=1}^k q_j(x)}\right)q_i(x) \tag{303}$$

$$= \frac{1}{k}\sum_{i=1}^k \frac{p(x)}{C\frac{1}{k}\sum_{j=1}^k q_j(x)}q_i(x) \tag{304}$$

$$= \frac{p(x)}{C\frac{1}{k}\sum_{j=1}^k q_j(x)}\frac{1}{k}\sum_{i=1}^k q_i(x) \tag{305}$$

$$= \frac{p(x)}{C} \tag{306}$$

$$\propto p(x). \tag{307}$$

An instantiation of multi-rejection sampling for our setup is provided in 15.

**Remark 10.** *Validity of rejection sampling requires the support assumption, that is for $\mathcal{X} := \{x : p(x) > 0\}$ it hods that $q_i(x) > 0, \forall i \in \{1, ..., k\}, \forall x \in \mathcal{X}$. Multiple-rejection sampling replaces the support condition from all $q_i$'s to **at least one** $q_i$. That is, we only require that $\frac{1}{k}\sum_{i=1}^k q_i(x) > 0, \forall x \in \mathcal{X}$. Alternatively, we only require that $\forall x \in \mathcal{X}, \{\exists i \in \{1, ..., k\}, s.t. \ q_i(x) > 0\}$.*

*For instance, if $q_1$ (the first exploration policy) is uniform, then subsequent $q_i$'s can even be deterministic and the data from it can still be used for multiple rejection sampling. This would not have been possible if we had used single-rejection sampling.*

**Remark 11.** *Multiple-rejection sampling can often also reduce the value of the normalizing constant significantly as now*

$$\bar{\rho}_{\max} := \max_x \frac{p(x)}{\frac{1}{k}\sum_{j=1}^k q_j(x)} \tag{308}$$

*which can reduce the failure rate of rejection sampling significantly.*

It can be shown that the expected number of samples accepted using multiple-rejection sampling will always be more than or equal to the number of samples accepted under individual rejection sampling, irrespective of how different are the proposal distributions $\{q_i\}_{i=1}^k$ to the target $p$. To see this, recall that for individual rejection sampling, the expected number of samples accepted from $N$ samples drawn from a proposal $q$ is $N/\rho_{\max}$ as

$$\Pr(\text{Accept } X) = \Pr\left(U \le \frac{p(X)}{\rho_{\max} q(X)}\right) \tag{309}$$

$$= \int \Pr\left(U \le \frac{p(x)}{\rho_{\max} q(x)} \bigg| X = x\right)q(x)\,dx \tag{310}$$

$$= \int \frac{p(x)}{\rho_{\max} q(x)}q(x)\,dx \tag{311}$$

$$= \frac{1}{\rho_{\max}}. \tag{312}$$

Let there be $k$ proposal distributions, with $N$ samples drawn from each distributions. Therefore, the expected number of samples accepted would be

$$\sum_{i=1}^{k} \frac{N}{\rho_{\max}^i} = N \sum_{i=1}^{k} \frac{1}{\rho_{\max}^i} = N \sum_{i=1}^{k} \frac{1}{\max_x \frac{p(x)}{q_i(x)}} = N \sum_{i=1}^{k} \min_x \left( \frac{q_i(x)}{p(x)} \right). \tag{313}$$

Contrast this with the expected number of samples accepted under multiple-rejection sampling, where all the $Nk$ samples are treated equally,

$$\frac{Nk}{\bar{\rho}_{\max}} = \frac{Nk}{\max_x \frac{p(x)}{\frac{1}{k} \sum_{i=1}^{k} q_i(x)}} = Nk \min_x \frac{\frac{1}{k} \sum_{i=1}^{k} q_i(x)}{p(x)} = N \min_x \left( \sum_{i=1}^{k} \frac{q_i(x)}{p(x)} \right). \tag{314}$$

Now,

$$\forall x, \quad \sum_{i=1}^{k} \frac{q_i(x)}{p(x)} \geq \sum_{i=1}^{k} \min_x \left( \frac{q_i(x)}{p(x)} \right) \tag{315}$$

$$\therefore \min_x \left( \sum_{i=1}^{k} \frac{q_i(x)}{p(x)} \right) \geq \sum_{i=1}^{k} \min_x \left( \frac{q_i(x)}{p(x)} \right) \tag{316}$$

$$\therefore \frac{Nk}{\bar{\rho}_{\max}} \geq \sum_{i=1}^{k} \frac{N}{\rho_{\max}^i} \geq \frac{N}{\rho_{\max}^i} \quad \forall i \in \{1, ..., k\}. \tag{317}$$

**Example 1.** consider the case where $N$ samples are drawn from each of the two proposal distributions $q_1$ and $q_2$ over $\{0, 1\}$, such that $q_1(x = 1) = q_2(x = 0) = 0.75$ and $q_1(x = 0) = q_2(x = 1) = 0.25$. Let target distribution be $p(x = 0) = p(x = 1) = 0.5$. In this case, it can be observed that individual rejection sampling will reject $50\%$ of the samples! In contrast, multiple-rejection sampling will accept *all* the samples as the average proposal distribution $(q_1 + q_2)/2 = p$.

**Example 2.** Consider the other extreme case where one of the proposal distribution is exactly the same as the target distribution and the other is completely opposite. We will see that multiple-rejection sampling stays robust against such settings as well.

Let $N$ samples be drawn from each of the two proposal distributions $q_1$ and $q_2$ over $\{0, 1\}$, such that $q_1(x = 1) = q_2(x = 0) \approx 0$ and $q_1(x = 0) = q_2(x = 1) \approx 1$ (here we use '$\approx$' instead of '$=$' to ensure that individual rejection sampling satisfies support assumption (note: multiple rejection sampling will be valid in the case of equality as well)). Let the target distribution be $p = q_1$.

In this case, both individual rejection sampling and multiple-rejection sampling accept $50\%$ of all the samples drawn (i.e., $100\%$ of the samples drawn from $q_1$). This can be worked out by observing that (a) Since $p(x = 1) = 0$, no samples will be selected from the samples drawn from $q_2$. (b) The average proposal distribution is $\bar{q} = [0.5, 0.5]$, therefore $\max_x p(x)/\bar{q}(x) = 2$. Since $q_1(x = 0) = 1$ only 0's are sampled from $q_1$. Probability of accepting zero under multiple rejection sampling is $(p(x = 0)/\bar{q}(x = 0))/(\bar{\rho}_{\max}) = (1/0.5)/2 = 1$.

## H  ALGORITHM

### H.1  PSEUDO-CODE

---

**Algorithm 1:** DIA: Designing Instruments Adaptively

---

1 **Input** Number of allocations $K$, Size of each allocation $n$, Subsampling factor $\alpha$
2 $\pi = \texttt{uniform}$
3 $D = \texttt{GetData}(\pi, n)$                        ▷ Sample $n$ new data using $\pi$
4 **for** $i \in [1, ..., K - 1]$ **do**
5     $\pi = \texttt{Optimize}(D, \alpha, Kn)$
6     $D = \texttt{GetData}(\pi, n)$
7 Return $\theta = \texttt{Estimate}(D)$

---

---

**Algorithm 2:** Optimize

---

1 **Input** Dataset $D$, Sub-sampling factor $\alpha$, Max-budget $N$
2 Initialize $\pi_\varphi$
3 Initialize $\Theta$             ▷ Set of $B$ parameters
4 $n = |D|$
5 $\hat{\theta}_0 = \texttt{Estimate}(D)$            ▷ Compute estimate on the entire data
6 **while** $\pi_\varphi$ *not converged* **do**
7     $\pi^{\text{eff}}(Z|X) \leftarrow \frac{n}{N}\Pr(Z|X;D) + \left(1 - \frac{n}{N}\right)\pi_\varphi(Z|X)$     ▷ Equation 16

     #Parallel For loops
8     **for** $b \in [1, ..., B]$ **do**
9        $D_k = \texttt{Multi-Reject-Sampler}(D, \pi^{\text{eff}}, \alpha)$     ▷ Equation 15, with $k = |D|^\alpha$
10       $\Theta[b] = \texttt{Estimate}(D_k, \text{Init} = \Theta[b])$     ▷ Optional: Init from previous solution
11       $\text{MSE} = (\Theta[b] - \hat{\theta}_0)^2$
12       $\mathcal{I} = \texttt{GetInfluences}(\text{MSE}, \Theta[b], D_k)$     ▷ Equation 12
13       $\nabla^{IF}[b] = \sum_{s \in D_k} \nabla_\varphi \log \pi^{\text{eff}}(s)\mathcal{I}(s)$     ▷ Equation 17
14     $\pi_\varphi \leftarrow \texttt{Update}(\pi_\varphi, \nabla^{IF})$     ▷ Any optimizer: SGD, Adam, etc.
15 **Return** $\pi_\varphi$

---

We provide a pseudo code for the proposed DIA algorithm in 1 and 2. In the following we provide a short description of the functions used there-in.

- `GetData`: Uses the given policy to sample $n$ new data samples

- `Estimate`: Uses 3 to solve for the required nuisances and the parameters of interest.

- `Multi-Reject-Sampler`: Takes as input dataset $D$ of size $n$ and returns a sub-sampled dataset of size $k = n^\alpha$ using the proposed multi-rejection sampling 15. Since the data is collected by the algorithm itself, DIA has access to $\pi$ and also all the prior policies (that are require for the denominator of $\bar{\rho}$) and can directly estimate $\bar{\rho}$. In practice, the max mutli-importance ratio $\bar{\rho}_{\max}$ may not be known. Therefore, we use the empirical supremum (Caffo et al., 2002), where empirical max of multi-importance ratios are used in the place of $\bar{\rho}_{\max}$.

- `GetInfluences`: Uses 12 to return influence of each of the samples $s \in D_k$ on the MSE. Here we discuss how to compute it efficiently using Hessian vector products (Pearlmutter, 1994). Recall, from 11 and 12,

$$\mathcal{I}_{\text{MSE}}^{(1)}(S_i) = \frac{\mathrm{d}}{\mathrm{d}\theta}\text{MSE}(\hat{\theta})\,\mathcal{I}_\theta^{(1)}(S_i) \tag{318}$$

$$\mathcal{I}_\theta^{(1)}(S_i) = -\frac{\partial M_n}{\partial\theta}^{-1}(\hat{\theta}, \hat{\phi})\left[m(S_i, \hat{\theta}, \hat{\phi}) - \frac{\partial M_n}{\partial\phi}(\hat{\theta}, \hat{\phi})\left[\frac{\partial Q_n}{\partial\phi}^{-1}(\hat{\phi})q(S_i, \hat{\phi})\right]\right]. \tag{319}$$

Note that $\frac{\mathrm{d}}{\mathrm{d}\theta}$ can be computed directly as it is just the gradient of the MSE. Below, we focus on $\mathcal{I}_\theta^{(1)}(S_i)$. To begin, Let $\theta \in \mathbb{R}^{d_1}$ and $\phi \in \mathbb{R}^{d_2}$. Let $H = \frac{\partial Q_n}{\partial\phi} \in \mathbb{R}^{d_2 \times d_2}$ and $v = q(S_i, \hat{\phi}) \in \mathbb{R}^{d_2}$, such that we want to compute $H^{-1}v$. Notice that this is a solution to the equation $Hp = v$, therefore we can solve for $\arg\min_{p \in \mathbb{R}^{d_2}} \|Hp - v\|^2$ using any method (we make use of conjugate gradients). Importantly, note that we no longer need to explicitly compute the inverse. Further, to avoid even explicitly computing or storing $H$, we use the 'stop-gradient' operator $\texttt{sg}$ in auto-diff libraries and express

$$Hp = \frac{\partial Q_n}{\partial\phi}p = \frac{\partial}{\partial\phi}\langle Q, \texttt{sg}(p)\rangle, \qquad\qquad \in \mathbb{R}^{d_2} \tag{320}$$

which reduces to merely taking a derivative of the scalar $\langle Q, \texttt{sg}(p)\rangle \in \mathbb{R}$. Thus using hessian-vector product and conjugate gradients we can compute $H^{-1}v$, even for arbitrary neural networks with $d_2$ parameters. Similarly, we can compute all the terms in $\mathcal{I}_\theta^{(1)}(S_i)$, by repeated use of Hessian-vector products. Other tricks can be used to scale up this computation to millions (Lorraine et al., 2020) and billions of hyper-parameters (Grosse et al., 2023).

Finally, we also observe that the higher order influence $\mathcal{I}_\theta^{(2)}(S_i)$ can be partially estimated using the first order $\mathcal{I}_\theta^{(1)}(S_i)$ terms. Recall from 68

$$\mathcal{I}_{\mathrm{MSE}}^{(1)}(S_i) = \frac{\mathrm{d}}{\mathrm{d}\theta}\mathrm{MSE}\Big(\hat{\theta}_{i,\epsilon,\hat{\phi}_{i,\epsilon}}\Big)\mathcal{I}_\theta^{(1)}(S_i)\bigg|_{\epsilon=0} \quad \text{where,} \quad \mathcal{I}_\theta^{(1)}(S_i) = \frac{\mathrm{d}\hat{\theta}_{i,\epsilon,\hat{\phi}_{i,\epsilon}}}{\mathrm{d}\epsilon}\bigg|_{\epsilon=0}. \tag{321}$$

Therefore,

$$\mathcal{I}_{\mathrm{MSE}}^{(2)}(S_i) = \frac{\mathrm{d}}{\mathrm{d}\epsilon}\left(\frac{\mathrm{d}}{\mathrm{d}\theta}\mathrm{MSE}\Big(\hat{\theta}_{i,\epsilon,\hat{\phi}_{i,\epsilon}}\Big)\mathcal{I}_\theta^{(1)}(S_i)\right)\bigg|_{\epsilon=0} \tag{322}$$

$$= \mathcal{I}_\theta^{(1)}(S_i)^\top \frac{\mathrm{d}^2}{\mathrm{d}^2\theta}\mathrm{MSE}\Big(\hat{\theta}_{i,\epsilon,\hat{\phi}_{i,\epsilon}}\Big)\mathcal{I}_\theta^{(1)}(S_i)\bigg|_{\epsilon=0} \tag{323}$$

$$+ \underbrace{\frac{\mathrm{d}}{\mathrm{d}\theta}\mathrm{MSE}\Big(\hat{\theta}_{i,\epsilon,\hat{\phi}_{i,\epsilon}}\Big)\frac{\mathrm{d}}{\mathrm{d}\epsilon}\mathcal{I}_\theta^{(1)}(S_i)\bigg|_{\epsilon=0}}_{\text{Ignore}} \tag{324}$$

$$\approx \mathcal{I}_\theta^{(1)}(S_i)^\top \frac{\mathrm{d}^2}{\mathrm{d}^2\theta}\mathrm{MSE}(\hat{\theta})\,\mathcal{I}_\theta^{(1)}(S_i). \tag{325}$$

Hessian-vector product can now again be used to copmute this approximation of $\mathcal{I}_{\mathrm{MSE}}^{(2)}(S_i)$ using the already computed $\mathcal{I}_\theta^{(1)}(S_i)$.

- `Update`: Uses any optimizer to update the parameters of the policy.

## H.2 INFLUENCE FUNCTION USAGE WHEN THE ESTIMATES ARE STOCHASTIC

When using influence functions for deep-learning based estimators, prior works raise the caution that influence function might not provide an accurate estimate of the leave-one-out loss (Basu et al., 2020; Bae et al., 2022). One of the reasons is that the optimization process is susceptible to converge to different (local) optimas based on the stochasticity in the training process. Therefore, the difference between an estimate of $\theta(D_n)$ and $\theta(D_{n\setminus i})$ is not only due to the sample $S_i$ but also on the other stochastic factors (e.g., what was the parameter initialization when computing $\theta(D_n)$ vs $\theta(D_{n\setminus i})$). However, computing the influence of sample $S_i$ using influence functions does not account for stochasticty in the training process. In the following, we discuss why this might be less of a concern for our use-case.

Let $\xi$ be a noise variable that governs all the stochastic factors (e.g., parameter initialization). Let $\theta(D_n, \xi)$ be a deterministic function given a dataset $D_n$ and $\xi$. Therefore, similar to (4),

$$\mathcal{L}(\pi) = \mathop{\mathbb{E}}_{D_n \sim \pi, \xi}[\mathrm{MSE}(\theta(D_n, \xi))]. \tag{326}$$

Then similar to (5), (7), and (9)

$$\nabla\mathcal{L}(\pi) = \mathop{\mathbb{E}}_{D_n \sim \pi, \xi}\left[\mathrm{MSE}(\theta(D_n, \xi))\sum_{i=1}^{n}\frac{\partial\log\pi(Z_i|X_i)}{\partial\pi}\right] \tag{327}$$

$$= \mathop{\mathbb{E}}_{D_n \sim \pi, \xi}\left[\sum_{i=1}^{n}\frac{\partial\log\pi(Z_i|X_i)}{\partial\pi}\big(\mathrm{MSE}(\theta(D_n, \xi)) - \mathrm{MSE}\big(\theta\big(D_{n\setminus i}, \xi\big)\big)\big)\right] \tag{328}$$

$$\approx \mathop{\mathbb{E}}_{D_n \sim \pi, \xi}\left[\sum_{i=1}^{n}\frac{\partial\log\pi(Z_i|X_i)}{\partial\pi}\mathcal{I}_{\mathrm{MSE}}(S_i, \xi)\right] \tag{329}$$

where, $\mathcal{I}_{\mathrm{MSE}}(S_i, \xi)$ is similar to (9) and (10), but $\theta$ is also a function of $\xi$. Therefore, from (328) it can be observed that, for our setting, we indeed want to compute the difference from the leave-one-out estimate, where there is no difference due to the stochasticity from $\xi$.

In practice, as we compute a sample estimate using a single sample of $D_n$ and $\xi$,

$$\hat{\nabla}^{\mathrm{IF}}\mathcal{L}(\pi) = \sum_{i=1}^{n}\frac{\partial\log\pi(Z_i|X_i)}{\partial\pi}\mathcal{I}_{\mathrm{MSE}}(S_i, \xi), \tag{330}$$

the resulting procedure is essentially the same as if running the DIA algorithm considering $\theta(D_n)$ to be deterministic. Our experimental setup uses DIA from Algorithm 1 as-is for all the domains considered. For instance, TripAdvisor setting uses deep learning based estimators whose outputs are stochastic (e.g., due to parameter initialization), and the synthtetic IV domain uses closed form solutions where there is no stochasticity in the learned estimator, given the data.

Beyond the initialization difference, Bae et al. (2022) also point out three other sources of issue with practical influence function computation. (a) Linearization: or using first order influences in (9). For our purpose, we formally characterize in Theorem 5 the bias due to such approximation. Further, in (325), we discussed how second-order influence can also be estimated partially and be used to obtain a better estimate of the leave-one-out difference. (b) Non-convergence of the parameters. As experiment design is most useful when sample complexity is much more expensive than compute complexity, this issue can be mitigated with longer training time. (c) Regularization when computing the inverse hessian vector product. For our purpose, we used conjugate gradients with hessian vector products to compute the inverse hessian vector product (see discussion around (320)). In our experimental setup we did not use any additional regularization for this computation.

More importantly, contributions of our proposed DIA framework is complementary to the solutions to the above mentioned challenges. Therefore, any advancements to resolve these challenges can also potentially be incorporated in DIA.

### H.3 PROXY MSE ALTERNATIVES

Recall from 4 that under the knowledge of the oracle $\theta_0$, the MSE can be expressed as,

$$\text{MSE}(\theta(D_k)) = \frac{1}{M} \sum_{i=1}^{M} (f(X_i, A_i; \theta_0) - f(X_i, A_i; \theta(D_k)))^2. \tag{331}$$

However, as we do not have access to either $\theta_0$, or (unbiased estimates of) the outcomes of $f(X_i, A_i; \theta_0)$, we cannot directly estimate (331). One way would be to decompose the MSE is bias and variance, and only consider the variance component,

$$\widehat{\text{Var}}(\theta(D_k)) = \frac{1}{M} \sum_{i=1}^{M} (\mathbb{E}[f(X_i, A_i; \theta(D_k))] - f(X_i, A_i; \theta(D_k)))^2 \tag{332}$$

Just optimizing for the variance is analogous to the objective considered by most prior work (Chaloner & Verdinelli, 1995; Rainforth et al., 2023) and can also be easily used in our framework. However, instead of discarding the bias term from the MSE's decomposition, we consider the proxy $\widehat{\text{MSE}}$ where $\theta_0$ is substituted with $\theta(D_n)$ (Lines 5 and 11 in Algorithm 2),

$$\widehat{\text{MSE}}(\theta(D_k)) = \frac{1}{M} \sum_{i=1}^{M} (f(X_i, A_i; \theta(D_n)) - f(X_i, A_i; \theta(D_k)))^2 \tag{333}$$

Intuitively, for a consistent estimator, $\theta(D_n)$ can be considered a reasonable approximation of $\theta_0$ for estimation of the MSE of $\theta(D_k)$, where $k$ is much smaller than $n$ (for e.g., $k = o(n)$). This approach is similar to MSE estimation using sub-sampling bootstrap (Romano, 1995) and we provide formal justification for this in Theorem 8.

As we noticed above, there are multiple different choices of target quantity that can be used, and this work focused on studying the one in (333). However, an important advantage of DIA is that the target is easily customizable, and one can use any desired target quantity $f_?$,

$$\text{MSE}_?(\theta(D_k)) = \frac{1}{M} \sum_{i=1}^{M} (f_?(X_i, A_i) - f(X_i, A_i; \theta(D_k)))^2. \tag{334}$$

## I ADDITIONAL EMPIRICAL DETAILS

To demonstrate the flexibility of the proposed approach, we use DIA across various settings, with different kinds of estimators,

- IV (synthetic): For this domain, we use 2SLS based linear regression for both the nuisance and ATE parameters, where the solutions can be obtained in a closed-form. The parameters $\varphi$ of the instrument-sampling policy $\pi_\varphi$ correspond to logits for each instrument. The results for this setting are reported using 100 trials.

- CIV (synthetic): For this domain, we use moment conditions from (Hartford et al., 2017) and use logistic regression with linear parameters for both the nuisance and CATE parameters, and thus require gradient descent to search for the optimal parameters (no closed-form solution). Further, to account for covariates, the parameters $\varphi$ of the instrument-sampling policy $\pi_\varphi$ correspond to the parameters of a non-linear function modeled using a neural network. The results for this setting are reported using 30 trials.

- TripAdvisor (semi-synthetic): For this domain, we use neural network based non-linear parameterization of both $\theta$ and $\phi$, for the CATE and the nuisance functions, respectively. The parameters $\varphi$ of the instrument-sampling policy $\pi_\varphi$ also correspond to the parameters of a non-linear function modeled using a neural network. The results for this setting are reported using 30 trials.

The policy parameters $\varphi$ in all the settings are searched using gradient expression in 17. Below, we discuss details for each of these settings.

### I.1 SYNTHETIC DOMAINS

For both the IV and the CIV setting, random $5\%$ (or 1, whichever is more) of the instruments had strong positive encouragement for treatment $A = 0$ and $5\%$ (or 1, whichever is more) had strong positive encouragement for treatment $A = 1$. The strength ($\gamma$) of other instruments was in between. Below, $|\mathcal{Z}|$ denotes the number of instruments.

The following provides the data-generating process for the IV setting:

$$\theta_0 \in \mathbb{R}^2, \qquad \gamma \in [0,1]^{|\mathcal{Z}|}, \qquad \pi(\cdot) \in \Delta(\mathcal{Z}) \tag{335}$$

$$Z \stackrel{(a)}{\sim} \pi(\cdot) \qquad\qquad\qquad \in \{0,1\}^{|\mathcal{Z}|} \tag{336}$$

$$U \sim \mathcal{N}(0, \sigma_u) \qquad\qquad\qquad \in \mathbb{R} \tag{337}$$

$$p = \texttt{clip}\big(Z^\top \gamma + U, \min = 0, \max = 1\big) \qquad\qquad \in [0,1] \tag{338}$$

$$A \sim \texttt{Bernoulli}(p) \qquad\qquad\qquad \in \{0,1\} \tag{339}$$

$$\xi \stackrel{(b)}{\sim} A\mathcal{N}(0, \sigma_1) + (1 - A)\mathcal{N}(0, \sigma_0) \qquad\qquad \in \mathbb{R} \tag{340}$$

$$Y = [A, 1 - A]^\top \theta_0 + U + \xi \qquad\qquad\qquad \in \mathbb{R} \tag{341}$$

where (a) samples from a multinomial and $Z$ is a one-hot vector that indicates the chosen instrument, and (b) induces heteroskedastic *noise* in the outcomes based on the chosen treatment.

For the IV setting, we make use of 2SLS which makes use of linear functions in both the first and the second stages of regression. The parameters for those settings can be obtained in closed form. Additionally, the influence function can also be computed in closed-form (Kahn, 2015). The instrument sampling policy $\pi_\varphi(\cdot)$ is an (unconditional) distribution on $\Delta(\mathcal{Z})$.

The following provides the data-generating process for the CIV setting:

$$\theta_0 \in \mathbb{R}, \qquad X \in \mathbb{R}^d, \qquad \gamma \in [0,1]^{|\mathcal{Z}|}, \qquad \forall x: \ \pi(\cdot|x) \in \Delta(\mathcal{Z}) \tag{342}$$

$$Z \stackrel{(a)}{\sim} \pi(\cdot|X) \qquad\qquad\qquad \in \{0,1\}^{|\mathcal{Z}|} \tag{343}$$

$$U \sim \mathcal{N}(0, \sigma_u) \qquad\qquad\qquad \in \mathbb{R} \tag{344}$$

$$C \stackrel{(b)}{=} X_1 \gamma + (1 - X_1)(1 - \gamma) \qquad\qquad \in [0,1]^{|\mathcal{Z}|} \tag{345}$$

$$p = \texttt{clip}\big(Z^\top C + U, \min = 0, \max = 1\big) \qquad\qquad \in [0,1] \tag{346}$$

$$A \sim \texttt{Bernoulli}(p) \qquad\qquad\qquad \in \{0,1\} \tag{347}$$

$$\xi \stackrel{(c)}{\sim} A\mathcal{N}(0, \sigma_1) + (1 - A)\mathcal{N}(0, \sigma_0) \qquad\qquad \in \mathbb{R} \tag{348}$$

$$Y \stackrel{(d)}{=} X_2 + A\theta_0 + U + \xi \qquad\qquad\qquad \in \mathbb{R} \tag{349}$$

where (a) samples from a multinomial and $Z$ is a one-hot vector that indicates the chosen instrument, (b) induces heteroskedastic compliance by flipping the compliance factor of the instruments based on the covariates (by construct, $X_1 \in \{0, 1\}$), (c) induces heteroskedastic *noise* in the outcomes based on the chosen treatment, and (d) induces heteroskedastic outcomes based on the covariates (by construct, $X_2 \in \mathbb{R}$).

For the CIV setting, $d = 2$ and we parameterize $\theta$ such that $g(X, A; \theta)$ as a linear function of $X, A$. Similarly, we $\phi$ is linearly parametrized to estimate the nuisance function $\mathrm{d}P(A|X, Z; \phi)$ using logistic regression. Parameters for both $g$ and $\mathrm{d}P$ are obtained using gradient descent and the moment conditions in 3. The instrument sampling policy $\pi_\varphi(\cdot|x)$ is modeled using a two-layered neural network with 8 hidden units.

## I.2    TRIPADVISOR DOMAIN (SEMI-SYNTHETIC)

Adaptivity and learning the distribution of instruments are most relevant when there are a lot of instruments. Our paper is developed for the setting where online sampling is feasible. Unfortunately, simulators for real-world scenarios that we are aware of and could access publicly (e.g., TripAdvisor's (Syrgkanis et al., 2019)) only consider a single/binary instrument. These simulator domains are modeled to represent the setting where expert knowledge was used to find optimal instruments (e.g., better/easier sign-up strategy to make the membership process easier). However, in reality, there could be multiple ways to encourage membership (emails, notifications, sign-up bonuses, etc.) that can all serve as instruments. Adaptivity is more relevant in such a setting to discover an optimal strategy for instrument selection, while minimizing the need for expert knowledge. Therefore, given the above considerations, we created a semi-synthetic TripAdvisor with varying numbers of potential instruments to simulate such settings.

The original data-generating process for TripAdvisor domain is available on Github. The co-variates $X \in \mathbb{R}^d$ correspond to how many times the user visited web pages of different categories (hotels, restaurants, etc.), how many times they visited the webpage through different modes (direct-search, ad-click, etc.), and other demographic features (geo-location, laptop operating system, etc.). The original setting only had a binary instrument. To make it resemble realistic settings that are more broadly applicable (e.g., where there are a multitude of different instruments, each corresponding to a different way to encourage a user to uptake the treatment), we incorporated several weak instruments. The compliance strength of the instrument varies non-linearly with the covariates. Our semi-synthetic version is available in the supplementary material.

For the TripAdvisor setting, we parameterize $\theta$ such that $g(X, A; \theta)$ as a non-linear function of $X, A$. Similarly, we $\phi$ is non-linearly parametrized to estimate the nuisance function $\mathrm{d}P(A|X, Z; \phi)$. In both cases, the non-linear function is a two-layered neural network with 8 hidden nodes and sigmoid activation units. Parameters for both $g$ and $\mathrm{d}P$ are obtained using gradient descent and the moment conditions in 3. The instrument sampling policy $\pi_\varphi(\cdot|x)$ is modeled using a two-layered neural network with 8 hidden units.

## I.3    HYPER-PARAMETERS:

The choice of sub-sampling size $k$ is used as a hyper-parameter in our work. We parametrize $k$ as $n^\alpha$. For our empirical results, we searched over the hyper-parameter settings of $\alpha \in [0.5, 0.8]$ and report the best-performing setting.

Auto-tuning suggestion for $\alpha$: As the expected number of samples that get accepted via rejection sampling from a set of $n$ samples is $n/\rho_{\max}$, it would be ideal for $\alpha$ to be such that $n^\alpha > n/\rho_{\max}$. Empirical supremum can be used as a proxy for $\rho_{\max}$.

Further, doing rejection sampling in higher dimensions can be challenging as $\rho_{\max}$ can be very large. A standard technique for density ratio estimation (i.e., $\rho$ that governs acceptance-rejection ratio in 13) in higher dimensions is through performing density ratio estimation on a lower dimensional subspace (Sugiyama et al., 2009, 2011). We believe that incorporating similar ideas into our framework can provide further improvements.

## I.4    DETAILS OF OTHER PLOTS:

**Figure 1:**    This figure provides a comparison with conventional experiment design (that assumes that there is no confounding between what group (treatment/control) the user was recommended to

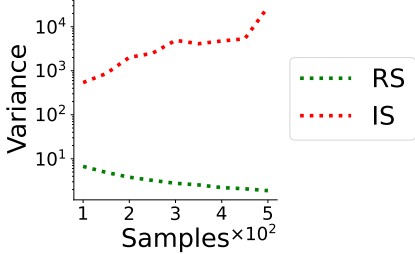

Figure 8: Log-scale version of Figure 4 to better visualize the difference in variances at smaller values of $n$.

be a part of (i.e., $Z$), and the group that the user actually became a part of (i.e., $A$)), and the proposed method DIA on the following data-generating process:

$$\theta_0 \in \mathbb{R}^2, \qquad \pi(\cdot) \in \Delta(\mathcal{Z}) \tag{350}$$

$$Z \sim \pi(\cdot) \qquad\qquad \in \{0,1\} \tag{351}$$

$$U \sim \mathcal{N}(0, \sigma_u) \qquad\qquad \in \mathbb{R} \tag{352}$$

$$p = \texttt{clip}(Z + U, \min = 0, \max = 1) \qquad\qquad \in [0,1] \tag{353}$$

$$A \sim \texttt{Bernoulli}(p) \qquad\qquad \in \{0,1\} \tag{354}$$

$$\xi \overset{(b)}{\sim} A\mathcal{N}(0, \sigma_1) + (1 - A)\mathcal{N}(0, \sigma_0) \qquad\qquad \in \mathbb{R} \tag{355}$$

$$Y = [A, 1 - A]^\top \theta_0 + U + \xi \qquad\qquad \in \mathbb{R} \tag{356}$$

The confounder strength corresponds to $\sigma_u$. Observe that when $\sigma_u = 0$, then the binary treatment's value is the same as that of the binary instrument's value, i.e., $A = Z$.

**Figure 3:** For the illustrative plots regarding bias and variance trade-offs of different gradient estimators and sampling strategies, we make use of the following data-generating process where the treatment $A \in \mathbb{R}$ is real-valued, the outcomes $Y$ are quadratic in $A$, and $|\mathcal{Z}| = 2$,

$$\theta_0 \in \mathbb{R}, \qquad \gamma \in [0,1]^{|\mathcal{Z}|}, \qquad \pi(\cdot) \in \Delta(\mathcal{Z}) \tag{357}$$

$$Z \sim \pi(\cdot) \qquad\qquad \in \{0,1\}^{|\mathcal{Z}|} \tag{358}$$

$$U \sim \mathcal{N}(0, \sigma_u) \qquad\qquad \in \mathbb{R} \tag{359}$$

$$A \sim Z^\top \gamma + U + \mathcal{N}(0, \sigma_a) \qquad\qquad \in \mathbb{R} \tag{360}$$

$$Y \sim A^2 \theta_0 + U + \mathcal{N}(0, \sigma_y) \qquad\qquad \in \mathbb{R} \tag{361}$$

This figure illustrates the bias and variance of different gradient estimators when the function approximator is mis-specified, i.e., $Y$ is modeled as a linear function of treatment $A$ but it is actually a *quadratic* function of treatment. We use the standard 2SLS estimator for this setting.

**Figure 4:** This domain also uses a 2SLS estimator and binary instrument with the data generating process in (357)-(361). **(Left)** Illustration of how the policy orderings are preserved when using different values of $\alpha$ in $k = n^\alpha$. $\pi_0(1) = 0.2$, $\pi_1(1) = 0.5$, and $\pi_2(1) = 0.8$, where the data collection policy is $\beta(1) = 0.5$. **(Right)** Variance of evaluating MSE under different sampling strategies. The data collection policy $\beta(1) = 0.5$ and the target policy $\pi(1) = 0.6$. Sub-sample size $k = o(n) = n^{0.75}$. To better visualize the difference in the variance for small sample sizes, we have also included a log-scale version of the plot in Figure 8

