# OpenReview forum: "Adaptive Instrument Design for Indirect Experiments"
_ICLR.cc/2024/Conference — ICLR 2024 poster_

### Official Review · Reviewer_QTvC · 2023-10-18

**Soundness:** 3 good
**Presentation:** 3 good
**Contribution:** 3 good
**Rating:** 6
**Confidence:** 3

**Summary:**

The paper introduces the computational procedure DIA to increase the sample efficiency for indirect experiments for CATE estimation.  Specifically, the procedure leverages influence functions for iteratively optimizing the data collection policy through adaptive instrument selection.  Optimality is defined through a minimal mean-squared error of the estimator.  The paper emphasizes the applicability of the theoretical procedure through experiments conducted on two synthetic datasets and an extended version of the semi-synthetic TripAdvisor dataset.

**Strengths:**

1) To the best of my knowledge, the presented problem statement is novel and of high interest for CATE estimation in scenarios in which direct interventions are impractical, costly or unethical.
2) The authors provide theory for the proposed method DIA.

**Weaknesses:**

1) The paper suffers from impreciseness and multiple open questions (see below).
2) The results are not reproducible: The experimental setting (e.g., the employed datasets) are not described, neither is code provided. Besides the reproducibility aspects, this also hinders understanding and interpretation of the presented figures and results.
3) The paper does not employ the required format of citations in the text, i.e. reference number instead of author name + year.
4) The motivation could be much improved by elaborating more exactly where the method is relevant and providing a real-world case study.

**Questions:**

The paper would benefit from additional details on the following questions:

1) Why is the MSE a good metric of choice? What is the benefit above other objectives (e.g., minimizing regret)?
2) What would be the general form of the k-th order influence function?
3) What is meant by the perturbation of the distribution function induced by Dn? In which sense are the samples perturbed?
4) Why can the estimator theta(Dn) be considered symmetric? A short mathematical statement would benefit the flow of reading.
5) What is the formulation of the importance ratio on the entire data? (page 6 might be difficult to follow for non-expert readers).
6) Similarly, a short introduction/theory of the acceptance-rejection method would be beneficial. I am not convinced that the average reader is familiar with the concept.
7) The authors mention the flexibility of their approach multiple times. Nevertheless, it is unclear what precisely is meant by flexibility/ which properties make the approach flexible.
9) How does the performance of DIA-X for different X relate to the data generating mechanism? Can the optimal X be deducted from the data for the presented experiments?
10) Figure 3: The funnel (standard dev.?) is only plotted for the IV-bias. Why is it not visible for the other cases or the variance plot?
11) TripAdvisor data: Why was it necessary to synthetically enhance the data? Could you also provide a comparison of the results of the method on the original real-world dataset and the enhanced semi-synthetic dataset? It would also be interesting for the reader if the authors could state the optimal instruments for the real-world dataset.

The paper has multiple imprecise parts:

1) Background: The theoretical background/ problem setting of the paper is taken from reference 23. In contrast to the original work, the paper lacks motivation for and explanation of the specific problem setting.
2) Control Variates: The choice of the control variate (section 3.1) should be justified mathematically (possibly in the appendix).
3) Figure 4: According to the text, one should observe an exponentially increasing variance for IS in the number of samples. However, the figure does not support this statement.
4) Experiments: The synthetic data generation is not properly described. Furthermore, no details on the networks employed for estimation are stated.
5) It would be nice to formalize \pi in Sec 2 upon first experience.
6) References: Multiple references are incomplete.

Things to improve the paper that did not impact the score:

1) Structure: The structure of the paper can be optimized to facilitate reading. A constant reminder of what will follow later (e.g., Section 4) might confuse the reader more than it aids the flow of the paper. Furthermore, key contributions (e.g. Theorem 2) are referred to the appendix. It would aid the understanding of the theory to have the main results presented as such in the main part of the paper.
2) Consistency in wording, e.g., importance weighting vs. importance sampling
3) Multiple times it is unclear if a certain aspect is only assumed or proven. Clarification emphasizes the contribution of the paper.

Minor comments:

1) There are some minor typos and grammar errors in the text.

---

> ### Author Response · Authors · 2023-11-16
> **Response to Reviewer QTvC (1/3)**
>
> We thank the reviewer for their appreciation of the novelty of the proposed results, and also for their valuable time to provide constructive feedback regarding all aspects of the draft. Your feedback has helped us significantly improve the accessibility of the paper for a wider audience.
>
> **W2. The results are not reproducible: The experimental setting (e.g., the employed datasets) are not described, neither is code provided. Besides the reproducibility aspects, this also hinders understanding and interpretation of the presented figures and results.**
>
> Thank you for sharing the concern about reproducibility. We have uploaded the code under the supplementary material section. It contains the data-generating processes and the proposed algorithm.  We will publicly release the code and expand Appendix I to include more experimental details.
>
> **W3. The paper does not employ the required format of citations in the text, i.e. reference number instead of author name + year.**
>
> We will update the reference style in the draft.
>
> **W4. The motivation could be much improved by elaborating more exactly where the method is relevant and providing a real-world case study.**
>
> Due to space constraints, we had presented the motivation concisely, but thank you for highlighting that further discussion on some real-world use cases would be helpful. Below we have mentioned three use cases across different fields of application:
>
> **Education:** It is important for designing an education curriculum to estimate the effect of homework assignments, extra reading material, etc. on a student’s final performance [1]. However, as students cannot be forced to (not) do these, conducting an RCT becomes infeasible. Nonetheless, students can be encouraged via a multitude of different methods to participate in these exercises. These different forms of encouragement provide various instruments and choosing them strategically can enable sample efficient estimation of the desired treatment effect.
>
>
> **Healthcare:** Similarly, for mobile gym applications, or remote patient monitoring settings [2], users retain the agency to decide whether they would like to follow the suggested at-home exercise or medical intervention, respectively. As we cannot perform random user assignment to a control/treatment group, RCTs cannot be performed. However, there is a variety of different messages and reminders (in terms of how to phrase the message, when to notify the patient, etc.) that serve as instruments. Being strategic about the choice of the instrument can enable sample efficient estimation of treatment effect.
>
>
> **Digital marketing:** Many online digital platforms aim at estimating the impact of premier membership on both the company’s revenue and also on customer satisfaction. However, it is infeasible to randomly make users a member or not, thereby making RCTs inapplicable. Nonetheless, various forms of promotional offers, easier sign-up options, etc. serve as instruments to encourage customers to become members. Strategically personalizing these instruments for customers can enable sample-efficient treatment effect estimation. In our work (Figure 1), we consider one such case study using the publicly available TripAdvisor domain [3].
>
> [1] Eren, Ozkan, and Daniel J. Henderson. "The impact of homework on student achievement." The Econometrics Journal 11, no. 2 (2008)
>
> [2] Ferstad, J., Prahalad, P., Maahs, D. M., Fox, E., Johari, R., & Scheinker, D. (2022). 1009-P: “The Association between Patient Characteristics and the Efficacy of Remote Patient Monitoring and Messaging.” Diabetes, 71
>
> [3] Syrgkanis, Vasilis, Victor Lei, Miruna Oprescu, Maggie Hei, Keith Battocchi, and Greg Lewis. "Machine learning estimation of heterogeneous treatment effects with instruments." Advances in Neural Information Processing Systems
>
> **Q1. Why is the MSE a good metric of choice? What is the benefit above other objectives (e.g., minimizing regret)?**
>
> That’s a great question! Our chosen metric is closely related to simple regret (that compares the final performance of the estimator to the ground truth). The proposed method DIA (Section 3.3) adapts the key ideas and aims to be optimal after each round of re-allocation, which is related to the cumulative regret as it aims to learn an estimator that is optimal not just at the end, but also after each re-allocation.

---

> ### Author Response · Authors · 2023-11-16
> **Response to Reviewer QTvC (2/3)**
>
> **Q2. What would be the general form of the k-th order influence function?**
>
> Thank you for highlighting that some readers might benefit from the general form of higher-order influence functions. Writing an explicit form for a general $k$-th order can often be tedious, and computing it can be practically challenging. Discussion about higher-order influences for the single-stage estimators can be found in the works by Alaa et al. [1] and Robins et al. [2]. As we formally show in Property 3, for our purpose, we recommend using $k=1$ as it suffices to dramatically reduce the variance of the gradient estimator, incurring bias that decays quickly, while also being easy to compute.
>
> [1] Alaa, Ahmed, and Mihaela Van Der Schaar. "Discriminative jackknife: Quantifying uncertainty in deep learning via higher-order influence functions." International Conference on Machine Learning. PMLR, 2020.
>
> [2] Robins, James, et al. "Higher order influence functions and minimax estimation of nonlinear functionals." Probability and statistics: essays in honor of David A. Freedman. Vol. 2. Institute of Mathematical Statistics, 2008. 335-422.
>
> **Q3. What is meant by the perturbation of the distribution function induced by Dn? In which sense are the samples perturbed?**
>
> We will update the draft to emphasize that perturbing the distribution function (in the direction of a sample S) induced by Dn corresponds to up-weighting the moment conditions corresponding to the data sample S. A formal discussion of perturbed moment conditions, along with exact equations, are also available in Appendix C.
>
> **Q4. Why can the estimator theta(Dn) be considered symmetric? A short mathematical statement would benefit the flow of reading.**
>
> Thank you for helping us improve the flow of the paper. We assume an estimator $\theta(Dn)$ to be symmetric in its arguments as popular estimators do not treat data samples differently based on their index number in the dataset. For example, let $D_2=(S_1, S_2)$ and $D_2’=(S_2, S_1)$, then $\theta(D_2) = \theta(D_2’)$.
>
> While it is possible to define contrived estimators that are not symmetric, (e.g., estimators that only use the first sample, and discard all other samples), we focus on popular estimators that are symmetric.
>
> **Q5. What is the formulation of the importance ratio on the entire data? (page 6 might be difficult to follow for non-expert readers).**
>
> We appreciate the reviewer’s feedback regarding accessibility. We will updated the draft to include the importance ratio formulation of the entire data.
>
> $\nabla \mathcal L(\pi)= \underset{D_n \sim \beta} {\mathbb E}\left[\left(\prod_{i=1}^n \frac{\pi(Z_i|X_i)}{\beta(Z_i|X_i)}\right)\text{MSE}(\theta(D_n)) \sum\limits_{i=1}^n \frac{\partial \log \pi(Z_i|X_i)}{\partial \pi} \right]$
>
> **Q6. Similarly, a short introduction/theory of the acceptance-rejection method would be beneficial. I am not convinced that the average reader is familiar with the concept.**
>
> We are happy to add this. Due to space limitations, we will put it in Appendix F.
>
> **Q7. The authors mention the flexibility of their approach multiple times. Nevertheless, it is unclear what precisely is meant by flexibility/ which properties make the approach flexible.**
>
> Thank you for raising this issue. By flexible, we mean that DIA can be used with a variety of estimators (it is suitable for approaches that assume a linear structure and non-linear), and settings (unconditional IV where instruments do not depend on the context, or conditional IV, where they do). We describe this at the start of Section 5, but we will clarify this earlier, when we first discuss the flexibility of our proposed approach.
>
> **Q8. How does the performance of DIA-X for different X relate to the data generating mechanism? Can the optimal X be deducted from the data for the presented experiments?**
>
> Thank you for asking this question! We will add more details about the domain to clarify that X is not a variable that is being optimized, and hence there is no optimal X. Instead, X is a part of the domain specification. For instance, DIA-10 corresponds to the performance of using DIA when the domain has 10 instruments. Of course, given these 10 instruments, DIA needs to learn a sampling distribution over these instruments, conditioned on the context.
>
> **Q9. Figure 3: The funnel (standard dev.?) is only plotted for the IV-bias. Why is it not visible for the other cases or the variance plot?**
>
> Your careful observation is much appreciated. We will update Figure 3 to include the variance/std-err in the variance estimates itself as well.

---

> ### Author Response · Authors · 2023-11-16
> **Response to Reviewer QTvC (3/3)**
>
> **Q10. TripAdvisor data: Why was it necessary to synthetically enhance the data? Could you also provide a comparison of the results of the method on the original real-world dataset and the enhanced semi-synthetic dataset? It would also be interesting for the reader if the authors could state the optimal instruments for the real-world dataset.**
>
> That’s an important point and we will include a discussion about this in the updated draft. As evident from Figures 1,5, and 7, adaptivity and learning the distribution of instruments are most relevant when there are a lot of instruments. Our paper is developed for the setting where online sampling is feasible. Unfortunately, simulators for real-world scenarios that we are aware of and could access publicly (e.g., TripAdvisor’s) only consider a single/binary instrument. These simulator domains are modeled to represent the setting where expert knowledge was used to find optimal instruments (e.g., better/easier sign-up strategy to make the membership process easier). However, in reality, there could be multiple ways to encourage membership (emails, notifications, sign-up bonuses, etc.) that can all serve as instruments. Adaptivity is more relevant in such a setting to discover an optimal strategy for instrument selection, while minimizing the need for expert knowledge. Therefore, given the above considerations, we created semi-synthetic domains with varying numbers of potential instruments to simulate such settings.
>
>
> **Background: The theoretical background/ problem setting of the paper is taken from reference 23. In contrast to the original work, the paper lacks motivation for and explanation of the specific problem setting.**
>
> We agree with the reviewer that having more motivation for the specific instrumental variable setting and method can potentially help some readers. Unfortunately, we were constrained by the page limit. Since the IV setting itself is already well established in prior literature, we chose to focus on the motivation regarding why strategic sampling of instruments is useful---that is the subject of our paper. We will revise the background to further emphasize and motivate our setting.
>
> **Control Variates: The choice of the control variate (section 3.1) should be justified mathematically (possibly in the appendix).**
>
> Thank you for raising this wasn’t clear– we did actually provide a formal justification of this (see Property 2 in Section 4 which states the proposed control variate-based gradient estimator is both unbiased and has lower variance: a formal proof is provided in the appendix under Theorem 4).  We will update the paper to mention to the readers earlier, in Section 3.1 itself when we introduce the control variate.
>
> **Figure 4: According to the text, one should observe an exponentially increasing variance for IS in the number of samples. However, the figure does not support this statement.**
>
> We should have and will highlight that the naive importance sampling-based estimator can have variance exponential in the number of samples, in the worst case, though it may not always have exponential dependence in all domains. In practice, we observe that the variance is too large to do any meaningful gradient estimation. For instance, in Figure 4, the variance of the IS estimator is around 100000, whereas the variance of the proposed method is <10.
>
> **Experiments: The synthetic data generation is not properly described. Furthermore, no details on the networks employed for estimation are stated.**
>
> We agree more details would improve the paper – we have uploaded the code under supplementary material. The code contains details about data generation and the models used. We will also expand Appendix I to include more experimental details.
>
> **It would be nice to formalize \pi in Sec 2 upon first experience.**
>
> We will update section 2 to emphasize that $\pi(\cdot|x) \in \Delta(Z)$ denotes a distribution over the instruments, conditional on the covariates.
>
> **References: Multiple references are incomplete.**
>
> We found one reference that lacked relevant details (the reference to Phil Thomas’s Safe RL was missing the school and that it was a Ph.D. thesis). If there are others with missing information, please let us know.
>
> **Things to improve the paper that did not impact the score**
>
> We thank the reviewer again for their careful review, and we will incorporate as many suggestions as possible to improve the readability of the draft further.

---

> > ### Comment · Reviewer_QTvC · 2023-11-20
> >
> > It is nice to see that the authors provide a very detailed and thoughtful response. I also appreciate that they promise to make several, important changes for the camera-ready version. However, it is unfortunate that the authors would do so only post rebuttal. Given the significance and scope of the changes, I would like to see the revised version before increasing my score.

---

> > > ### Author Response · Authors · 2023-11-21
> > > **Response to Reviewer QTvC**
> > >
> > > We thank the reviewer for taking out their valuable time to engage in a discussion and provide positive feedback on the proposed changes. Based on your feedback, we have uploaded a revised version of the draft (summary of the changes are provided in the global response to all the reviewers) and also uploaded the code.

---

> > > > ### Comment · Reviewer_QTvC · 2023-11-22
> > > >
> > > > Thank you for providing an updated version of your draft. We took of note that the revisions have improved the paper to some amount.
> > > > Nevertheless, there are still some open problems:
> > > > Flow of paper:
> > > > - The introduction of notation is not comprehensive (e.g., $\pi$, $\Pi$). Rewriting the beginning of Section 2 to include all notation used in this Section would significantly help the flow of the paper.
> > > > - It is still difficult to follow the argument in the text (e.g. continuous references to later sections). Main results are still referred to the Appendix.
> > > > - The restriction to symmetric estimators should be mentioned early in the paper, not only one page 5.
> > > > Influence functions:
> > > > - How would a practicioneer know if all k-th order influence functions exist?
> > > > - Introducing eq.9 is highly confusing at this point in the text. The reader would expect a general form of influence functions, not only the form for $k=1$.
> > > > If presenting a general form (or even forms for other small $k$ is not possible, then the order of equations 8 to 12 should be revised.
> > > > It should be clear from the beginning, that only $k=1$ is needed.
> > > > - Why is $k=1$ suffient? Property 3 does not directly state this, since $\mathcal(L)\pi$ was defined over all k-th order functions.
> > > > Figure 4):
> > > > - We agree that the variance of IS is multiple times higher that the variance of RS. Nevertheless, it does not increase exponentially in the sample size (one could argue that this might happen only after ~4500 samples)
> > > > Therefore, the figure still does not support the statement in the text.

---

> ### Author Response · Authors · 2023-11-23
> **Further clarification (1/2)**
>
> We thank the reviewer again for taking out their valuable time and engaging in a constructive discussion. We were glad to hear that our previous response was able to address some of the concerns. Below we provide comments for the remaining ones:
>
> **1. Flow of paper:**
>
> **The introduction of notation is not comprehensive (e.g., ,). Rewriting the beginning of Section 2 to include all notation used in this Section would significantly help the flow of the paper.**
>
> Thank you for helping us improve the readability of the draft further. We have updated Section 2 to ensure that all the relevant notation is introduced before the problem statement is discussed.
>
> **It is still difficult to follow the argument in the text (e.g. continuous references to later sections). Main results are still referred to the Appendix.**
>
> We appreciate the reviewer’s encouragement to make the paper even clearer. As the reviewer can imagine, the page limits mean that many of our results and details had to be deferred to the appendix, and we do appreciate that the comments from other reviewers’ suggest that the current organization does appeal to some of the target audience. Of course, we would be grateful if the reviewer could point out particular results in the appendix that they think would be beneficial to appear in the main text.
>
>
> **The restriction to symmetric estimators should be mentioned early in the paper, not only one page 5.**
>
> Thank you for bringing this up.  We have also mentioned this now in Section 2, right where we first introduce the estimators, before stating the problem statement. And then in Section 3.2 again as a reminder to the readers, when the symmetric condition is used.
>
> **2. Influence functions:**
>
> **Introducing eq.9 is highly confusing at this point in the text. The reader would expect a general form of influence functions, not only the form for k=1. If presenting a general form (or even forms for other small is not possible, then the order of equations 8 to 12 should be revised. It should be clear from the beginning, that only k=1 is needed.**
>
> We like the reviewer’s suggestion! We have updated the draft to revise the ordering of this section, such that it is clear that only $k=1$ is needed. Thank you for helping us improve the clarity of this section.

---

> ### Author Response · Authors · 2023-11-23
> **Further Clarification (2/2)**
>
> **How would a practitioner know if all k-th order influence functions exist?
> Why is k=1 sufficient? Property 3 does not directly state this, since was defined over all k-th order functions.**
>
> Thanks for letting us know we had not yet clearly explained this. The existence of higher-order influence requires smoothness of the estimate to perturbations in sample weights in the loss. For example, let $l_i(\theta)$ be the loss from the $i^{th}$ sample, such that the total loss is $L(\theta) = \frac{1}{n} \sum_i l_i(\theta)$. Here, the weight associated with the loss of each sample is $1/n$. Here smoothness would refer to how would the minimizer of $L(\theta)$ change if for one of the samples, the weight is changed from $1/n$ to $(1/n +\epsilon)$, as $\epsilon \rightarrow 0$.
>
> Note that such an assumption on the smoothness of estimators is not new and is used across a wide range of applications of influence functions. For example: for model selection [1], dataset distillation [2], uncertainty quantification [3], studying model generalization [4], studying interpretability of models [5], and many other applications [6]. We have highlighted these works in the extended related work section in Appendix A.
>
> Fortunately, under this (common) assumption, Property 3 ensures that $k=1$ suffices for our setting to reduce variance, incur small bias, and remain computationally feasible. It is worth pointing out that the above-mentioned works [1,2,3,4,5,6] also only consider $k=1$ for their respective applications.
>
> Additionally, performance improvement achieved by DIA suggests that this assumption also seems robust to different kinds of estimators (linear with closed form (Fig 5), logistic regression (Fig 6), and neural-network-based non-linear estimators (Fig 1)) used for our experiments.
>
> [1] Alaa, Ahmed, and Mihaela Van Der Schaar. "Validating causal inference models via influence functions." In International Conference on Machine Learning, pp. 191-201. PMLR, 2019.
>
> [2] Loo, Noel, Ramin Hasani, Mathias Lechner, and Daniela Rus. "Dataset Distillation with Convexified Implicit Gradients." arXiv preprint arXiv:2302.06755 (2023).
>
> [3] Alaa, Ahmed, and Mihaela Van Der Schaar. "Discriminative jackknife: Quantifying uncertainty in deep learning via higher-order influence functions." In International Conference on Machine Learning, pp. 165-174. PMLR, 2020
>
> [4] Grosse, Roger, Juhan Bae, Cem Anil, Nelson Elhage, Alex Tamkin, Amirhossein Tajdini, Benoit Steiner et al. "Studying large language model generalization with influence functions." arXiv preprint arXiv:2308.03296 (2023).
>
> [5] Guo, Han, Nazneen Fatema Rajani, Peter Hase, Mohit Bansal, and Caiming Xiong. "Fastif: Scalable influence functions for efficient model interpretation and debugging." arXiv preprint arXiv:2012.15781 (2020).
>
> [6] Koh, Pang Wei, and Percy Liang. "Understanding black-box predictions via influence functions." In International conference on machine learning, pp. 1885-1894. PMLR, 2017.
>
>
> **3. Figure 4**
>
> **We agree that the variance of IS is multiple times higher that the variance of RS. Nevertheless, it does not increase exponentially in the sample size (one could argue that this might happen only after ~4500 samples) Therefore, the figure still does not support the statement in the text.**
>
> Thank you for helping us improve this illustrative example. We have updated it to bring out this worst-case behavior better. We have also removed the log scale on the y-axis that was present earlier. Appendix I.4 contains more details about the plot, as well as the log-scale version of this plot (Fig 8) to highlight the difference in variances even at smaller values of n.

---

### Official Review · Reviewer_7XxM · 2023-11-02

**Soundness:** 3 good
**Presentation:** 3 good
**Contribution:** 3 good
**Rating:** 6
**Confidence:** 4

**Summary:**

The paper investigates enhancing sample efficiency for indirect experiments by adaptively designing a data collection policy over instrumental variables. The authors propose a practical computational procedure that utilizes influence functions to search for an optimal data collection policy, together with the idea of rejection sampling. By many experiments in various domains, the authors showcase the significant improvement of the sample efficiency in indirect experiments.

**Strengths:**

1.	The problem considered in this paper, from my perspective, is important and interesting from both a theoretical and practical perspective.
2.	The ideas of generalizing the influence functions and adopting rejection sampling are quite inspiring. Although the core concepts are standard, the generalization is still novel, in my opinion.
3.	The paper is well written and relatively easy to follow, considering it is a quite theoretical one.

**Weaknesses:**

1.	Adaptivity can bring us a lot of benefits like the improvement of sample efficiency. However, the technical challenge is that adaptivity usually harms the independence among the data. The adaptively collected data is usually challenging to analysis, even for the basic M-estimator (see, e.g., [A]). The authors do not seem to cover any points along this line. It will be beneficial to present whether the adaptivity in this paper affects the independence structure and how the authors handle such difficulties.
2.	For the multi-rejection sampling, if $\pi’$ and $\pi_i$ are very different, the algorithm might reject many samples before getting a useful one, which can be very inefficient and impractical. Things become even worse when the dimensions of the problem get larger, for example, as the authors mentioned, when the natural language serves as instruments Additionally, it would be good to know how to choose the value of "k" in practice.
3.	In the introduction, when summarizing the contributions, I didn't notice a clear mention of the technical challenges and novel aspects of the work. I found several such points in the main text. It would be beneficial, in my opinion, to include them in the introduction, especially since many of the ideas are quite traditional.
4.	One minor point is that the writing can be further improved.	When referring to an equation, it is better to put the equation number in parentheses. The phrase to define $D_{n\i}$ ($D_n$ except the one in the $log \pi (Z_i | X_i)$ term) is a little confusing.


Reference:
[A] Zhang, K., Janson, L., & Murphy, S. (2021). Statistical inference with m-estimators on adaptively collected data. Advances in neural information processing systems, 34, 7460-7471.

**Questions:**

See above.

---

> ### Author Response · Authors · 2023-11-16
> **Response to Reviewer 7XxM**
>
> We thank the reviewer for appreciating the importance of the problem and the novelty of the contributions. We were glad to see that we were able to convey the key ideas adequately and we agree with the reviewer on the next big steps in this direction. We answer the questions below, in order.
>
> **1. Adaptivity can bring us a lot of benefits like the improvement of sample efficiency. However, the technical challenge is that adaptivity usually harms the independence among the data. The adaptively collected data is usually challenging to analysis, even for the basic M-estimator (see, e.g., [A]). The authors do not seem to cover any points along this line. It will be beneficial to present whether the adaptivity in this paper affects the independence structure and how the authors handle such difficulties.**
>
> That is a great point! While in this work we focused on adaptive data collection and empirical finite sample performance, we will update the conclusion to cite [A] and highlight that adaptive data collection can harm the independence assumption and can make inference more nuanced. Similar to [A], recent work by Gupta et al. [B] also presents initial steps to show consistency and asymptotic normality for simpler GMM estimators under adaptive data collection. While generalizing these results is important, it is orthogonal to our current contribution, and addressing this would result in interesting future work.
>
> [B] Gupta, Shantanu, Zachary Lipton, and David Childers. "Efficient online estimation of causal effects by deciding what to observe." Advances in Neural Information Processing Systems 34 (2021)
>
> **2. For the multi-rejection sampling, if and are very different, the algorithm might reject many samples before getting a useful one, which can be very inefficient and impractical. Things become even worse when the dimensions of the problem get larger, for example, as the authors mentioned, when the natural language serves as instruments Additionally, it would be good to know how to choose the value of "k" in practice.**
>
> Your observation is accurate. A standard technique for density ratio estimation (that governs acceptance-rejection ratio in our setup) in higher dimensions is through performing density ratio estimation on a lower dimensional subspace [1, 2]. We believe that incorporating similar ideas into our framework can provide further improvements.
>
> Regarding setting the value of k: We parametrize $k$ as $n^\alpha$. As the expected number of samples that get accepted via rejection sampling from a set of $n$ samples is $n/\rho_\text{max}$, it would be ideal for $\alpha$ to be such that $n^\alpha > n/\rho_\text{max}$. Empirical supremum can be used as a proxy for $\rho_\text{max}$.
>
> [1] Yamada, Makoto, and Masashi Sugiyama. "Direct density-ratio estimation with dimensionality reduction via hetero-distributional subspace analysis." In Proceedings of the AAAI Conference on Artificial Intelligence, vol. 25, no. 1, pp. 549-554. 2011.
>
> [2] Sugiyama, Masashi, and Motoaki Kawanabe. "Dimensionality reduction for density ratio estimation in high-dimensional spaces." 人工知能学会第二種研究会資料 2009, no. DMSM-A901 (2009).
>
> **3. In the introduction, when summarizing the contributions, I didn't notice a clear mention of the technical challenges and novel aspects of the work. I found several such points in the main text. It would be beneficial, in my opinion, to include them in the introduction, especially since many of the ideas are quite traditional.**
>
> Thank you for the good suggestion– we will update the introduction accordingly.
>
> **4. One minor point is that the writing can be further improved. When referring to an equation, it is better to put the equation number in parentheses. The phrase to define ... is a little confusing.**
>
> We thank the reviewer again for their careful review, and we will incorporate these suggestions to improve the readability of the draft further.

---

> > ### Comment · Reviewer_7XxM · 2023-11-22
> >
> > Thank you for all your efforts and clarifications! I really appreciate it! However, since there is no choices between 6 and 8, I choose to remain my score.

---

> > > ### Author Response · Authors · 2023-11-23
> > >
> > > We are glad to know that the our responses have addressed your questions, and we appreciate your continued positive support for our paper. Thank you for the time and effort you invested in reviewing our paper and our response!

---

### Official Review · Reviewer_SUta · 2023-11-07

**Soundness:** 3 good
**Presentation:** 4 excellent
**Contribution:** 4 excellent
**Rating:** 8
**Confidence:** 3

**Summary:**

Authors present a current and well motivated problem, execute the ideas well, communicate results clearly and as such make a relevant contribution the literature.

The problem statement is delivered at different levels of intuition and formality which helps exposition (i.e. “consider how to automatically learn data-efficient adaptive instrument-selection decision policies, in order to quickly estimate conditional average treatment effects” and “aim to develop an adaptive data collection policy π(·|x) ∈ ∆(Z) over the instruments Z for all x ∈ X that can improve sample-efficiency of the estimate of the counterfactual prediction f. Importantly, we aim to develop an algorithmic framework that can work with general (non)-linear two-stage estimators.”)

Their “general framework for adaptive indirect experiments” is well placed with a reasonable long-term perspective and careful analysis of limitations. They also maturely accommodate natural limitations by presenting a framework that is “designed to minimize the need for expert knowledge and can readily scale with computational capabilities”.

I am unable to comment on the correctness of the theorems, though it seems the authors heavily rely on previously shown results and plug and play those, which, given the empirical results, suggests they are successful.

Empirical examples substantiated the theoretical claims well, though the final discussion has significant potential for development. It is possibly a bit underdeveloped with a weak conclusion of the impact of reallocation.

The conclusion of a ‘Unshaped’ trade off between number of instruments and number of reallocations might oversimplify deeper issues, though for the given work is perfectly adequate.

Overall, the paper makes a relevant contribution to the literature, tackling an important problem with skilful navigation of theoretical tools (influence functions and multi-rejection importance sampling) resulting in convincing results and arguments for using DIA.

**Strengths:**

- [ ] The background is curated at the right level of depth and supports exposition very well. They navigate the page limit well with a carefully formulated and rich appendix.
- [ ] The conceptual development throughout the paper is very consistent and easy to follow.
- [ ] Identifiability conditions are present effectively for the given task.
- [ ] Figure 3 is incredibly effective to demonstrate the improvements, the last sentence could be boldened/underlined/emphasised more. (“Observe the scale on the y-axes.”)
- [ ] The exposition of a linear example in the example is very much appreciated.

**Weaknesses:**

- [ ] I was able to understand “Indirect experiments” on an intuitive level from the problem statement and the given examples, but a less informed reader might struggle and benefit from a more thorough/formal definition, though I am aware authors can’t cater to all levels and the current introduction of the term is just fine for the usual causa ICLR reader.
- [ ] The conclusion is possibly a bit underdeveloped with a weak conclusion of the impact of reallocation, e.g. Figure 5.
- [ ] The conclusion of a ‘Unshaped’ trade off between number of instruments and number of reallocations might oversimplify deeper issues, though for the given work is perfectly adequate.

**Questions:**

- [ ] Do you have more empirical or theoretical evidence to enrich the discussion on the ‘U-shaped’ trade off between number of instruments and number of reallocations? It seem like you might be running out of space, but if camera-ready allows another page, please enrich.
- [ ] Figure 2: I am not sure what the dotted red lines indicate. I assume they are related to special conditions and assumptions on the SCM, but these are not connected to the text as far as I can see. Did I miss it?
- [ ] Typo: Page 5 bottom: “multi-rejection important sampling” should be importance I assume.

---

> ### Author Response · Authors · 2023-11-16
> **Response to Reviewer SUta**
>
> We thank the reviewer for their support and valuable time to provide constructive feedback! We answer the questions, in order, below.
>
> **[ ]I was able to understand “Indirect experiments” on an intuitive level from the problem statement and the given examples, but a less informed reader might struggle and benefit from a more thorough/formal definition, though I am aware authors can’t cater to all levels and the current introduction of the term is just fine for the usual causa ICLR reader.**
>
> We agree with the reviewer. We had to indeed navigate the issues of addressing this to different audiences, within the prescribed page limit, while also discussing all the contributions. We wanted to provide some informal intuition and thus had to defer the formal problem statement to Eqn 4 in Section 2.
>
> **"...impact of reallocation.."  "The conclusion of a ‘Unshaped’ trade off between number of instruments and number of reallocations might oversimplify deeper issues, though for the given work is perfectly adequate." , "more empirical or theoretical evidence to enrich the discussion on the ‘U-shaped’ trade off between number of instruments and number of reallocations? It seem like you might be running out of space, but if camera-ready allows another page, please enrich."**
>
> Thank you for your suggestions to improve the conclusions further! While in this work we took the first step towards addressing several bias-variance challenges underlying a general framework for adaptively designing the instrument sampling policy, our results do provide interesting next challenges to work on and improve the method further. We plan to continue this investigation in future work.
>
> (Unfortunately, it looks like the camera-ready version this year does not allow an additional page.)
>
> **[ ]Figure 2: I am not sure what the dotted red lines indicate. I assume they are related to special conditions and assumptions on the SCM, but these are not connected to the text as far as I can see. Did I miss it?**
>
> That’s correct, the dotted lines visually depict the standard IV condition (Assumption 1) for the SCM. We will update the text to mention this.
>
> **[ ]Typo: Page 5 bottom: “Multi-rejection important sampling” should be importance I assume.**
>
> Thank you for catching this typo. We will fix it.

---

> > ### Comment · Reviewer_SUta · 2023-11-22
> > **Response**
> >
> > Thanks for the response!
> >
> > Without the extra page, it will be hard to supply a rich discussion of the trade-offs, as the appendix would not be the right place, so I hope that content with lesser priority can be moved into the supplement leaving some more room for more detailed analysis of the trade-offs.
> >
> > Thank you for your contribution to the literature, clear writing and sensible design choices and discussion of limitations.

---

> > > ### Author Response · Authors · 2023-11-23
> > >
> > > Thank you once more for your valuable insights and suggestions during the rebuttal process. We will strive to further refine the discussion regarding the trade-offs. Finally, we appreciate the time and effort you invested in reviewing our paper and our response, and your continued positive support for our work.

---

### Official Review · Reviewer_ST2k · 2023-11-08

**Soundness:** 3 good
**Presentation:** 2 fair
**Contribution:** 3 good
**Rating:** 6
**Confidence:** 3

**Summary:**

This paper discusses how to design an adaptive data collection process over the instruments that can improve the sample-efficiency of the estimation of the counterfactual prediction. The previous works mitigate the bias in estimation of the predictor $f$, but they are often subject to high variance problem. This paper mitigates the high variance problem of the gradient estimator by using influence functions and multi-rejection importance sampling and provides theoretical analysis of the bias and variance of the proposed gradient estimator. Specifically, although there exists a slight estimation error, this paper proposes $∇^{IF}L(π)$ estimator based on influence functions to solve the time-consuming problem caused by re-training and the high variance problem. Meanwhile, this paper proposes a multi-rejection important sampling approach, which has lower variance compared to the traditional importance sampling approach. In addition, this paper proposes an algorithm that is able to design instruments adaptively. Moreover, experiments are conducted on three regimes to demonstrate the effectiveness of the proposed method.

**Strengths:**

S1: The motivation is clear. How to design indirect experiments when direct assign interventions is impractical is a important problem.

S2: The techniques adopted in this paper are sound, and are supported by solid theoretical analysis.

S3: The proposed algorithm is novel and demonstrates good performance.

S4: The experimental results on the three datasets are convincing.

**Weaknesses:**

W1: The paper uses uniform data collection method as the baseline, however, I think evaluating some more RL algorithms and comparing their performance will make the paper more convincing.

W2:  In practice, the max mutli-importance ratio may not be known, so this paper use the empirical supremum instead. May $\bar{ρ}(S_{i})$ also be unknown?

W3: Is there some clear pattern for DIA-X with different X in Figure 1, 5 and 6? For example, the U-trend is not obvious in Figure 1.

W4: Is Figure 3 showing the result with K=1 in $∇^{IF}L(π)$? Providing more results with varying K may helpful.

**Questions:**

Please refer to the weaknesses part.

***

After rebuttal: Thank you very much to the authors for their answers to our reviews and for improving the paper during the rebuttal period. The modifications bring valuable content. I read also carefully the other reviews and the corresponding answers. My (positive) recommendation remains unchanged.

---

> ### Author Response · Authors · 2023-11-16
> **Response to Reviewer ST2k**
>
> We were glad to hear that the reviewer found the motivation clear, and the theoretical and empirical analyses useful. We thank the reviewer for appreciating the novelty of the algorithm and taking their valuable time to provide constructive feedback.
>
> **W1: The paper uses a uniform data collection method as the baseline, however, I think evaluating some more RL algorithms and comparing their performance will make the paper more convincing.**
>
> We agree that some justification regarding other RL methods would be useful. We will update the draft to mention that while we leverage tools from RL, our problem setup presents several unique challenges that make the application of standard RL methods as-is ineffective.
>
> Off-policy learning is an important requirement for our method, and policy-gradient and Q-learning are the two main pillars of RL. As we discussed in Section 4, conventional importance-sampling-based policy gradient methods can have variance exponential in the number of samples, rendering them practically useless for our setting. On the other hand, it is unclear how to efficiently formulate the optimization problem using Q-learning. From an RL point of view, the “reward” corresponds to the MSE, and the “actions” corresponds to instruments. Since this reward depends on all the samples, the effective “state space” for Q-learning style methods would depend on the combinatorial set of the covariates, i.e., let $\mathcal X$ be the set of covariates and $n$ be the number of samples, then the state $s \in \mathcal X^n$. This causes additional trouble as $n$ continuously changes/increases in our setting as we collect more data.
>
> **W2: In practice, the max multi-importance ratio may not be known, so this paper uses the empirical supremum instead. May the importance ratio also be unknown?**
>
> Thanks for asking this question! We should have and will clarify in the draft that the multi-importance ratio $\bar \rho(S)$ is always known in our setting. Since the proposed method does the data collection itself, we have access to both the current and the prior data-collection policies that are needed to compute the multi-importance ratio.
>
> **W3: Is there some clear pattern for DIA-X with different X in Figures 1, 5, and 6? For example, the U-trend is not obvious in Figure 1.**
>
> That’s a great observation. Indeed, the severity of the U-trend will vary from problem to problem. In general, our goal in mentioning the U-trend was to highlight the over-arching tension that as the number of potential instruments increases, there is a greater opportunity to be more effective. However, an increase in the number of instruments also increases the challenge with the optimization routine. Therefore, this results in the aforementioned U-trend.
>
> **W4: Is Figure 3 showing the result with K=1? Providing more results with varying K may helpful.**
>
> While we agree that having $k>1$ would be interesting, writing an explicit form for a general $k$-th order influence function can often be tedious, and computing it can be practically challenging. For instance, see the higher-order influences for the single-stage estimators in the works by Alaa et al. [1] and Robins et al. [2]. Fortunately, as we formally show in Property 3, for our purpose, using $k=1$  suffices to dramatically reduce the variance of the gradient estimator, incur a bias that decays quickly, while also being easy to compute.
>
> [1] Alaa, Ahmed, and Mihaela Van Der Schaar. "Discriminative jackknife: Quantifying uncertainty in deep learning via higher-order influence functions." International Conference on Machine Learning. PMLR, 2020.
>
> [2] Robins, James, et al. "Higher order influence functions and minimax estimation of nonlinear functionals." Probability and statistics: essays in honor of David A. Freedman. Vol. 2. Institute of Mathematical Statistics, 2008. 335-422.

---

> > ### Author Response · Authors · 2023-11-23
> > **Kind request for feedback**
> >
> > We would like to express our gratitude for your assessment of both our paper and our rebuttal. If time permits during the remaining author-reviewer discussion period, it would great to receive feedback on whether our response has sufficiently addressed your questions and concerns.
> >
> > Thank you again for your support and constructive suggestions during the rebuttal process!

---

### Author Response · Authors · 2023-11-21
**Global comment to all the reviewers**

To all,

We thank all the reviewers for their valuable time to provide constructive feedback regarding all aspects of the draft. We were glad to know that the reviewers found the problem important (#ST2k, #SUta, #7XxM), concepts easy to follow (#SUta, #7XxM) and novel  (#ST2k, #7XxM, #QTvC), and our empirical and theoretical analyses useful (#ST2k, #QTvC, #SUta).

Based on all feedback and suggestions, we have updated the draft. The requested changes are highlighted in blue. We provide a summary below:

	- Justification for not using other RL methods
	- Highlighting that multi-importance ratio $\bar \rho$ is always computable.
	- Discussion for higher-order influence
	- Clarifying the role of red-arrows in Fig 1
	- Mentioning next important challenges for future work
		○ Incorporating high-dimensional rejection sampling into DIA
		○ Extending existing works on inference for GMM estimators with adaptively collected data
	- Discussion for setting the sub-sampling size $k$
	- More experimental details
		○ Data generating processes
		○ Details of function approximator
	- Updated reference style
	- More discussion on potential real-world use cases
	- Clarification on perturbation of the distribution function
	- Importance sampling formulation for our estimator, and clarification about its worst case behavior
	- Background on rejection sampling
	- Clarification on flexibility of the proposed method
	- Added std-err bars for Fig 3
	- Clarification for the need of semi-synthetic modification to TripAdvisor.
	- Justification for the choice of control variate
	- Other minor typo/fixes/suggestions.

Additionally,

	- The code is now available in the supplementary material
	- Appendix has been updated to improve readability (new index to aid navigation, and some of the proofs have been made more verbose)

---

### Meta-Review · Area_Chair_BK28 · 2023-12-06

**Metareview:**

The reviewers in general found the contribution a worthwhile addition to the literature. Idealized causation with perfect control is often not possible and a focus on indirect experiments is a solid way to go. A clearer contrast to policy optimization and other approaches in reinforcement learning helped in the revised manuscript.

**Justification For Why Not Higher Score:**

I was happy to see more of a discussion with the links to RL and other approaches where explicit policy variables are added, and the contribution is real. Having said that, I still think the results are less surprising in light of existing results for e.g. doubly-robust policy optimization.

**Justification For Why Not Lower Score:**

The IV setup is a good way of thinking about "indirect" experiments. I believe the contribution makes a good use of this setup and it is likely to inspire further work.

---

### Decision · Program_Chairs · 2024-01-16

Accept (poster)